# STANDARDIZING STRUCTURAL CAUSAL MODELS

**Weronika Ormaniec**[*]
ETH Zürich, Switzerland
wormaniec@ethz.ch

**Scott Sussex**[*]
ETH Zürich, Switzerland
ssussex@ethz.ch

**Lars Lorch**[*]
ETH Zürich, Switzerland
llorch@ethz.ch

**Bernhard Schölkopf**
MPI for Intelligent Systems, Tübingen, Germany
bs@tuebingen.mpg.de

**Andreas Krause**
ETH Zürich, Switzerland
krausea@ethz.ch

## ABSTRACT

Synthetic datasets generated by structural causal models (SCMs) are commonly used for benchmarking causal structure learning algorithms. However, the variances and pairwise correlations in SCM data tend to increase along the causal ordering. Several popular algorithms exploit these artifacts, possibly leading to conclusions that do not generalize to real-world settings. Existing metrics like Var-sortability and $R^2$-sortability quantify these patterns, but they do not provide tools to remedy them. To address this, we propose *internally-standardized structural causal models (iSCMs)*, a modification of SCMs that introduces a standardization operation at each variable during the generative process. By construction, iSCMs are not Var-sortable. We also find empirical evidence that they are mostly not $R^2$-sortable for commonly-used graph families. Moreover, contrary to the post-hoc standardization of data generated by standard SCMs, we prove that linear iSCMs are less identifiable from prior knowledge on the weights and do not collapse to deterministic relationships in large systems, which may make iSCMs a useful model in causal inference beyond the benchmarking problem studied here. Our code is publicly available at: https://github.com/werkaaa/iscm.

## 1 INTRODUCTION

Predicting the effects of interventions and policy decisions requires reasoning about causality. Consequently, scientific fields ranging from biology and earth sciences to economics and statistics are interested in modeling causal structure (Pearl, 2009; Maathuis et al., 2010; Imbens and Rubin, 2015; Runge et al., 2019). A wide array of causal discovery algorithms has been proposed with the goal of inferring causal structure from data (e.g., Squires and Uhler, 2022; Vowels et al., 2022). However, benchmarking these algorithms is challenging, since real-world datasets with an agreed-upon, ground-truth causal structure are rare (e.g., Sachs et al., 2005; see Mooij et al., 2020). The community predominantly relies on synthetic data for evaluating structure learning algorithms, where observations are generated according to a predetermined causal structure and system mechanisms. The inferred causal structures can then be directly compared to the ground truth. To generate synthetic data, it is common practice to sample from structural causal models with additive noise (SCMs) (Reisach et al., 2021). Unless stated otherwise, this work considers SCMs in which the variance scale of the additive noise is the same for all variables, a typical simplification made in benchmarking.

Under common benchmarking practices, synthetic datasets generated by SCMs contain patterns that are directly exploitable to make structure discovery easier. We will refer to such patterns as *artifacts*. In SCMs, the pairwise correlations between variables tend to increase along the causal ordering, since variance builds up downstream and, as a result, the proportion of the variance driven by the additive noise vanishes (Figure 1a). Reisach et al. (2024) characterize this phenomenon through an increase of the coefficients of determination ($R^2$) of the variables regressed on all others. Crucially, this artifact occurs both in the raw data and when shifting and scaling (standardizing) the variables to have zero mean and unit variance. One of the implications is that downstream causal dependencies in SCMs become effectively deterministic, especially in large-scale systems. As Reisach et al. (2024) demonstrate, simple causal discovery baselines can perform competitively on benchmarks of this kind by directly exploiting this phenomenon. This makes SCMs alone in their general definition

---

[*]Equal contribution.

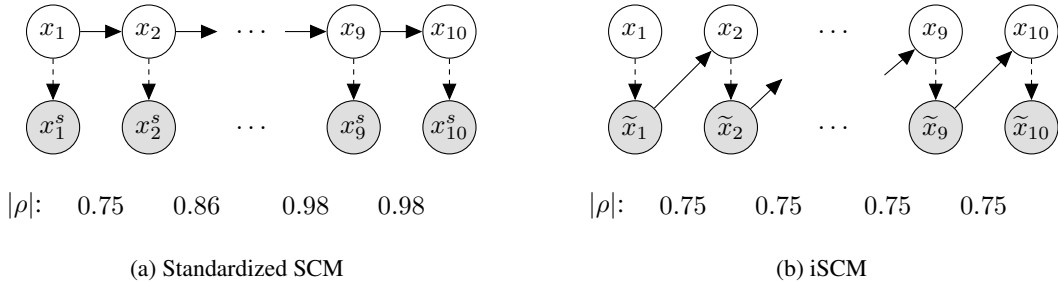

$|\rho|$:   0.75    0.86    0.98    0.98         $|\rho|$:   0.75    0.75    0.75    0.75

(a) Standardized SCM                    (b) iSCM

Figure 1: **Standardizing SCMs two ways.** Generative process for a chain graph of (a) standard SCMs, with data $\mathbf{x}$ standardized post-hoc, and (b) SCMs with standardization performed during the generative process (iSCMs). Dashed arrows indicate z-standardization. Solid arrows indicate linear functions with weights from $\text{Unif}_{\pm}[0.5, 2.0]$ and additive noise from $\mathcal{N}(0, 1)$. We report absolute correlations $|\rho|$ of two consecutive observed variables, (a) $x_j^s$ and $x_{j+1}^s$, or (b) $\widetilde{x}_j$ and $\widetilde{x}_{j+1}$, averaged over 100,000 models. In standard SCMs (a), correlations tend to increase along the causal ordering.

possibly insufficient for benchmarking. Ultimately, evaluating on synthetic data with these patterns could lead to conclusions that do not generalize as expected to real-world scenarios.

In this work, we propose a simple modification of SCMs that stabilizes the data-generating process and thereby removes exploitable covariance artifacts. Our models, denoted *internally-standardized SCMs (iSCMs)*, introduce a standardization operation at each variable during the generative process (Figure 1b). In Section 4, we provide a theoretical motivation for this idea by studying linear iSCMs. We prove that, contrary to SCMs, the causal dependencies of iSCMs under mild assumptions never collapse to deterministic mechanisms as the graph size becomes large. Moreover, we formalize the correlation artifact commonly observed in benchmarks by proving that linear SCM structures in a Markov equivalence class (MEC) are partially identifiable for certain graph classes, given weak prior knowledge on the weight distribution of the ground-truth SCM. Most importantly, we show that this is not the case for the corresponding iSCMs. In Section 5, we empirically demonstrate that the baselines proposed in Reisach et al. (2021; 2024) are unable to exploit covariance artifacts in iSCMs, while practical classes of causal discovery algorithms are still able to learn causal structures in both linear and nonlinear systems. Our findings reveal that SCM artifacts affect structure learning both positively and negatively, making iSCMs a practical tool, alongside SCMs, for disentangling the drivers of causal discovery performance of different algorithms in practice.

## 2   BACKGROUND AND RELATED WORK

We begin by introducing structural causal models and the problem of causal structure learning, before discussing how synthetic data is often generated for evaluating structure learning algorithms. We then review existing works that study identifiability and patterns frequently present in synthetic data.

**Structural causal models**   A structural causal model (SCM) (Peters et al., 2017) of $d$ variables $\mathbf{x} = \{x_1, \ldots, x_d\}$ consists of a collection of structural assignments, each given by

$$x_i := f_i(\mathbf{x}_{\text{pa}(i)}, \varepsilon_i)\,, \tag{SCM}$$

where $\mathbf{x}_{\text{pa}(i)} \subseteq \mathbf{x} \setminus \{x_i\}$ are called the *parents* of $x_i$. Here, $f_i$ are arbitrary functions, and $\varepsilon_i$ are independent random variables that model exogenous noise (or unexplained variation). Together, they entail a joint probability distribution $p(\mathbf{x})$ over the variables $\mathbf{x}$. It is common to consider SCMs with additive noise, e.g., with linear functions $f_i$, as given by

$$f_i(\mathbf{x}_{\text{pa}(i)}, \varepsilon_i) = \mathbf{w}_i^\top \mathbf{x}_{\text{pa}(i)} + \varepsilon_i\,, \tag{1}$$

where $w_{i,j} \in \mathbb{R}$ denotes the weight from $j \in \text{pa}(i)$ to $i$. The structural assignments in (SCM) induce a causal graph $\mathcal{G} = (\mathcal{V}, \mathcal{E})$ over the variables $x_i$, which is assumed to be acyclic. Specifically, the directed acyclic graph (DAG) $\mathcal{G}$ has vertices $v_i \in \mathcal{V}$ for every $x_i \in \mathbf{x}$ and a directed edge $(i, j) \in \mathcal{E}$ if $x_i \in \mathbf{x}_{\text{pa}(j)}$. We will explicitly distinguish this DAG $\mathcal{G}$ and its vertices $\mathcal{V}$ from the variables $\mathbf{x}$. The *skeleton* of $\mathcal{G}$ denotes $\mathcal{G}$ with all edges undirected. If the skeleton of $\mathcal{G}$ is acyclic, we call $\mathcal{G}$ a *forest*.

**Structure learning and benchmarking**   Given a set of *i.i.d.* observations from the probability distribution $p(\mathbf{x})$ induced by an unknown SCM, causal structure learning aims to infer the causal

graph $\mathcal{G}$ underlying the SCM. In this work, we focus on structure learning from observational data and only consider SCMs with no latent confounders. Because it is difficult to obtain the true $\mathcal{G}$ for many real-world datasets, it is common to evaluate structure learning algorithms on synthetic data where $\mathcal{G}$ is known. A ubiquitous approach is to sample a DAG $\mathcal{G}$, then SCM functions defined over $\mathcal{G}$, and finally a dataset from this SCM, with the goal of later recovering $\mathcal{G}$ from the data. It is common to consider $\varepsilon_i$ with mean 0 and fixed variance (often 1), and for linear systems, to sample each $w_{i,j}$ uniformly and *i.i.d.* with support bounded away from 0 (Shimizu et al., 2011; Peters and Bühlmann, 2014; Zheng et al., 2018; Yu et al., 2019; Lachapelle et al., 2020; Zheng et al., 2020; Ng et al., 2020; Reisach et al., 2021; Lorch et al., 2022; Reisach et al., 2024). There exist alternative benchmarking strategies with domain-specific simulators (Schaffter et al., 2011; Dibaeinia and Sinha, 2020).

**Data standardization and artifacts of SCMs**   Previous work shows that generating data as described above can lead to strong artifacts. Reisach et al. (2021) observe that the variance of variables tends to increase along the topological ordering of $\mathcal{G}$. This leads to the Var-SORTNREGRESS baseline, which sorts variables based on their empirical variance and then performs sparse regression to infer $\mathcal{G}$. Seng et al. (2024) show that structure learning algorithms minimizing an MSE-based loss (e.g., Zheng et al., 2018) can identify $\mathcal{G}$ under similar conditions. Therefore, Reisach et al. (2021) propose using standardization (Figure 1a) to remove this variance artifact from benchmarks. Specifically, they first sample all $x_i$ according to a standard SCM and then *post-hoc* transform the variables as

$$x_i^s := \frac{x_i - \mathbb{E}[x_i]}{\sqrt{\mathrm{Var}[x_i]}} \,, \qquad \text{(Standardized SCM)}$$

such that our observations correspond to samples from $p(\mathbf{x^s})$. Standardization, however, only removes the variance artifact. Even in standardized SCMs, the fraction of a variable's variance that is explained by all others, measured by the coefficient of determination $\mathrm{R}^2$, tends to increase along the topological ordering (Reisach et al., 2024). $\mathrm{R}^2$-SORTNREGRESS exploits this correlation artifact analogously to Var-SORTNREGRESS. Existing heuristics aiming to avoid the variance accumulation adjust the sampling process of $f_i$, but they ultimately limit the causal dependencies that can be modeled, e.g., to certain levels of correlations among the observed $\mathbf{x}$ (Mooij et al., 2020) or a constant proportion of variance explained by the parents $\mathbf{x}_{\mathrm{pa}(i)}$ (Squires et al., 2022) and fail to induce data free from both artifacts (Appendix D.1). To our knowledge, there are currently no general methods for generating SCM data without strong correlation artifacts or significant limitations on the functions $f_i$ and noise $\varepsilon_i$.

**Identifiability**   Given a class of SCMs, there may be several SCMs with different causal graphs $\mathcal{G}$ that entail the same distribution $p(\mathbf{x})$ (Peters et al., 2017). Thus, even with infinite observations from $p(\mathbf{x})$, we may be unable to identify the causal graph $\mathcal{G}$ that generated the observations. However, some identifiability results are known depending on the class of functions and noise distributions of the SCMs considered. For example, among all linear SCMs (1) with Gaussian noise $\varepsilon_i \sim \mathcal{N}(0, \sigma_i^2)$, the graph $\mathcal{G}$ can only be uniquely identified up to its MEC (Verma and Pearl, 2013). However, if the noise is Gaussian with equal variances $\sigma_i^2 = \sigma^2$ (Peters and Bühlmann, 2014) or the noise is non-Gaussian (Shimizu et al., 2006), $\mathcal{G}$ can be uniquely identified given $p(\mathbf{x})$.

In this work, we present, to our knowledge, the first (partial) identifiability result for *standardized* SCMs in the linear Gaussian case. Since standardization affects the *implied* noise scales, existing linear Gaussian identification results, which rely on $\sigma_i^2 = \sigma^2$, no longer hold when observing $p(\mathbf{x^s})$. Other identifiability results, e.g., based on non-Gaussian noise, do continue to hold for standardized SCMs (e.g., Shimizu et al., 2006). Our result concerns a setting with prior knowledge on the magnitudes of $\mathbf{w}$ in Equation (1), an assumption underlying common benchmarking practices. Under this setup, we show a stark difference in the identifiability of standardized SCMs and the iSCMs we propose, which provides a novel explanation for what we empirically observe in benchmarks.

## 3   SCMs WITH INTERNAL STANDARDIZATION

### 3.1   DEFINITION

We propose *internally-standardized SCMs (iSCMs)* as a modification to the standard data-generating process of SCMs. An iSCM $(\mathbf{S}, \mathcal{P}_{\boldsymbol{\varepsilon}})$ consists of $d$ pairs of assignments, where for each $i \in \{1, \ldots, d\}$,

$$x_i := f_i(\widetilde{\mathbf{x}}_{\mathrm{pa}(i)}, \varepsilon_i) \quad \text{and} \quad \widetilde{x}_i := \frac{x_i - \mathbb{E}[x_i]}{\sqrt{\mathrm{Var}[x_i]}} \qquad \text{(iSCM)}$$

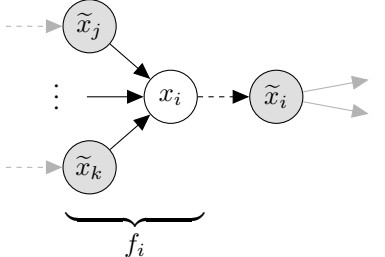

Figure 2: **Causal mechanisms in iSCMs.** The function $f_i$ modeling $x_i$ depends on the standardized $\widetilde{\mathbf{x}}_{\mathrm{pa}(i)}$. Dashing indicates z-standardization.

---
**Algorithm 1** Sampling from an iSCM
---
**Input**: DAG $\mathcal{G}$, noise distribution $\mathcal{P}_{\boldsymbol{\varepsilon}}$,
          functions $\{f_1, ..., f_d\}$
$\pi \leftarrow$ topological ordering of $\mathcal{G}$
**for** $i = 1$ to $d$ **do**
    $\varepsilon_{\pi_i} \sim \mathcal{P}_{\varepsilon_{\pi_i}}$
    $x_{\pi_i} \leftarrow f_{\pi_i}(\widetilde{\mathbf{x}}_{\mathrm{pa}(\pi_i)}, \varepsilon_{\pi_i})$
    $\widetilde{x}_{\pi_i} \leftarrow \dfrac{x_{\pi_i} - \mathbb{E}[x_{\pi_i}]}{\sqrt{\mathrm{Var}[x_{\pi_i}]}}$
**return** $[\widetilde{x}_1, \ldots, \widetilde{x}_d]$       $\triangleright \in \mathbb{R}^d$
---

with parents $\widetilde{\mathbf{x}}_{\mathrm{pa}(i)} \subseteq \widetilde{\mathbf{x}} \setminus \{\widetilde{x}_i\}$ of $\widetilde{x}_i$ in the underlying DAG. In the above, $f_i$ are general functions, and the exogenous noise variables $\boldsymbol{\varepsilon} = [\varepsilon_1, ..., \varepsilon_d] \sim \mathcal{P}_{\boldsymbol{\varepsilon}}$ are jointly independent, as for SCMs. The variables $x_i$ are latent, and the variables $\widetilde{x}_i$ are observed. Figure 2 illustrates the generative process. Algorithm 1 summarizes how to sample from (iSCM). If computing the population expectations and variances of $x_i$ is intractable, the empirical statistics obtained from $n$ samples can be used for standardization at each loop iteration of Algorithm 1.

**Motivation** By construction, iSCMs model observed variables with zero mean and unit marginal variance. Contrary to standard SCMs, iSCMs avoid the accumulation of variance downstream in the causal ordering that can occur in standard SCMs (see Figure 1) through the standardization operation. Because each variable $x_i$ only depends on the standardized variables $\widetilde{\mathbf{x}}_{\mathrm{pa}(i)}$, the relative scales of the noise distribution $\mathcal{P}_{\varepsilon_i}$ and the causal mechanisms $f_i$ are the same everywhere in the system and do not change, for example, downstream in the causal ordering. The causal mechansims of iSCMs are thus *scale-free*, in that the local interaction of mechanism $f_i$ and noise $\varepsilon_i$ occurs at a scale independent of the position of $x_i$ in the global ordering. This property makes iSCMs particularly useful for benchmarking, where random ground-truth models are commonly generated from a fixed distribution over functions $f_i$ and noise $\varepsilon_i$. Contrary to existing heuristics (Section 2), iSCMs model arbitrarily strong or weak causal dependencies and levels of cause-explained variance.

**Interventions** Analogous to standard SCMs, interventions in iSCMs can be defined as modifications of the structural assignments $f_i$ in (iSCM) (Figure 2), while keeping the standardization operation based on the observational distribution. When the population statistics for standardization are intractable, we first sample observational data to obtain empirical statistics. Since we do not study interventions in this work, we defer a further discussion of interventions in iSCMs to Appendix B.

**Units** When modeling a physical system, the functional mechanisms in standard SCMs have to account for the difference in units between the variables for the model to be *unit-covariant* (see Villar et al., 2023). A side-effect of internal standardization is that variables of iSCMs become *unit-less*, so iSCMs obey the passive symmetry of unit covariance by construction. Therefore, iSCMs naturally model both unit-less quantities and variables measured in different units, which can make them useful beyond benchmarking. Learned iSCMs would be invariant to the units chosen by the experimenter, similar to the physical world being independent of the mathematical models chosen to describe it.

### 3.2 IMPLIED SCMs

It is natural to investigate whether SCMs can generate the same observations as standardized SCMs or iSCMs, given the same causal graph $\mathcal{G}$ and exogenous variables $\boldsymbol{\varepsilon}$. In other words, can standardized SCMs and iSCMs be written as SCMs? For both models, the answer is yes. Specifically, we can express the generative process of $\mathbf{x}^{\mathbf{s}}$ in (Standardized SCM) and $\widetilde{\mathbf{x}}$ in (iSCM) as

$$x_i^s = g_i^s(\mathbf{x}_{\mathrm{pa}(i)}^s) + \theta_i^s \varepsilon_i \qquad \text{and} \qquad \widetilde{x}_i = \widetilde{g}_i(\widetilde{\mathbf{x}}_{\mathrm{pa}(i)}) + \widetilde{\theta}_i \varepsilon_i \,, \tag{2}$$

respectively, by moving the standardization operations into the causal mechanisms of the observables but leaving the DAG $\mathcal{G}$ and the variables $\boldsymbol{\varepsilon}$ unchanged. Appendix A describes how to construct these *implied causal mechanisms* $g_i^s$ and $\widetilde{g}_i$ and *implied noise scales* $\theta_i^s$ and $\widetilde{\theta}_i$. We refer to the above SCM form of a standardized SCM or an iSCM with additive noise as their ***implied (SCM) model***. Correspondingly, the implied SCMs have zero mean and unit variance. The notion of implied

SCMs is powerful, because it enables us to analyze standardized SCMs and iSCMs as SCMs, and it sheds light on the performance of structure learning algorithms that assume unstandardized SCMs to underlie the generative process of the data (e.g., Shimizu et al., 2011; Zheng et al., 2018; Yu et al., 2019; Lachapelle et al., 2020; Zheng et al., 2020).

To provide a first characterization of standardized SCMs and iSCMs, our theoretical analyses focus on systems where $f_i$ are linear functions with additive, zero-mean noise as given by Equation (1). As a stepping stone for this analysis, we use an analytical expression for the covariance of linear SCMs, whose variables have unit variance by construction, without any form of standardization:

**Lemma 1** (Covariance in linear SCMs with unit marginal variances). *Let* $\mathbf{x}$ *be modeled by a linear SCM defined by* (1) *with DAG* $\mathcal{G}$ *that satisfies* $\mathrm{Var}[x_i] = 1$. *Then, the covariance* $\mathrm{Cov}[x_i, x_j]$ *is the sum of products of the weights along all unblocked paths between the nodes of* $x_i$ *and* $x_j$ *in* $\mathcal{G}$. *Specifically, for any* $i, j \in \{1, ..., d\}$ *such that* $i \neq j$, *it holds that*

$$\mathrm{Cov}[x_i, x_j] = \sum_{p_{j \leftrightarrow i} \in P_{j \leftrightarrow i}} \prod_{(l,m) \in p_{j \leftrightarrow i}} w_{l,m} \,, \tag{3}$$

*where* $P_{j \leftrightarrow i}$ *are all unblocked paths from* $x_j$ *to* $x_i$ *in* $\mathcal{G}$, *and* $(l, m) \in p_{j \leftrightarrow i}$ *indicates that the directed edge* $(l, m)$ *is part of the path* $p_{j \leftrightarrow i}$.

This lemma, also called the *trek rule*, is originally due to Wright (1934). We give a proof in Appendix C.2. Since the implied SCMs of linear standardized SCMs and iSCMs are linear SCMs, the setting of Lemma 1 applies precisely to the SCM forms of both models. Thus, Lemma 1 enables us to study the covariances in standardized SCMs and iSCMs, and as we show next, derive conditions for the (non)identifiability of their DAGs $\mathcal{G}$ from the observational distribution.

## 4 ANALYSIS

In this section, we give two theoretical results that support the suitability of iSCMs over standard SCMs for causal discovery benchmarking. First, we prove the general case of Figure 1. Contrary to standardized SCMs, iSCMs do not degenerate towards deterministic implied SCM mechanisms in deep graphs. Moreover, we prove that the DAGs of linear iSCMs cannot be identified beyond their MEC, assuming the DAG is a forest, even if the support of $\mathbf{w}$ is known. Crucially, we also show that this is not generally true for standardized SCMs. This suggests that algorithms can less easily game benchmarks based on linear iSCMs when knowing the data-generating process. For all results, we consider linear SCMs (1) with zero-mean additive noise and equal noise variances. All results are at the population level, so assume we know $p(\mathbf{x^s})$ or $p(\widetilde{\mathbf{x}})$. Proofs are given in Appendix C.

### 4.1 BEHAVIOR WITH INCREASING GRAPH DEPTH

Standardized SCMs tend towards increasing correlations between adjacent nodes down the topological ordering. This correlation artifact makes standardized SCMs problematic for benchmarking, because it may not be a property we expect to underlie real data. Reisach et al. (2024) show, under some assumptions on $\mathbf{w}$, that the dependencies in standardized SCMs become *deterministic* with increasing graph depth. This implies that any exogenous variation $\varepsilon_i$ vanishes lower down in the system. Unless prior domain knowledge leads us to assume this holds in applications of interest, it may not be desirable to implicitly bias structure learning benchmarks towards such systems. For example, if the causal ordering represents time (Pamfil et al., 2020), the mechanisms of standardized SCMs are unable to model or characterize time-invariant or stable processes. Moreover, if we expect causal mechanisms to be independent (Schölkopf, 2022), the qualitative behavior of a causal mechanism should not provide information about its position in the topological ordering relative to other mechanisms, as it would in SCMs. Reisach et al. (2024) show that baselines like $\mathrm{R}^2$-SORTNREGRESS can perform competitively on benchmarks by exploiting this artifact (Section 2).

iSCMs do not tend towards determinism with increasing graph depth (Figure 1b). In standardized SCMs, the correlations increase downstream, because the marginal variances of the underlying SCM increase with node depth, while the variance scale is fixed (Reisach et al., 2021). Thus, for large $i$, the variance scale of $x_{i-1}$ becomes large relative to the scale of $\varepsilon_i$, and the correlation of $x_i$ and $x_{i-1}$ tends towards 1. Since $x_i^s$ and $x_{i-1}^s$ are just standardized versions of these variables, they maintain

the same correlation. iSCMs avoid this by standardizing internally, which scales the variance of any parents in a mechanism $f_i$ to 1, modulating the relative variance of $\varepsilon_i$ and $\mathbf{x}_{\mathrm{pa}(i)}$. In the following, we formalize this result for general graphs by bounding the fraction of cause explained variance (CEV). The fraction of CEV for $x_i$ is the proportion of $\mathrm{Var}[x_i]$ explained by its causal parents and given by

$$\mathrm{CEV}_{\mathrm{f}}[x_i] = 1 - \frac{\mathrm{Var}[x_i - \mathbb{E}[x_i|\mathbf{x}_{\mathrm{pa}(i)}]]}{\mathrm{Var}[x_i]} \,. \tag{4}$$

The following results shows that we can bound the fraction of CEV for any variable in a linear iSCM:

**Theorem 2** (Bound on $\mathrm{CEV}_{\mathrm{f}}$ in linear iSCMs)**.** *Let $\mathbf{x}$ be modeled by a linear iSCM (1) with DAG $\mathcal{G}$ and additive noise of equal variances $\mathrm{Var}[\varepsilon_i] = \sigma^2$. Suppose any node in $\mathcal{G}$ has at most $m$ parents and $w = \max_{i,j \in \{1,...,d\}} |w_{i,j}|$. Then, for any $i \in \{1, ..., d\}$, the fraction of CEV for $\widetilde{x}_i$ is bounded as*

$$\mathrm{CEV}_{\mathrm{f}}[\widetilde{x}_i] \leq 1 - \frac{\sigma^2}{m^2 w^2 + \sigma^2} \,.$$

Since the fraction of CEV is bounded, iSCMs are guaranteed not to collapse to determinism in large systems, alleviating several of the concerns with (standardized) SCMs discussed above.

## 4.2 IDENTIFIABILITY

Figure 1a illustrates that the pairwise correlations in SCMs over chain graphs depend on the position in the topological ordering. This can allow algorithms like $\mathrm{R}^2$-SORTNREGRESS to infer the graph. By contrast, Figure 1b shows that iSCMs do no exhibit this pattern, with correlations between variables not increasing the identifiability of any part of the system.

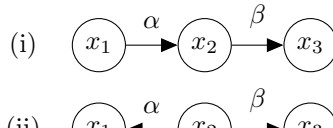

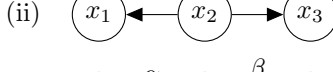

(a) DAGs with edge weights $\alpha$ and $\beta$

In the following, we formalize this phenomenon for forests, that is, all DAGs with acyclic skeletons (Section 2). Specifically, we prove two results concerning the identifiability of the DAG $\mathcal{G}$ from the observational distribution, for standardized SCMs and iSCMs. This makes our finding the first identifiability result for *standardized* SCMs. While not every DAG is a forest, DAGs have forests as subgraphs and resemble forests as sparsity increases, thus providing us with intuition for generally sparse systems (e.g., Alon and Spencer, 2016, Chapter 11).

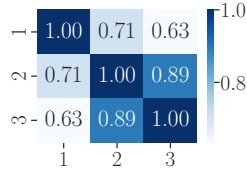

(b) Cov. matrix of linear iSCMs

Our first result leverages the observation that, for standardized SCMs, many DAGs in an MEC are infeasible given $p(\mathbf{x^s})$ when their edge directions are not consistent with the direction of increasing absolute covariance. To illustrate this idea, suppose our goal is to distinguish between the DAGs in the MEC $\widetilde{\mathcal{G}} = \{(\mathrm{i}), (\mathrm{ii}), (\mathrm{iii})\}$ in Figure 3a. We overload notation and denote the weights of the edges $\alpha$ and $\beta$ regardless of orientation. For standardized SCMs, we can apply Lemma 1 to the implied SCM of graph (i) to obtain the covariances

Figure 3: **iSCMs with the same covariance matrix.** (a) DAGs in an MEC with the same edge weights. (b) Covariance matrix for all linear iSCMs in (a) when $\alpha = 1$, $\beta = 2$.

$$\mathrm{Cov}[x_1^s, x_2^s] = \frac{\alpha}{\sqrt{\alpha^2+1}} \qquad \text{and} \qquad \mathrm{Cov}[x_2^s, x_3^s] = \beta \sqrt{\frac{\alpha^2+1}{\beta^2(\alpha^2+1)+1}} \,.$$

See Appendix C.4.1. Together, both expressions imply that standardized SCMs with DAG (i) satisfy

$$|\mathrm{Cov}[x_1^s, x_2^s]| < |\mathrm{Cov}[x_2^s, x_3^s]| \quad \Longleftrightarrow \quad \frac{\alpha^2}{\alpha^2+1} < \beta^2 \,. \tag{5}$$

If $|\beta| \geq 1$, then the right-hand side of Equation (5) is always true. In this case, the absolute covariance increases from $x_1$ to $x_3$ in all standardized SCMs with DAG (i). By symmetry, the covariance in SCMs with DAG (iii) increases from $x_3$ to $x_1$ when $|\alpha| \geq 1$. Therefore, if both weights are greater than 1, the absolute covariance increases downstream in all SCMs of (i) and (iii). This implies that, among (i) and (iii), only the DAG whose edges align with the covariance ordering in $p(\mathbf{x^s})$ can induce $p(\mathbf{x^s})$. Irrespectively, the DAG (ii) remains plausible. We can extend the intuition of this 3-variable example to identify almost all edges in any forest MEC:

**Theorem 3** (Partial identifiability of standardized linear SCMs with forest DAGs). *Let $\mathbf{x}^s$ be modeled by a standardized linear SCM (1) with forest DAG $\mathcal{G}$, additive noise of equal variances $\mathrm{Var}[\varepsilon_i] = \sigma^2$, and $|w_{i,j}| > 1$ for all $i \in pa(j)$. Then, given $p(\mathbf{x}^s)$ and the partially directed graph $\tilde{\mathcal{G}}$ representing the MEC of $\mathcal{G}$, we can identify all but at most one edge of the true DAG $\mathcal{G}$ in each undirected connected component of the MEC $\tilde{\mathcal{G}}$.*

Our proof of Theorem 3 considers each undirected component separately from the rest of the MEC $\tilde{\mathcal{G}}$. Hence, the identifiability result extends to undirected tree components of arbitrary, non-forest MECs as well. Theorem 3 shows that, when using standardized SCM data for benchmarking, algorithms can use pairwise correlations to orient additional edges correctly. The weights assumption of Theorem 3 is relevant to causal discovery benchmarking, because weights are often sampled *i.i.d.* from intervals bounded away from 0 (Section 2). Hence, empirical evaluations may render standardized linear SCMs identifiable only through the design of their weights distribution. In the following, we show that, under similar conditions, iSCMs are more difficult to identify from their MEC. In the 3-variable example above, we can show that the observational distribution of iSCMs is the same for all DAGs (i), (ii), and (iii) when the weights $\alpha$ and $\beta$ are shared over the corresponding edges in the MEC (Figure 3b; see Appendix C.4.1). This result generalizes to forests:

**Theorem 4** (Nonidentifiability of linear Gaussian iSCMs with forest DAGs). *Let $\widetilde{\mathbf{x}}$ be modeled by a linear iSCM (1) with forest DAG $\mathcal{G}$ and additive Gaussian noise of equal variances $\mathrm{Var}[\varepsilon_i]$. Then, for every DAG $\mathcal{G}'$ in the MEC of $\mathcal{G}$, there exists a linear iSCM with DAG $\mathcal{G}'$ that has the same observational distribution as $\widetilde{\mathbf{x}}$, the same noise variances, and the same weights on the corresponding edges in the MEC.*

Our proof consists of showing that the covariance matrices of these systems are equal. For linear Gaussian iSCMs, this then implies that their observational distributions are identical. Theorem 4 thus shows that additional knowledge of the weight distribution in a benchmark does not allow identifying any additional edges beyond the MEC. By contrast, Theorem 3 shows that, for standardized SCMs, lower-bounding the weight magnitudes is sufficient for identifying most of the graph from its MEC. Without standardization, $\mathcal{G}$ is fully identified from its observational distribution under even weaker assumptions (Peters and Bühlmann, 2014). Importantly, Theorem 4 does not generalize to arbitrary graphs beyond forests. Appendix C.4.2 provides a counterexample involving a 3-node skeleton. As we study in the next section, this implies that causal structure can still be learned from nontrivial iSCMs. However, DAGs in benchmarks are often sparse, so we expect the implications of our identifiability results to capture relevant parts of empirical phenomena in benchmarking settings.

## 5    EXPERIMENTAL RESULTS

Our analyses suggest that iSCMs address shortcomings of naive standardization, in particular, when sampling each $f_i$ and $\varepsilon_i$ from the same distribution, as common in benchmarking. In this section, we now also provide empirical evidence that iSCMs do not contain the covariance artifacts of SCMs. This makes iSCMs a useful tool for disentangling, alongside SCMs, which data patterns drive causal discovery in practice. To show this, we benchmark the SORTNREGRESS baselines and a suite of popular structure learning algorithms to gain insights into how their performance varies when benchmarked on standardized SCMs and iSCMs. Appendix E provides complete details of the experimental setup.

### 5.1    $R^2$-SORTABILITY

Reisach et al. (2024) introduce the $R^2$-sortability metric to evaluate the correlation artifact underlying a dataset. $R^2$-sortability measures (rescaled to $[0, 1]$) the association between the variables' causal ordering and the $R^2$ coefficients obtained from regressing each variable onto all others (Appendix D.2). An $R^2$-sortability close to $0.5$ suggests that the $R^2$ coefficients from regression contain no information about the causal ordering. Conversely, an $R^2$-sortability of $0$ or $1$ implies that the causal ordering can be completely identified from this information. The metric gives rise to the $R^2$-SORTNREGRESS baseline described in Section 2. Reisach et al. (2024) show that $R^2$-sortability in SCMs is driven by an interplay of graph connectivity and the weight distribution of $f_i$.

Figure 4 summarizes the $R^2$-sortability statistics for linear SCM and iSCM data. We write $\mathrm{ER}(d, k)$ and $\mathrm{SF}(d, k)$ to denote Erdős-Rényi and scale-free graphs of size $d$ and (expected) degree $k$, respectively (see Appendix E.2 for details). We find that iSCMs generate datasets that are not $R^2$-sortable

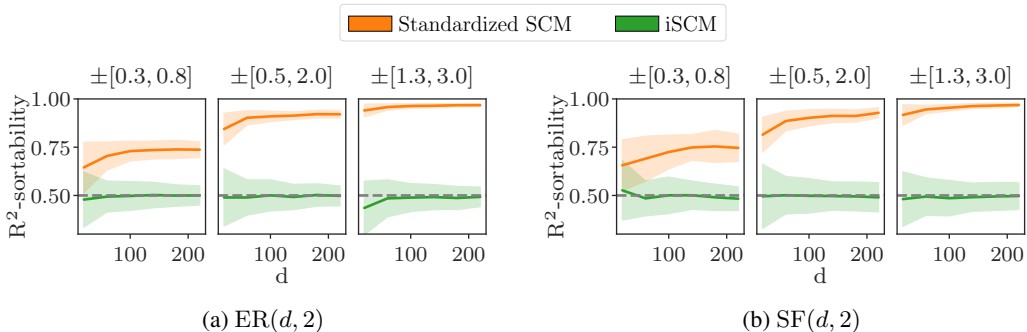

(a) $\mathrm{ER}(d, 2)$        (b) $\mathrm{SF}(d, 2)$

Figure 4: $R^2$**-sortability for different graph sizes.** Linear standardized SCMs and iSCMs with $\varepsilon_i \sim \mathcal{N}(0, 1)$ and weights drawn from uniform distributions with supports given above each plot. For every model, we evaluate 100 systems and $n = 1000$ samples each. Lines and shaded regions denote mean and standard deviation. Datasets that satisfy $R^2$-sortability $= 0.5$ (dashed) are not $R^2$-sortable.

($R^2$-sortability $\approx 0.5$) and thus artifact-free while sampling over common graph structures (e.g., Zheng et al., 2018; Yu et al., 2019; Reisach et al., 2021). Conversely, standardized SCMs generate datasets that are strongly $R^2$-sortable ($|R^2$-sortability $- 0.5| \gg 0$). Since $R^2$-sortability can be exploited for causal discovery, iSCM data serves as a test for evaluating whether algorithms utilize any data properties beyond the association between $R^2$ and the causal ordering in SCMs. Our results do not exclude the possibility of iSCM configurations that still produce $R^2$-sortable datasets. However, we show empirically that, for commonly-used $\mathcal{G}$, $\mathcal{P}_\varepsilon$, and $\mathbf{w}$, iSCM datasets are not $R^2$-sortable with high probability. Appendix D.1 reports the sortability metrics of the existing heuristics in Section 2, showing that neither mitigate both Var- and $R^2$-sortability. Appendix F provides results for denser graphs.

## 5.2 STRUCTURE LEARNING

Under the same weight and noise distributions, standardized SCMs and iSCMs have different implied SCMs and generate qualitatively different datasets. Here, we study how this affects causal structure learning in practice. For this, we evaluate Var- and $R^2$-SORTNREGRESS (SR) (Reisach et al., 2021; 2024) as well as an SR variant that uses a random ordering (Random SR). In addition, we evaluate representative algorithms from several approaches to learning structure from (co)variance information. NOTEARS by Zheng et al. (2018) leverages continuous optimization to minimize an MSE loss, which is affected by noise scaling (Loh and Bühlmann, 2014; Seng et al., 2024). GOLEM (Ng et al., 2020), similar to NOTEARS, formulates causal discovery as a continuous optimization problem. Its EV and NV versions assume equal and potentially unequal noise scales, respectively. CAM (Bühlmann et al., 2014) searches over causal orderings and performs sparse nonlinear regression to find the parents, while also estimating the noise scales. PC (Spirtes and Glymour, 1991) and GES (Chickering, 2002) are approaches based on statistical independence testing and greedy search, respectively. Finally, AVICI by Lorch et al. (2022) predicts graphs using a model pretrained on simulated data and is thus optimized to exploit any artifacts that improve predictive accuracy. To investigate its susceptibility to artifacts, we evaluate the public model checkpoints trained on standardized SCMs.

Figure 5 summarizes the results for linear and nonlinear systems with Gaussian noise (see Figure 18, Appendix F for non-Gaussian systems). The nonlinear mechanisms $f_i$ are samples from a Gaussian process with squared exponential kernel. As expected, Var-SORTNREGRESS performs best when SCMs are not standardized. Likewise, $R^2$-SORTNREGRESS performs better on SCMs and standardized SCMs, as iSCMs have $R^2$-sortability close to 0.5 (Section 5.1). AVICI shows the same trend, suggesting it may indeed be exploiting the correlation artifacts present in its training distribution. Like Reisach et al. (2021), we find that NOTEARS performs best on unstandardized data. However, and more interestingly, NOTEARS also performs better on iSCMs than on standardized SCMs, especially in linear and larger systems. As we investigate later on, this gap may be explained by the fact that the implied models of standardized SCMs violate the assumptions of NOTEARS more strongly than iSCMs. Overall, we find GOLEM-EV shows the same patterns as NOTEARS, severely underperforming on standardized SCMs and slightly improving the predictive accuracy on iSCMs. CAM and GOLEM-NV, which do not assume equal noise scales, perform equally well or better on standardized data, respectively, and generally better on iSCMs. The poor performance of GOLEM-NV

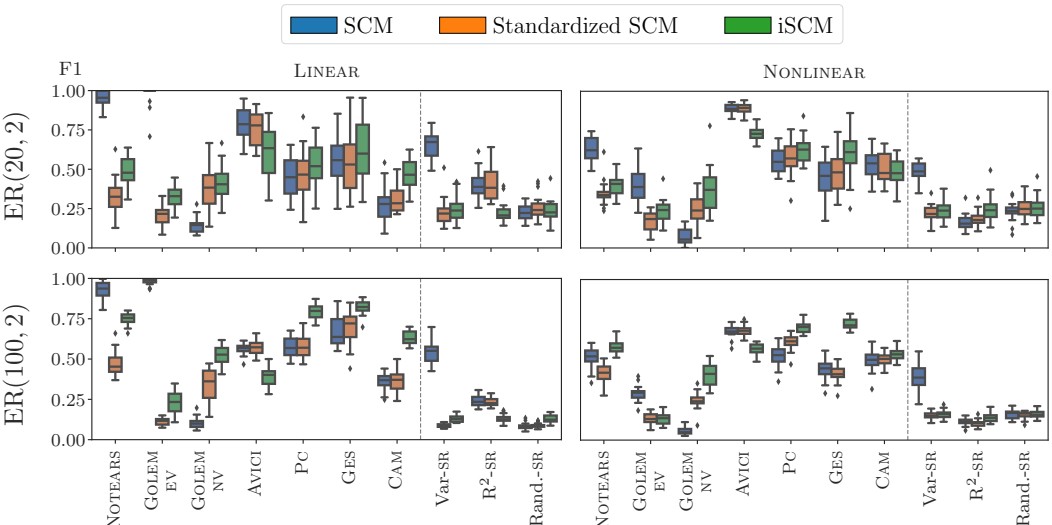

Figure 5: **Structure learning performance on SCM and iSCM data.** F1 scores for recovering the edges of the true graph. Box plots show median and interquartile range (IQR). Whiskers extend to the largest value inside $1.5 \times$ IQR from the boxes. Left (right) column shows results for linear (nonlinear) causal mechanisms with additive noise $\varepsilon_i \sim \mathcal{N}(0, 1)$ and $w_{i,j} \sim \mathrm{Unif}_{\pm[0.5, 2.0]}$ (Appendix E). For every model, we evaluate 20 systems each using $n = 1000$ data points.

on unstandardized SCMs was also observed by Reisach et al. (2021). In addition, for approaches based on discrete search, we find that, in particular on large systems, the PC and GES algorithms perform better on iSCMs. Overall, performance differences tend to be more pronounced for linear systems, where the downstream variance accumulation in SCMs is unbounded. Appendix F reports the results for the structural Hamming distance (SHD) and different weight ranges.

**Properties of the implied SCMs** When standardizing SCM data, the implied SCM corresponds to the SCM that could have generated the observations. Therefore, algorithms assuming that unstandardized SCMs generated the data will be susceptible to any assumption violations of the implied SCM, such as assumptions about the exogenous noise. Figure 6 (bottom) shows the distribution of inverse implied noise scales $1/\theta_i^2$ for the variables of the implied models (see Equation 2). Since $\mathrm{Var}[\varepsilon_i] = 1$ in our experiments, these inverse squared noise scales are equal to the inverse variances of the full additive noise terms. We find that standardized SCMs induce inverse noise scales that are orders of magnitude greater than those of iSCMs. This distribution is essentially the footprint of the determinism in the depth limit discussed in Section 4.1. This observation also provides empirical support for our earlier explanation for the improved performance of the PC algorithm on iSCM data. The modes at $1/\theta_i^2 = 1$ and at $1/\theta_i^2 > 1$ in the iSCM plot correspond to root and non-root nodes, respectively.

Figure 6 (top) shows the performance of NOTEARS when isolating the noise properties of the implied models from the fact that standardized SCMs and iSCMs are not Var-sortable. For this, we construct SCMs that have the marginal variances (and Var-sortability, here 0.99 on average) of unstandardized

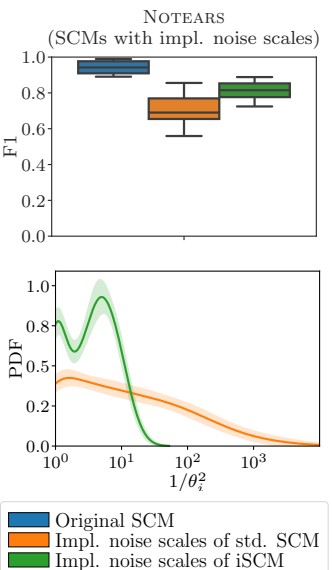

Figure 6: **Implied Noise.** Bottom panel shows the distribution over inverse implied noise scales in the implied SCMs for ER$(100, 2)$ graphs (kernel density estimate). Lines and shading denote mean and standard deviation. Top panel shows the performance of NOTEARS on systems with these noise scale statistics but the same Var-sortability as SCMs (see Appendix E.2 and E.5).

SCMs but the noise variances of the implied models by correcting their weights (see Appendix E.5). NOTEARS performs better in such systems, suggesting that (i) the noise statistics may indeed explain the performance difference on iSCM data, and (ii) Var-sortability may not be the only reason why NOTEARS performs significantly worse on standardized data (Reisach et al., 2021). Conversely, when the weight ranges of (standardized) SCMs are smaller, the phenomenon of exploding marginal variances is less pronounced (Figure 19 in Appendix F.3). In this case, we indeed find that NOTEARS performs similarly on standardized SCMs and iSCMs (Figure 15, left, in Appendix F.1).

This sheds light on previous benchmarking results, where MSE-based algorithms perform below expectations despite perhaps not intending to evaluate the algorithms under model mismatch (e.g., Reisach et al., 2021; Kaiser and Sipos, 2021). For the MSE loss, Loh and Bühlmann (2014) and Seng et al. (2024) show that smaller ratios of noise variances increase the magnitude of weights required for the true DAG to be the unique minimizer. The MSE loss ultimately does not account for the inverse variance factor in the Gaussian noise likelihood. Overall, the statistics of the implied models of standardized SCMs are empirically further from SCMs with equal noise variances than their iSCM counterparts.

## 6 CONCLUSIONS

We describe the iSCM, a one-line modification of the SCM that modulates the scale of interaction between the causal mechanism $f_i$ and noise $\varepsilon_i$ at each variable $x_i$. Through several theoretical and experimental results, we study its properties in relation to standard SCMs and its ramifications for benchmarking causal discovery algorithms. To conclude, we highlight the following key takeaways:

**Standardizing during the generative process removes sortability artifacts.** When the functions $f_i$ and the noise $\varepsilon_i$ are, for example, sampled *i.i.d.* for each variable $x_i$, SCMs exhibit artifacts that are not removed when shifting and scaling the generated data. Our results in Section 5 show that iSCMs are effective at removing Var- and $R^2$-sortability. This makes iSCMs a useful complement to structure learning benchmarks with SCMs, enabling a specific evaluation of the ability of algorithms to transfer to real-world settings that do not exhibit $R^2$ artifacts. Despite the removed sortability artifacts, causal discovery algorithms are able to infer nontrivial structure from iSCM data (Figure 5).

**Standardizing post-hoc can lead to partial identifiability and degenerate implied SCMs.** Scaling the units of SCM data is not innocuous. Theorem 3 shows that mild knowledge on the distribution of $f_i$ can identify edges in standardized SCMs that are typically not identifiable from observational data. To our knowledge, our result is the first concerning the identifiability of $\mathcal{G}$ from the standardized observational distribution of linear SCMs. This may make benchmarks, where similar assumptions on $f_i$ often hold, trivial under standardized SCMs. Moreover, Figure 6 shows that standard SCMs can collapse to modeling near-zero exogenous noise. Theorems 2 and 4 demonstrate that neither property appears in the analogous iSCMs. Ultimately, (non)identifiability may be either a feature or bug, depending on whether assumptions are verifiable in practice or a priori known during evaluation.

**iSCMs are stable and scale-free, making them useful models beyond benchmarking.** Beyond data generation, the stable generative process of iSCMs might also provide insights for modeling, e.g., large, temporal (Kilian, 2013; Pamfil et al., 2020) or physical systems. In iSCMs, the scale of a causal mechanism $f_i$ and its unexplained variation $\varepsilon_i$ are both unit-less and independent from its position in the causal ordering (Section 3). If we think of each structural assignment as a physical mechanism, energy conservation must be respected, since a mechanism can only output as much energy as it receives from its inputs (including unexplained noise). Standardization may thus not be completely unrealistic, since it naturally bounds the output scale of every mechanism.

Since each iSCM implies a standard SCM, iSCMs can also be viewed as a reparameterization of SCMs that facilitates modeling and learning the functions $f_i$ on the same scale, e.g., under a shared prior or level of regularization. Conceptually, iSCMs are related to batch normalization (Ioffe and Szegedy, 2015), a technique used to stabilize the optimization of neural networks, which compose sequences of functions like SCMs, by adding internal standardization. Overall, these properties may make the iSCM a useful structural equation model beyond the benchmarking problem studied here.

## REPRODUCIBILITY STATEMENT

To facilitate reproducibility, we provide code, configuration files, and the commands used to obtain all the experimental results in this manuscript as supplementary material. They are also available at: https://github.com/werkaaa/iscm. In Appendix E , we describe the experimental setup, including the computational resources and wall time used to produce the results. Finally, we provide detailed proofs of our theoretical results in Appendix C.

## ACKNOWLEDGEMENTS

This research was supported by the European Research Council (ERC) under the European Union's Horizon 2020 research and innovation program grant agreement no. 815943 and the Swiss National Science Foundation under NCCR Automation, grant agreement 51NF40 180545. This work was also supported by the German Federal Ministry of Education and Research (BMBF): Tübingen AI Center, FKZ: 01IS18039B, and by the Machine Learning Cluster of Excellence, EXC number 2064/1, project number 390727645.

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

CONTENTS

# A    IMPLIED MODELS

In this section, we describe how to express the assignments of the observed variables of standardized SCMs and iSCMs with a general additive noise mechanism

$$f_i(\mathbf{x}, \varepsilon_i) = f_i(\mathbf{x}) + \varepsilon_i, \tag{6}$$

in the form of (SCM), while sharing the same causal graph $\mathcal{G}$ and exogenous noise variables $\boldsymbol{\varepsilon}$. We obtain the SCM form by moving the standardization steps into the causal mechanisms by linearly rescaling $f_i$ and $\varepsilon_i$, such that each observed variable is only a function of observed variables and the noise $\varepsilon_i$. Throughout this work, the ***implied (SCM) model*** denotes the specific construction given in the following two subsections. For this, we assume that we can express the first two moments of the system in closed form. Similar to the main text, we overload notation for both standardized SCMs and iSCMs and write

$$\mu_i := \mathbb{E}[x_i] \qquad \text{and} \qquad s_i := \sqrt{\mathrm{Var}[x_i]}\,.$$

We also derive analytic expressions for the weights of the implied models of linear iSCMs defined by Equation (1), which we later use in our proofs.

## A.1    IMPLIED MODEL OF A STANDARDIZED SCM

Let $\mathbf{x^s}$ be modeled by (Standardized SCM) with causal mechanisms defined by Equation (6). We recall that $\mathbf{x^s}$ are the observations obtained after standardizing $\mathbf{x}$. Thus, we can rearrange $x_i^s$ as

$$x_i = s_i x_i^s + \mu_i$$

and substitute every unstandardized variable $x_i$ by a function of its standardized parents $\mathbf{x}_{\mathrm{pa}(i)}^s$ as

$$x_i^s = \frac{x_i - \mu_i}{s_i} = \frac{f_i(\mathbf{x}_{\mathrm{pa}(i)}) + \varepsilon_i - \mu_i}{s_i} = \frac{f_i(\mathbf{x}_{\mathrm{pa}(i)}^s \odot \boldsymbol{s}_{\mathrm{pa}(i)} + \boldsymbol{\mu}_{\mathrm{pa}(i)}) - \mu_i}{s_i} + \frac{1}{s_i}\varepsilon_i\,,$$

where $\odot$ denotes elementwise multiplication, and $\boldsymbol{\mu}_{\mathrm{pa}(i)}$ and $\boldsymbol{s}_{\mathrm{pa}(i)}$ are the vectors of the parent means and standard deviations before standardization. Thus, the assignments of $\mathbf{x^s}$ in a standardized SCM can be written as the SCM given by

$$x_i^s = g_i^s(\mathbf{x}_{\mathrm{pa}(i)}^s) + \theta_i^s \varepsilon_i\,,$$

with implied noise scales $\theta_i^s := 1/s_i$ and implied causal mechanisms

$$g_i^s(\mathbf{x}_{\mathrm{pa}(i)}^s) := \begin{cases} \dfrac{f_i(\mathbf{x}_{\mathrm{pa}(i)}^s \odot \boldsymbol{s}_{\mathrm{pa}(i)} + \boldsymbol{\mu}_{\mathrm{pa}(i)}) - \mu_i}{s_i} & \text{if } i \text{ is a non-root variable, and} \\ \dfrac{f_i - \mu_i}{s_i} & \text{if } i \text{ is a root variable.} \end{cases}$$

## A.2    IMPLIED MODEL OF AN iSCM

Let $\widetilde{\mathbf{x}}$ be modeled by (iSCM) with causal mechanisms defined by Equation (6). In an iSCM, $\widetilde{\mathbf{x}}$ are the observed variables and $\mathbf{x}$ are the latent variables. We can express every observation $\widetilde{x}_i$ in terms of its observed parents $\widetilde{\mathbf{x}}_{\mathrm{pa}(i)}$ as

$$\widetilde{x}_i = \frac{x_i - \mu_i}{s_i} = \frac{f_i(\widetilde{\mathbf{x}}_{\mathrm{pa}(i)}) + \varepsilon_i - \mu_i}{s_i} = \frac{f_i(\widetilde{\mathbf{x}}_{\mathrm{pa}(i)}) - \mu_i}{s_i} + \frac{1}{s_i}\varepsilon_i\,.$$

Thus, the assignments of $\widetilde{\mathbf{x}}$ in a iSCM can be written as the SCM given by

$$\widetilde{x}_i = \widetilde{g}_i(\widetilde{\mathbf{x}}_{\mathrm{pa}(i)}) + \widetilde{\theta}_i \varepsilon_i\,,$$

with implied noise scales $\widetilde{\theta}_i := 1/s_i$ and implied causal mechanisms

$$\widetilde{g}_i(\widetilde{\mathbf{x}}_{\mathrm{pa}(i)}) := \begin{cases} \dfrac{f_i(\widetilde{\mathbf{x}}_{\mathrm{pa}(i)}) - \mu_i}{s_i} & \text{if } i \text{ is a non-root variable, and} \\ \dfrac{f_i - \mu_i}{s_i} & \text{if } i \text{ is a root variable.} \end{cases}$$

### A.3 WEIGHTS OF THE IMPLIED MODEL OF A LINEAR iSCM

Here, we derive the analytical form for the mechanisms of the implied model of a linear iSCM with zero-centered, additive noise $\varepsilon_i$. This iSCM is given by

$$x_i := \mathbf{w}_i^T \widetilde{\mathbf{x}}_{\mathrm{pa}(i)} + \varepsilon_i \qquad \text{and} \qquad \widetilde{x}_i := \frac{x_i}{\sqrt{\mathrm{Var}[x_i]}} \, ,$$

where $\varepsilon_i$ satisfies $\mathbb{E}[\varepsilon_i] = 0$ and $\mathrm{Var}[\varepsilon_i] = \sigma_i^2$. We can write the above as

$$\widetilde{x}_i = \frac{\mathbf{w}_i^T \widetilde{\mathbf{x}}_{\mathrm{pa}(i)} + \varepsilon_i}{\sqrt{\mathrm{Var}[x_i]}} = \frac{\sum_{j \in \mathrm{pa}(i)} w_{j,i} \widetilde{x}_j + \varepsilon_i}{\sqrt{\mathrm{Var}[x_i]}} = \sum_{j \in \mathrm{pa}(i)} \frac{w_{j,i}}{\sqrt{\mathrm{Var}[x_i]}} \widetilde{x}_j + \frac{1}{\sqrt{\mathrm{Var}[x_i]}} \varepsilon_i \, .$$

It follows that the implied SCM of a linear iSCM is also linear, with weights and noise variances given by

$$\widetilde{w}_{j,i} = \frac{w_{j,i}}{\sqrt{\mathrm{Var}[x_i]}} \qquad \text{and} \qquad \widetilde{\sigma}_i^2 = \frac{\sigma_i^2}{\mathrm{Var}[x_i]} \, . \tag{7}$$

In the above, we can write the variance of $x_i$ explicitly as

$$\mathrm{Var}[x_i] = \mathrm{Var}\left[ \sum_{j \in \mathrm{pa}(i)} w_{j,i} \widetilde{x}_j + \varepsilon_i \right] = \mathrm{Var}\left[ \sum_{j \in \mathrm{pa}(i)} w_{j,i} \widetilde{x}_j \right] + \sigma_i^2$$

$$\overset{\text{\textcircled{1}}}{=} \sum_{k \in \mathrm{pa}(i)} \sum_{j \in \mathrm{pa}(i)} \mathrm{Cov}[w_{k,i} \widetilde{x}_k, w_{j,i} \widetilde{x}_j] + \sigma_i^2 \tag{8}$$

$$\overset{\text{\textcircled{2}}}{=} \sum_{k \in \mathrm{pa}(i)} \sum_{j \in \mathrm{pa}(i)} w_{k,i} w_{j,i} \mathrm{Cov}[\widetilde{x}_k, \widetilde{x}_j] + \sigma_i^2 \, ,$$

where ① follows from Bienaymé's identity and ② from covariance being bilinear. Substituting the variance into the expressions for the weights and noise variances, we obtain

$$\widetilde{w}_{j,i} = \frac{w_{j,i}}{\sqrt{\sum_{k \in \mathrm{pa}(i)} \sum_{j \in \mathrm{pa}(i)} w_{k,i} w_{j,i} \mathrm{Cov}[\widetilde{x}_k, \widetilde{x}_j] + \sigma_i^2}} \, , \tag{9}$$

$$\widetilde{\sigma}_i^2 = \frac{\sigma_i^2}{\sum_{k \in \mathrm{pa}(i)} \sum_{j \in \mathrm{pa}(i)} w_{k,i} w_{j,i} \mathrm{Cov}[\widetilde{x}_k, \widetilde{x}_j] + \sigma_i^2} \, . \tag{10}$$

Finally, by construction, the variables $\widetilde{\mathbf{x}}$ of an iSCM have unit marginal variances. Thus, when the parents of $\widetilde{x}_i$ are pairwise independent, Equation (10) simplifies to

$$\widetilde{w}_{j,i} = \frac{w_{j,i}}{\sqrt{\sum_{j \in \mathrm{pa}(i)} w_{j,i}^2 + \sigma_i^2}} \, . \tag{11}$$

This independence condition always holds when the DAG $\mathcal{G}$ is a forest.

**Efficient computation** We can efficiently compute the implied model weights using a bottom-up dynamic programming approach. This allows sampling data directly from the exact implied model of an iSCM without resorting to empirical standardization statistics. Algorithm 2 describes the procedure. We iteratively compute the weights and noise variances of the implied model following Equations (9) and (10). At each iteration, we update the covariance matrix according to Lemma 1. The algorithm processes the nodes in topological order, mirroring the proof by induction of Lemma 1.

---

**Algorithm 2** Computing the Implied Model Parameters of Linear iSCMs

---

**Input:** DAG $\mathcal{G}$, weight matrix $[W]_{i,j} := w_{i,j}$, noise variances $\boldsymbol{\sigma}^2 \in \mathbb{R}_+^d$
$\widetilde{W} \leftarrow 0_{d \times d}$
$\Sigma \leftarrow I_d$
$\pi \leftarrow$ topological ordering of $\mathcal{G}$
**for** $i = 1$ to $d$ **do**
$\quad \mathbf{w} \leftarrow W_{:,\pi_i}$ $\qquad\qquad\qquad\qquad\qquad\qquad\qquad$ ▷ Edge weights ingoing to $\pi_i$
$\quad \mathrm{Var}[x_{\pi_i}] \leftarrow \mathbf{w}^\top \Sigma \mathbf{w} + \sigma_{\pi_i}^2$ $\qquad\qquad\qquad\qquad\qquad$ ▷ Equation (8)
$\quad \widetilde{W}_{:,\pi_i} \leftarrow \mathbf{w}/\sqrt{\mathrm{Var}[x_{\pi_i}]}$ $\qquad\qquad\qquad\qquad\qquad$ ▷ Equation (9)
$\quad \widetilde{\sigma}_{\pi_i}^2 \leftarrow \sigma_{\pi_i}^2 / \mathrm{Var}[x_{\pi_i}]$ $\qquad\qquad\qquad\qquad\qquad$ ▷ Equation (10)
$\quad$ **for** $j = 1$ to $i$ **do**
$\quad\quad \Sigma_{\pi_j,\pi_i} \leftarrow (\Sigma_{\pi_j,:})^\top \widetilde{W}_{:,\pi_i}$
$\quad\quad \Sigma_{\pi_i,\pi_j} \leftarrow \Sigma_{\pi_j,\pi_i}$
**return** implied weights $\widetilde{W}$, implied noise variances $\widetilde{\boldsymbol{\sigma}}^2$

---

## B   INTERVENTIONS IN iSCMs

For an iSCM $(\mathbf{S}, \mathcal{P}_{\boldsymbol{\varepsilon}})$, we can formalize interventions as changes to its causal mechanisms $f_i$, analogous to the common definition for SCMs (Peters et al., 2017). Specifically, let $\mu_i := \mathbb{E}[x_i]$ and $s_i := \sqrt{\mathrm{Var}[x_i]}$ be the mean and standard deviation of the latent variable $x_i$. We define an *intervention* as replacing one (or several) of the assignments to the latent variables as

$$x_i := h_i(\widetilde{\mathbf{x}}_{\mathrm{pa}(i)}, \varepsilon_i),$$

for some function $h_i$. Importantly, the statistics $\mu_i$ and $s_i$ used for the standardization operation

$$\widetilde{x}_i := \frac{x_i - \mu_i}{s_i}$$

remain *unchanged*. Thus, if we intervene on mechanisms of iSCMs, the variables $\widetilde{\mathbf{x}}$ may no longer have zero mean and unit variance, and the perturbations of $x_i$ propagate downstream through the causal mechanisms. We note that, under the above definition, intervening on an iSCM through a new mechanism $h_i$ is equivalent to intervening on the implied SCM of an iSCM with the mechanism

$$\widetilde{h}_i(\mathbf{x}, \varepsilon) = \frac{h_i(\mathbf{x}, \varepsilon) - \mu_i}{s_i}.$$

Appendix A.2 provides details on the implied models of iSCMs.

## C   PROOFS

### C.1   DEFINITIONS

We define the key concepts used throughout our analysis. A *path* $p_{j \leftrightarrow i}$ between $v_i$ and $v_j$ is a set of directed edges that allows reaching $v_i$ from $v_j$ (and vice versa), not taking into account edge directionality, and that joins unique vertices. We call a node a *collider* in a path if the node has two ingoing directed edges in the path. We say that a path between $v_i$ and $v_j$ is *unblocked* if and only if there is no node $v_k$ that is a collider in the path (see Figure 10a). Finally, we use the term *undirected connected component* to refer to any maximal subgraph of $\widetilde{\mathcal{G}}$ in which any two nodes are connected by a path containing only undirected edges (Wienöbst et al., 2023).

### C.2   EXPLICIT COVARIANCE IN LINEAR SCMs WITH UNIT MARGINAL VARIANCES

**Lemma 1** (Covariance in linear SCMs with unit marginal variances)**.** *Let $\mathbf{x}$ be modeled by a linear SCM defined by (1) with DAG $\mathcal{G}$ that satisfies $\mathrm{Var}[x_i] = 1$. Then, the covariance $\mathrm{Cov}[x_i, x_j]$ is the sum of products of the weights along all unblocked paths between the nodes of $x_i$ and $x_j$ in $\mathcal{G}$.*

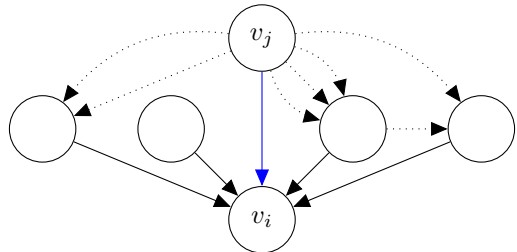

Figure 7: **Lemma 1 inductive step.** If $v_j$ is before $v_i$ in the topological ordering, then all unblocked paths from $v_j$ to $v_i$ must contain a parent of $v_i$ as the second to last node. To see this, suppose an unblocked path from $v_j$ to $v_i$ would instead contain a child of $v_i$ as the last node. Then, there either exists a collider on the path to $v_j$, contradicting that the path is unblocked, or all edges in the path point away from $v_i$, implying that $v_j$ is a descendant of $v_i$ and contradicting the topological ordering. Dotted lines represent unblocked paths (which may have common nodes). Solid lines represent edges. $v_j$ may or may not be a parent of $v_i$, which we illustrate with a blue arrow.

*Specifically, for any $i, j \in \{1, ..., d\}$ such that $i \neq j$, it holds that*

$$\text{Cov}[x_i, x_j] = \sum_{p_{j \leftrightarrow i} \in P_{j \leftrightarrow i}} \prod_{(l,m) \in p_{j \leftrightarrow i}} w_{l,m} \,, \tag{3}$$

*where $P_{j \leftrightarrow i}$ are all unblocked paths from $x_j$ to $x_i$ in $\mathcal{G}$, and $(l, m) \in p_{j \leftrightarrow i}$ indicates that the directed edge $(l, m)$ is part of the path $p_{j \leftrightarrow i}$.*

*Proof.* We will give a proof by induction on the number of vertices $d = |\mathcal{V}|$ in the DAG $\mathcal{G}$. Without loss of generality, we assume that the indices of the nodes are ordered according to some fixed topological ordering $\pi$, so $\pi(j) < \pi(i)$ if $j < i$. By the unit marginal variance assumption,

$$\text{Cov}[x_i, x_i] = \text{Var}[x_i] = 1 \,. \tag{12}$$

From now on and without loss of generality, we consider two arbitrary indices $j < i$. The covariance between $x_i$ and $x_j$ is symmetric.

**Base case ($d = 2$)** If $v_j$ is not an ancestor of $v_i$ in graph $\mathcal{G}$, they both must be root nodes, because the edge $v_i \leftarrow v_j$ is the only possible edge when $\pi(j) < \pi(i)$. Since $x_i$ and $x_j$ are root nodes, they are independent and $\text{Cov}[x_i, x_j] = 0$. Since a path of one edge cannot contain a collider, there are no unblocked paths between $v_i$ and $v_j$, so the RHS of Equation (3) is also 0.

Conversely, if $v_j$ is an ancestor of $v_i$ in graph $\mathcal{G}$, $v_j$ is the only parent and ancestor of $v_i$. This implies that

$$\begin{aligned}
\text{Cov}[x_i, x_j] &= \text{Cov}[w_{j,i} x_j + \varepsilon_i, x_j] \\
&= w_{j,i} \text{Cov}[x_j, x_j] \\
&= w_{j,i} \,,
\end{aligned}$$

where the last equality follows from Equation (12). This is exactly Equation (3) for a two-node graph.

**Induction step ($d > 2$)** Let us assume that Equation (3) holds for all graphs of size $d - 1$, and let $\mathcal{G}$ have $d$ nodes. We will apply the inductive hypothesis to the subgraph of the first $d - 1$ nodes in $\mathcal{G}$ and show that the full DAG $\mathcal{G}$ including the $d$-th vertex still satisfies Equation (3). First, we note that, since the $d$-th vertex is last in the topological ordering, it has no outgoing edges. Because the node has no outgoing edges, it is not visited on any unblocked paths between $v_j$ and $v_i$ for $i, j < d$, as $v_d$ must be a collider in any path. Second, adding the node $v_d$ to a subsystem containing $x_1, \ldots, x_{d-1}$ results in no change to the joint distribution of $x_i, x_j$. Therefore, it has no effect on the covariance between $x_i, x_j$. Hence, both sides of Equation (3) are unchanged by the presence of a node $v_d$ for all $i, j < d$ and the equation still holds for all $i, j < d$.

We want to show that Equation (3) also holds for $i = d$ and any $j < i$. For this, we first construct all unblocked paths from $v_j$ to $v_i$. First, we note that any unblocked path must go through the parents

$k \in \mathrm{pa}(i)$, because $j < i$ in the topological ordering (see Figure 7). Moreover, for any $k \in \mathrm{pa}(i)$, appending $k \to i$ to an unblocked path $p_{j\leftrightarrow k}$ between $v_j$ and $v_k$, creates a new unblocked path between $v_j$ and $v_i$. Hence, for $i = d$ and any $j < i$, it holds that

$$
\begin{aligned}
\mathrm{Cov}[x_i, x_j] &= \mathrm{Cov}[\sum_{k \in \mathrm{pa}(i)} w_{k,i} x_k + \varepsilon_i, x_j] \\
&= \sum_{k \in \mathrm{pa}(i)} w_{k,i} \, \mathrm{Cov}[x_k, x_j] \\
&\overset{①}{=} w_{j,i} \, \mathrm{Cov}[x_j, x_j] + \sum_{k \in \mathrm{pa}(i)\backslash j} w_{k,i} \, \mathrm{Cov}[x_k, x_j] \\
&\overset{②}{=} w_{j,i} + \sum_{k \in \mathrm{pa}(i)\backslash j} w_{k,i} \sum_{p_{j\leftrightarrow k} \in P_{j\leftrightarrow k}} \prod_{(l,m) \in p_{j\leftrightarrow k}} w_{l,m} \\
&= w_{j,i} + \sum_{k \in \mathrm{pa}(i)\backslash j} \left( \sum_{p_{j\leftrightarrow k} \in P_{j\leftrightarrow k}} w_{k,i} \prod_{(l,m) \in p_{j\leftrightarrow k}} w_{l,m} \right) \\
&\overset{③}{=} \sum_{k \in \mathrm{pa}(i)} \left( \mathbb{1}[k=j] w_{j,i} + \mathbb{1}[k \neq j] \left( \sum_{p_{j\leftrightarrow k} \in P_{j\leftrightarrow k}} w_{k,i} \prod_{(l,m) \in p_{j\leftrightarrow k}} w_{l,m} \right) \right) \\
&\overset{④}{=} \sum_{p_{j\leftrightarrow i} \in P_{j\leftrightarrow i}} \prod_{(l,m) \in p_{j\leftrightarrow i}} w_{l,m} \,.
\end{aligned}
$$

For step ①, consider two cases. If $j \notin \mathrm{pa}(i)$, then $w_{j,i} = 0$ and the equality trivially holds. If $j \in \mathrm{pa}(i)$, then it holds by pulling the term for $j$ out of the sum in the previous line. In ②, we apply the inductive hypothesis to express the covariances in terms of a sum of products of weights. In ③, we rearrange terms to pull the $w_{j,i}$ term into the sum over parents. In ④, we use the fact that the set of unblocked paths from $v_j$ to $v_i$ corresponds to all paths from $v_j$ to any parent of $v_i$, which is $v_k$ here, with an extra edge $k \to i$ appended, and a possible single-edge path directly connecting $v_j$ with $v_i$ (if $j \in \mathrm{pa}(i)$).

This completes the induction step and the proof. □

### C.3 BOUND ON THE FRACTION OF CEV

**Theorem 2** (Bound on $\mathrm{CEV_f}$ in linear iSCMs). *Let $\mathbf{x}$ be modeled by a linear iSCM (1) with DAG $\mathcal{G}$ and additive noise of equal variances $\mathrm{Var}[\varepsilon_i] = \sigma^2$. Suppose any node in $\mathcal{G}$ has at most $m$ parents and $w = \max_{i,j \in \{1,...,d\}} |w_{i,j}|$. Then, for any $i \in \{1, ..., d\}$, the fraction of CEV for $\widetilde{x}_i$ is bounded as*

$$
\mathrm{CEV_f}[\widetilde{x}_i] \leq 1 - \frac{\sigma^2}{m^2 w^2 + \sigma^2} \,.
$$

*Proof.* We begin by bounding the variance of the latent variables $x_i$ in iSCMs. Starting from Equation (8), we can bound the covariances with a product of unit variances as

$$
\begin{aligned}
\mathrm{Var}[x_i] &= \sum_{k \in \mathrm{pa}(i)} \sum_{j \in \mathrm{pa}(i)} w_{k,i} w_{j,i} \, \mathrm{Cov}[\widetilde{x}_j, \widetilde{x}_k] + \sigma^2 \\
&\overset{①}{\leq} \sum_{k \in \mathrm{pa}(i)} \sum_{j \in \mathrm{pa}(i)} w_{k,i} w_{j,i} + \sigma^2 \\
&= \left( \sum_{j \in \mathrm{pa}(i)} w_{j,i} \right)^2 + \sigma^2 \\
&\overset{②}{\leq} m^2 w^2 + \sigma^2 \,,
\end{aligned}
$$

where ① uses $\text{Cov}[\widetilde{x}_j, \widetilde{x}_k] \leq 1$ since $\text{Var}[\widetilde{x}_j] = 1$ and $\text{Var}[\widetilde{x}_k] = 1$, and ② applies the Cauchy-Schwartz inequality. Since we obtain $\widetilde{x}_i$ from $x_i$ just by shifting and scaling the latter, we observe that $\text{CEV}_{\text{f}}[\widetilde{x}_i] = \text{CEV}_{\text{f}}[x_i]$. Using the upper bound on the variance of $x_i$ and the definition of the fraction of cause-explained variance in Equation (4)), we get

$$\text{CEV}_{\text{f}}[\widetilde{x}_i] = \text{CEV}_{\text{f}}[x_i] = 1 - \frac{\text{Var}[x_i - \mathbb{E}[x_i | \mathbf{x}_{\text{pa}(i)}]]}{\text{Var}[x_i]} = 1 - \frac{\text{Var}[x_i - \mathbf{w}_i^\top \mathbf{x}_{\text{pa}(i)}]}{\text{Var}[x_i]}$$

$$= 1 - \frac{\text{Var}[\varepsilon_i]}{\text{Var}[x_i]} = 1 - \frac{\sigma^2}{\text{Var}[x_i]} \leq 1 - \frac{\sigma^2}{m^2 w^2 + \sigma^2}.$$

$\square$

## C.4  IDENTIFIABILITY

In this section, we prove Theorems 3 and 4. We begin by deriving the covariances for the 3-node example in Section 4.2 and then give the general proofs for forests. The proofs of both theorems share the same underlying argument. We first derive the SCM forms of the original models, i.e., standardized SCMs in Theorem 3 and iSCMs in Theorem 4. By showing that the standardized SCMs and iSCMs are SCMs with the same causal graphs $\mathcal{G}$ and observational distributions $p(\mathbf{x})$, we can leverage Lemma 1 to obtain the covariances between the observed variables in both model classes. Ultimately, these covariances allow us to derive (non)identifiability conditions for the DAGs $\mathcal{G}$ in an MEC underlying the original models.

Theorems 3 and 4 assume that the exogenous noise is sampled from a zero-centered distribution with equal variance across variables. Since the results are based on the analysis of covariances, they also hold with the assumption that $\mathbb{E}[\varepsilon_i] \neq 0$, but the zero-mean assumption simplifies notation. To derive the results for iSCMs, we additionally assume that the noise is Gaussian (see Theorem 4) . When referring to an undirected edge between nodes $v_i, v_j$, for example, in an MEC, we still denote the edge with $(v_i, v_j)$, but the ordering of the nodes is arbitrary.

### C.4.1  3-NODE CASE

We begin by studying the 3-node example of Figure 3 in Section 4.2. Let $\alpha_i, \beta_i, \gamma_i, \lambda_i \in \mathbb{R}$ be linear function weights, and consider the following three causal graphs $\mathcal{G}$ belonging to the same MEC, along with their corresponding SCMs and iSCMs.

| $\mathcal{G}$ | SCM | iSCM |
|---|---|---|

$$x_1 := \varepsilon_1$$
$$x_2 := \alpha_1 x_1 + \varepsilon_2 \quad (13)$$
$$x_3 := \beta_1 x_2 + \varepsilon_3$$

$$x_1 := \varepsilon_1$$
$$x_2 := \gamma_1 \widetilde{x}_1 + \varepsilon_2 \quad (14)$$
$$x_3 := \lambda_1 \widetilde{x}_2 + \varepsilon_3$$

$$x_1 := \alpha_2 x_2 + \varepsilon_1$$
$$x_2 := \varepsilon_2 \quad (15)$$
$$x_3 := \beta_2 x_2 + \varepsilon_3$$

$$x_1 := \gamma_2 \widetilde{x}_1 + \varepsilon_1$$
$$x_2 := \varepsilon_2 \quad (16)$$
$$x_3 := \lambda_2 \widetilde{x}_2 + \varepsilon_3$$

$$x_1 := \alpha_3 x_2 + \varepsilon_1$$
$$x_2 := \beta_3 x_3 + \varepsilon_2 \quad (17)$$
$$x_3 := \varepsilon_3$$

$$x_1 := \gamma_3 \widetilde{x}_2 + \varepsilon_1$$
$$x_2 := \lambda_3 \widetilde{x}_3 + \varepsilon_2 \quad (18)$$
$$x_3 := \varepsilon_3$$

In the following subsections, we derive the covariance matrices of each of the three systems, respectively. This leads us to the equivalence presented in Equation (5) for standardized SCMs. Moreover, we show that, for iSCMs, all three systems induce exactly the same observational distribution if and only if $\lambda_1 = \lambda_2 = \lambda_3$ and $\gamma_1 = \gamma_2 = \gamma_3$. These are the 3-node special cases of Theorems 3 and 4.

STANDARDIZED SCM

To obtain the covariances between the observed variables in the standardized SCMs of Equations (13), (15), and (17), we first show that the assignments to the observed variables in standardized SCMs can be written in the form of linear SCMs over the same causal graph, which allows us to use Lemma 1. In all three systems, every vertex has at most one parent. When the node $v_j$ is the only parent of $v_i$, under our assumptions on the noise, we have $x_j = \sqrt{\mathrm{Var}[x_j]}x_j^s$, so the assignment of $x_i^s$ can be written in the form of an SCM over $\mathbf{x^s}$ as

$$x_i^s := \frac{x_i}{\sqrt{\mathrm{Var}[x_i]}} = \frac{w_{j,i}x_j + \varepsilon_i}{\sqrt{\mathrm{Var}[x_i]}} = \frac{w_{j,i}\sqrt{\mathrm{Var}[x_j]}x_j^s + \varepsilon_i}{\sqrt{\mathrm{Var}[x_i]}} = w_{j,i}\sqrt{\frac{\mathrm{Var}[x_j]}{\mathrm{Var}[x_i]}}x_j^s + \frac{\varepsilon_i}{\sqrt{\mathrm{Var}[x_i]}} \ . \quad (19)$$

To use Equation (19), we first need to compute the marginal variances of the unstandardized observations $x_i$. For the standardized SCMs, these marginal variances are, respectively:

| for Equation (13): | for Equation (15): | for Equation (17): |
|---|---|---|
| $\mathrm{Var}[x_1] = \sigma^2$ | $\mathrm{Var}[x_1] = (\alpha_2^2 + 1)\sigma^2$ | $\mathrm{Var}[x_1] = (\alpha_3^2(\beta_3^2 + 1) + 1)\sigma^2$ |
| $\mathrm{Var}[x_2] = (\alpha_1^2 + 1)\sigma^2$ | $\mathrm{Var}[x_2] = \sigma^2$ | $\mathrm{Var}[x_2] = (\beta_3^2 + 1)\sigma^2$ |
| $\mathrm{Var}[x_3] = (\beta_1^2(\alpha_1^2 + 1) + 1)\sigma^2$ | $\mathrm{Var}[x_3] = (\beta_2^2 + 1)\sigma^2$ | $\mathrm{Var}[x_3] = \sigma^2$ |

Given Equation (19) and the marginal variances, we know the weights of all three implied SCMs explicitly. Since all implied SCMs are linear, have unit marginal variances, and share the same causal graph, we can apply Lemma 1 and obtain the covariances of the observational distributions in the original models:

| for Equation (13): | for Equation (15): | for Equation (17): |
|---|---|---|
| $\mathrm{Cov}[x_1^s, x_2^s] = \frac{\alpha_1}{\sqrt{\alpha_1^2 + 1}}$ | $\mathrm{Cov}[x_1^s, x_2^s] = \frac{\alpha_2}{\sqrt{\alpha_2^2 + 1}}$ | $\mathrm{Cov}[x_1^s, x_2^s] = \alpha_3\sqrt{\frac{\beta_3^2 + 1}{\alpha_3^2(\beta_3^2 + 1) + 1}}$ |
| $\mathrm{Cov}[x_1^s, x_3^s] = \frac{\alpha_1\beta_1}{\sqrt{\beta_1^2(\alpha_1^2 + 1) + 1}}$ | $\mathrm{Cov}[x_1^s, x_3^s] = \frac{\alpha_2\beta_2}{\sqrt{(\alpha_2^2 + 1)(\beta_2^2 + 1)}}$ | $\mathrm{Cov}[x_1^s, x_3^s] = \frac{\alpha_3}{\sqrt{\alpha_3^2(\beta_3^2 + 1) + 1}}$ |
| $\mathrm{Cov}[x_2^s, x_3^s] = \beta_1\sqrt{\frac{\alpha_1^2 + 1}{\beta_1^2(\alpha_1^2 + 1) + 1}}$ | $\mathrm{Cov}[x_2^s, x_3^s] = \frac{\beta_2}{\sqrt{\beta_2^2 + 1}}$ | $\mathrm{Cov}[x_2^s, x_3^s] = \frac{\beta_3}{\sqrt{\beta_3^2 + 1}}$ |

In the standardized SCM (13), the causal graph is $v_1 \rightarrow v_2 \rightarrow v_3$. Hence, the edge directions of the DAG $\mathcal{G}$ are consistent with the direction of increasing absolute covariance if and only if

$$
\begin{aligned}
|\mathrm{Cov}[x_1^s, x_2^s]| < |\mathrm{Cov}[x_2^s, x_3^s]| &\iff \left|\frac{\alpha_1}{\sqrt{\alpha_1^2 + 1}}\right| < \left|\beta_1\sqrt{\frac{\alpha_1^2 + 1}{\beta_1^2(\alpha_1^2 + 1) + 1}}\right| \\
&\iff \frac{\alpha_1^2}{\alpha_1^2 + 1} < \beta_1^2\frac{\alpha_1^2 + 1}{\beta_1^2(\alpha_1^2 + 1) + 1} \\
&\iff \alpha_1^2(\beta_1^2(\alpha_1^2 + 1) + 1) < \beta_1^2(\alpha_1^2 + 1)^2 \qquad (20) \\
&\iff \beta_1^2\alpha_1^4 + \beta_1^2\alpha_1^2 + \alpha_1^2 < \beta_1^2\alpha_1^4 + 2\beta_1^2\alpha_1^2 + \beta_1^2 \\
&\iff \alpha_1^2 < \beta_1^2(\alpha_1^2 + 1) \\
&\iff \frac{\alpha_1^2}{\alpha_1^2 + 1} < \beta_1^2 \ .
\end{aligned}
$$

In the above equivalences, we always multiply or divide by quantities greater than $0$, so the direction of the inequality does not change, and transformations are equivalent. For the standardized SCM (17) with causal graph $v_1 \leftarrow v_2 \leftarrow v_3$, we get an analogous condition for the edges to be aligned with the order of increasing absolute covariance when following the same algebraic manipulations:

$$|\mathrm{Cov}[x_3^s, x_2^s]| < |\mathrm{Cov}[x_2^s, x_1^s]| \iff \frac{\beta_3^2}{\beta_3^2 + 1} < \alpha_3^2.$$

We make use of both of these conditions in Section 4. Since $z/(z+1) < 1$ for any $z > 0$, the right-hand sides of both conditions are true if all weights are greater than 1. In this case, the absolute covariance increases downstream in all SCMs of Equations (13) and (17). Hence, among these two systems, only the DAG $\mathcal{G}$ whose edges aligns with the covariance ordering in the observed $p(\mathbf{x^s})$ can induce $p(\mathbf{x^s})$, and we can conclude that the other DAG is not the true causal graph.

iSCM

To derive the observational distributions of the iSCMs in Equations (14), (16), and (18), we proceed in the same way as we did for standardized SCMs. We first show that the iSCM is an SCM with a specific set of mechanisms and then apply Lemma 1 to obtain the covariances between the observed variables. To see this, we write the assignment of $\widetilde{x}_i$ as

$$\widetilde{x}_i := \frac{x_i}{\sqrt{\mathrm{Var}[x_i]}} = \frac{w_{j,i}\widetilde{x}_j + \varepsilon_i}{\sqrt{\mathrm{Var}[x_i]}} = \frac{w_{j,i}}{\sqrt{\mathrm{Var}[x_i]}}\widetilde{x}_j + \frac{\varepsilon_i}{\sqrt{\mathrm{Var}[x_i]}} \tag{21}$$

As before, using Equation (21) requires first computing the marginal variances of the latent variables $x_i$. For the iSCMs defined by Equations (14), (16), and (18), they are given by

| for Equation (14): | for Equation (16): | for Equation (18): |
|---|---|---|
| $\mathrm{Var}[x_1] = \sigma^2$ | $\mathrm{Var}[x_1] = \gamma_2^2 + \sigma^2$ | $\mathrm{Var}[x_1] = \gamma_3^2 + \sigma^2$ |
| $\mathrm{Var}[x_2] = \gamma_1^2 + \sigma^2$ | $\mathrm{Var}[x_2] = \sigma^2$ | $\mathrm{Var}[x_2] = \lambda_3^2 + \sigma^2$ |
| $\mathrm{Var}[x_3] = \lambda_1^2 + \sigma^2$ | $\mathrm{Var}[x_3] = \lambda_2^2 + \sigma^2$ | $\mathrm{Var}[x_3] = \sigma^2$ |

Given Equation (21) and the marginal variances, we obtain an explicit form for the weights of all three implied SCMs. Since the implied SCMs are linear, have unit marginal variances, and share the same causal graph, we can apply Lemma 1 and obtain the covariances of the observational distributions in the original models. It turns out that the observational distribution of all three ground-truth systems $(\widetilde{x}_1, \widetilde{x}_2, \widetilde{x}_3)$ in Equations (14), (16), and (18) is a multivariate Gaussian with *the same covariance matrix*, with the diagonal elements equal to 1 and the off-diagonal elements given by

$$\begin{aligned}
\mathrm{Cov}[\widetilde{x}_1, \widetilde{x}_2] &= \frac{\gamma_i}{\sqrt{\gamma_i^2 + \sigma^2}} \\
\mathrm{Cov}[\widetilde{x}_1, \widetilde{x}_3] &= \frac{\gamma_i \lambda_i}{\sqrt{(\lambda_i^2 + \sigma^2)(\gamma_i^2 + \sigma^2)}} \\
\mathrm{Cov}[\widetilde{x}_2, \widetilde{x}_3] &= \frac{\lambda_i}{\sqrt{\lambda_i^2 + \sigma^2}}
\end{aligned} \tag{22}$$

Since the observational distribution of all three SCMs is a zero-centered multivariate Gaussian, the distributions are equal if and only if their their covariance matrices are identical. The covariances are equal if and only if $\lambda_1 = \lambda_2 = \lambda_3$ and $\gamma_1 = \gamma_2 = \gamma_3$, because the function $f(z) = z/\sqrt{z^2 + \sigma^2}$ appearing in $\mathrm{Cov}[\widetilde{x}_1, \widetilde{x}_2]$ and $\mathrm{Cov}[\widetilde{x}_2, \widetilde{x}_3]$ of Equation (22) is *injective* for any $\sigma > 0$, which means that distinct weights $z$ are mapped to distinct covariances. Therefore, the three node linear iSCMs in the above MEC share the same observational distribution if and only if they also share the same weights for each edge, regardless of edge orientation.

This implies that the three DAGs $\mathcal{G}$ in the MEC of Equations (14), (16), and (18) are not identifiable from $p(\widetilde{\mathbf{x}})$: given $p(\widetilde{\mathbf{x}})$ induced by an iSCM with DAG in this 3-node MEC, the two other DAGs with the same linear function weights induce the same distribution $p(\widetilde{\mathbf{x}})$.

### C.4.2 FORESTS

In this section, we generalize the above partial identifiability result for standardized SCMs to arbitrary forest DAGs (Theorem 3). After that, we similarly generalize the nonidentifiability of iSCMs to forests (Theorem 4). Our results concern the identification edge directions in an MEC represented by its partially directed graph $\tilde{\mathcal{G}} = (\mathcal{V}, \tilde{\mathcal{E}})$, where $\tilde{\mathcal{E}}$ contains both directed and undirected edges.

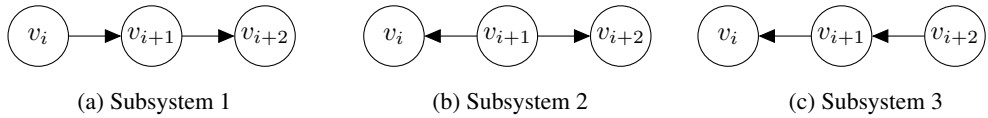

(a) Subsystem 1         (b) Subsystem 2         (c) Subsystem 3

Figure 8: **Proof subcases of Lemma 5.** Three possible subgraphs in a chain without a collider.

STANDARDIZED SCM

Before proving the main theorem, we extend the 3-node example to chains of arbitrary length. We show that all but at most one edge in the MEC can be correctly oriented from observational data using the assumption on the support of the weights. Analogous to the three node case, we then use this to prove a similar result for forest graphs.

**Lemma 5** (Orientation of edges in undirected chains of standardized SCMs). *Let $\mathbf{x^s}$ be modeled by a standardized linear SCM (1) with chain DAG $\mathcal{G} = (\mathcal{V}, \mathcal{E})$, where $\mathrm{Var}[\varepsilon_i] = \sigma^2$ for non-root nodes and $|w_{i,j}| > 1$ for all $i \in pa(j)$. Additionally, suppose $\mathcal{G}$ contains no colliders. Then, given $p(\mathbf{x^s})$ and the partially directed graph $\tilde{\mathcal{G}}$ representing the MEC of $\mathcal{G}$, we can identify all but at most one edge $(v_i, v_j)$ of the true DAG $\mathcal{G}$ in each undirected connected component of the MEC $\tilde{\mathcal{G}}$. The possible undirected edge has the smallest absolute covariance of all variables connected by edges in the MEC, satisfying $|\mathrm{Cov}[x_i^s, x_j^s]| < |\mathrm{Cov}[x_k^s, x_l^s]|$ for all $(k, l) \in \tilde{\mathcal{E}} \setminus (i, j)$.*

*Proof.* Throughout the proof, we label the nodes $v_i \in \mathcal{V}$ such that $v_{i-1}$ and $v_{i+1}$ are its neighbors for $i \in \{2, \ldots, d-1\}$. We start with the analysis of three arbitrary, consecutive vertices in a chain graph. The three possible subgraphs are depicted in Figure 8. We can always find $p \in \mathbb{R}$ such that the variance of the latent root of this directed subgraph is $p^2 \sigma^2$. This relaxed assumption on specifically the root node allows for the root of the subgraph to have potential parents outside the subgraph, or to be the root of the whole chain, when later using this lemma to prove the main theorem.

We will follow similar derivations as in Section C.4.1. Specifically, we first write the observed variables of the standardized SCM in SCM form, and then invoke Lemma 1 to obtain the covariances of the observed variables. To use Equation (19), we again need to compute the marginal variances of the variables before standardization. For the subsystems in Figures 8a and 8b, these are, respectively:

for Figure 8a:                        for Figure 8b:

$$\mathrm{Var}[x_i] = p^2 \sigma^2 \qquad\qquad\qquad \mathrm{Var}[x_i] = (w_{i+1,i}^2 p^2 + 1)\sigma^2$$

$$\mathrm{Var}[x_{i+1}] = (w_{i,i+1}^2 p^2 + 1)\sigma^2 \qquad\qquad \mathrm{Var}[x_{i+1}] = p^2 \sigma^2$$

$$\mathrm{Var}[x_{i+2}] = (w_{i+1,i+2}^2(w_{i,i+1}^2 p^2 + 1) + 1)\sigma^2 \quad \mathrm{Var}[x_{i+2}] = (w_{i+1,i+2}^2 p^2 + 1)\sigma^2$$

By substituting the expressions for the marginal variances into Equation (19), we obtain the weights of the implied models of the standardized SCM. Using Lemma 1, we obtain the covariances between the observed variables $x_{i-1}^s, x_i^s, x_{i+1}^s$. By construction, the marginal variances of the observed variables are equal to 1. We treat each subsystem separately:

**Subsystem 1** (Figure 8a)    Given the marginal variances and Lemma 1, the covariances are

$$\mathrm{Cov}[x_i^s, x_{i+1}^s] = \frac{w_{i,i+1} p}{\sqrt{w_{i,i+1}^2 p^2 + 1}}$$

$$\mathrm{Cov}[x_{i+1}^s, x_{i+2}^s] = w_{i+1,i+2} \sqrt{\frac{w_{i,i+1}^2 p^2 + 1}{w_{i+1,i+2}^2(w_{i,i+1}^2 p^2 + 1) + 1}}$$

Following the same algebraic manipulations as in Equation (20), substituting $\alpha_1 := w_{i,i+1}p$ and $\beta_1 := w_{i+1,i+2}$ in the derivation, we obtain

$$\left| \text{Cov}[x_i^s, x_{i+1}^s] \right| < \left| \text{Cov}[x_{i+1}^s, x_{i+2}^s] \right| \quad \Longleftrightarrow \quad \frac{w_{i,i+1}^2 p^2}{w_{i,i+1}^2 p^2 + 1} < w_{i+1,i+2}^2 . \tag{23}$$

The left-hand side of the right-hand inequality in Equation (23) is upper-bounded by 1, similar to the 3-node case. Therefore, if we assume that $|w_{i+1,i+2}| \geq 1$, it must hold that $|\text{Cov}[x_i^s, x_{i+1}^s]| < |\text{Cov}[x_{i+1}^s, x_{i+2}^s]|$ for any choice of $p$.

**Subsystem 2** (Figure 8b)  Given the marginal variances and Lemma 1, the covariances are

$$\text{Cov}[x_i^s, x_{i+1}^s] = \frac{w_{i+1,i}p}{\sqrt{w_{i+1,i}^2 p^2 + 1}}$$

$$\text{Cov}[x_{i+1}^s, x_{i+2}^s] = \frac{w_{i+1,i+2}p}{\sqrt{w_{i+1,i+2}^2 p^2 + 1}} .$$

The ordering of the covariances in this case depends on the specific choice of the weights.

**Subsystem 3** (Figure 8c)  Following steps analogous to the symmetric subsystem 1, we conclude that, if $|w_{i+1,i}| \geq 1$, it must hold that $|\text{Cov}[x_i^s, x_{i+1}^s]| > |\text{Cov}[x_{i+1}^s, x_{i+2}^s]|$ for any $p$.

Given the above, we can now study the relationship between the underlying DAG $\mathcal{G}$ and the absolute covariance magnitudes under the assumption that $|w_{i,i+1}| > 1$. We will use the fact that, if the chain does not contain a collider, then there can be at most one node contained in edges pointing in opposite directions.

First, we treat the case where there exists a vertex $v_i$ such that $|\text{Cov}[x_{i-1}^s, x_i^s]| = |\text{Cov}[x_i^s, x_{i+1}^s]|$, that is, where some neighboring covariances are equal. If this occurs in a 3-node subsystem, only subsystem 2 can describe the true graph. To be consistent with the assumption that there are no colliders in the graph (see Lemma 5), all other edges must be oriented in a direction away from $v_i$, which completely identifies the graph $\mathcal{G}$ in the MEC.

In the second case, $|\text{Cov}[x_{j-1}^s, x_j^s]| \neq |\text{Cov}[x_j^s, x_{j+1}^s]|$ holds for all nodes $v_j$ that have two neighbors in the path. Let $x_i^s, x_{i+1}^s$ be the unique pair of consecutive variables in the chain that minimizes $|\text{Cov}[x_i^s, x_{i+1}^s]|$. We can show that this pair is the unique minimizer using a proof by contradiction. Suppose there exist two pairs $x_i^s, x_{i+1}^s$ and $x_j^s, x_{j+1}^s$ such that $|\text{Cov}[x_i^s, x_{i+1}^s]| = |\text{Cov}[x_j^s, x_{j+1}^s]|$ is the minimum covariance. Without loss of generality, let $j + 1 < i$. Then, the triple $x_{i-1}^s, x_i^s, x_{i+1}^s$ is consistent with only subsystems 2 or 3 based on their relative covariances, which implies that we must have $v_{i-1} \leftarrow v_i$. Using the fact that we have no colliders, we can then orient all edges $v_{k-1} \leftarrow v_k$ for $1 < k < i$. Thus, we can find a subsystem containing $v_j, v_{j+1}, v_{j+2}$, which has been already oriented as subsystem 3, meaning $|\text{Cov}[x_j^s, x_{j+1}^s]| > |\text{Cov}[x_{j+1}^s, x_{j+2}^s]|$, a contradiction.

Given $x_i^s, x_{i+1}^s$ is the unique pair of consecutive variables that minimizes $|\text{Cov}[x_i^s, x_{i+1}^s]|$, we now show that we can orient all edges except $(v_i, v_{i+1})$. We will do this in two parts. First, we show that one can orient all edges $(v_j, v_{j+1})$ with $j < i$, and then we show that we can do the same for all edges $(v_j, v_{j+1})$ with $j > i$. If $i > 1$, consider the subsystem $v_{i-1}, v_i, v_{i+1}$. Since $|\text{Cov}[x_{i-1}^s, x_i^s]| > |\text{Cov}[x_i^s, x_{i+1}^s]|$, only subsystems 2 and 3 are possible for this subgraph. We can therefore orient $v_{i-1} \leftarrow v_i$. Similarly, if $i < d - 1$, by a symmetric argument on $v_i, v_{i+1}, v_{i+2}$, we can orient $v_{i+1} \rightarrow v_{i+2}$. Since the graph cannot contain colliders, all other edges must be oriented as $v_j \leftarrow v_{j+1}$ for $j < i$, and $v_j \rightarrow v_{j+1}$ for $j > i$. In other words, all edges except $(v_i, v_{i+1})$ point away from the two vertices $v_i, v_{i+1}$, and one of the two variables must be the root of the chain. Therefore, if $|\text{Cov}[x_{j-1}^s, x_j^s]| \neq |\text{Cov}[x_j^s, x_{j+1}^s]|$ holds for all vertices $v_j$ that have two neighbors, then there exists a unique covariance minimizing pair $x_i^s, x_{i+1}^s$, and all edges except $(v_i, v_{i+1})$ are oriented.

The two cases above are exhaustive, and in the worst case at most one edge $(v_j, v_{j+1})$ is left unoriented in the chain. This edge always corresponds to the minimizer of $|\text{Cov}[x_j^s, x_{j+1}^s]|$. This completes the proof.  $\square$

**Remark**   From the proof of Lemma 5, it follows that if we are able to orient all the edges in the chain, then the root of the chain is the node joining the two edges with minimum absolute covariance. When we orient all but one edge $(v_i, v_{i+1})$, the root node of the chain is either $v_i$ or $v_j$.

We can extend Lemma 5 to forest graphs. For this, we will make use of the first Meek rule (Meek, 1995). The first Meek rule concerns an MEC $\tilde{\mathcal{G}}$, containing the undirected edges $(v_i, v_j), (v_j, v_k)$ but not the edge $(v_i, v_k)$. It states that, if one can orient $v_i \to v_j$, we must have $v_j \to v_k$.

**Theorem 3** (Partial identifiability of standardized linear SCMs with forest DAGs). *Let $\mathbf{x^s}$ be modeled by a standardized linear SCM* (1) *with forest DAG $\mathcal{G}$, additive noise of equal variances $\mathrm{Var}[\varepsilon_i] = \sigma^2$, and $|w_{i,j}| > 1$ for all $i \in pa(j)$. Then, given $p(\mathbf{x^s})$ and the partially directed graph $\tilde{\mathcal{G}}$ representing the MEC of $\mathcal{G}$, we can identify all but at most one edge of the true DAG $\mathcal{G}$ in each undirected connected component of the MEC $\tilde{\mathcal{G}}$.*

*Proof.* The undirected parts of an MEC $\tilde{\mathcal{G}}$ are disjoint undirected connected components. Orienting the edges in all these undirected connected components without introducing a v-structure produces a valid DAG $\mathcal{G}$ in $\tilde{\mathcal{G}}$ (Andersson et al., 1997). Each undirected connected components represents a Markov equivalence class of its own (Andersson et al., 1997). Thus, to prove the theorem, we consider these undirected connected components independently with respect to the rest of the graph and show how to orient the edges in each undirected connected component.[1] In the following argument, we therefore consider $\tilde{\mathcal{G}}$ to be a single undirected connected component, with no directed edges by definition, and show that we can orient all but one edge in $\tilde{\mathcal{G}}$. This argument then extends to all undirected connected components of the original MEC $\tilde{\mathcal{G}}$, implying the statement made in Theorem 3.

If $\tilde{\mathcal{G}}$ is an undirected connected component with no directed edges, we only have to consider SCMs with a ground-truth DAG $\mathcal{G}$ that are members of this MEC $\tilde{\mathcal{G}}$ to distinguish among possible edge orientations in $\tilde{\mathcal{G}}$. In the case of undirected trees, the ground-truth DAG $\mathcal{G}$ must be a tree with no colliders and the same skeleton as $\tilde{\mathcal{G}}$, since any other DAGs would belong to a different MEC.

We give a proof by strong induction on the number of vertices $|\mathcal{V}|$ in the MEC $\tilde{\mathcal{G}}$. The base case of the induction argument is an MEC with $|\mathcal{V}| = 2$ nodes. This case holds trivially, since this MEC can contain at most one undirected edge. For the inductive step, we consider an undirected tree MEC $\tilde{\mathcal{G}}$ with $|\mathcal{V}| = d$ and assume that we can orient all but one edge of undirected tree MECs with $|\mathcal{V}| < d$.

Our argument will proceed by considering the longest chain of the undirected tree $\tilde{\mathcal{G}}$. We will use Lemma 5 to orient all but at most one edge in this chain and then apply the first Meek rule to possibly orient additional edges in $\tilde{\mathcal{G}}$ outside the chain. After orienting these edges, we show that we reduced the original problem of orienting all but one edge in $\tilde{\mathcal{G}}$ with $|\mathcal{V}| = d$ to orienting all but one edge in a single undirected connected component that has strictly fewer than $d$ nodes. This allows us to apply the inductive hypothesis and complete the proof (see Figure 9).

Consider a longest undirected chain $\tilde{\mathcal{G}}_C = (\mathcal{V}_C, \tilde{\mathcal{E}}_C)$ that is a subgraph of the undirected tree $\tilde{\mathcal{G}}$. Let $\mathcal{G}_C$ refer to the directed subgraph of the DAG $\mathcal{G}$ induced by considering only the vertices $\mathcal{V}_C$. We label the $k$ vertices in $\mathcal{V}_C$ as $v_1, ..., v_k$, with undirected edges $(v_i, v_{i+1}) \in \tilde{\mathcal{E}}$ for all $i \in \{1, ..., k-1\}$. The nodes $v_1, v_k$ can have no undirected neighbours in $\tilde{\mathcal{G}}$ outside the chain, because otherwise we could construct a longer chain in $\tilde{\mathcal{G}}$.

The only vertex in $\mathcal{V}_C$ that can have a parent in the DAG $\mathcal{G}$ outside the chain $\mathcal{G}_C$, that is, in $\mathcal{V} \backslash \mathcal{V}_C$, is the unique root of $\mathcal{G}_C$. To see this, we first note that all nodes $v_i$ have at most one parent in $\mathcal{G}$, because any $v_i$ with $|pa(v_i))| > 1$ in $\mathcal{G}$ would be a collider, but $\mathcal{G}$ contains no colliders. Since non-root nodes in $\mathcal{G}_C$ have an in-chain parent, they cannot have a parent outside of $\mathcal{V}_C$. Therefore, besides the root node of $\mathcal{G}_C$ via its potential outside parent, $\mathcal{G}_C$ is a completely disconnected subgraph from the rest of $\mathcal{G}$. This implies that we may treat $\mathcal{G}_C$ as a separate standardized SCM with undirected chain MEC, in which the potential parent of the root of $\mathcal{G}_C$ is modeled as part of the exogenous noise of the root. This allows us to apply Lemma 5 to the variables of the subgraph $\mathcal{G}_C$.

---

[1]Orienting edges of an undirected connected component that touch a directed edge in $\tilde{\mathcal{G}}$ never introduces an additional v-structure. If a directed edge pointed into the undirected connected component, the undirected edge downstream would have had to already be directed in $\tilde{\mathcal{G}}$ by the first Meek rule. Hence, all directed edges bordering the undirected connected component must be oriented away from it, and none of the possible undirected edge orientations creates a new collider at the border node. This implies that all undirected connected components in $\tilde{\mathcal{G}}$ are upstream of the colliders and directed subgraphs of $\tilde{\mathcal{G}}$.

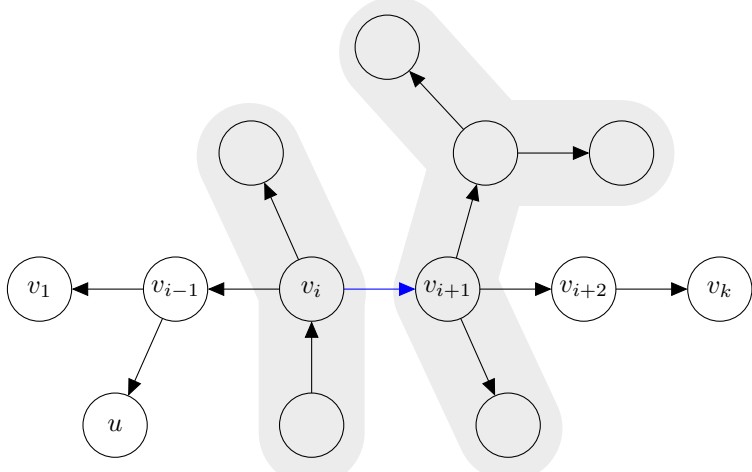

Figure 9: **Inductive step of the proof of Theorem 3.** Ground-truth DAG $\mathcal{G}$ underlying an undirected connected component $\tilde{\mathcal{G}}$ in some given MEC. The nodes $\mathcal{V}_C = \{v_1, \ldots, v_k\}$ are a longest chain in $\mathcal{G}$. Using Lemma 5, we can orient all edges in $\tilde{\mathcal{G}}_C$ except possibly $(v_i, v_{i+1})$ (blue). Edges like $(v_{i-1}, u)$ are oriented by the first Meek rule. After Lemma 5, we are left with either the single undirected tree of $v_i$ (left shaded tree) or the single undirected tree consisting of $(v_i, v_{i+1})$ (blue) and both undirected trees of $v_i$ and $v_{i+1}$ (both shaded trees). Either $v_i$ or $v_{i+1}$ must be the root of $\mathcal{G}_C$. In this specific example, $v_i$ is the root of $\mathcal{G}_C$ and is therefore the only node that can have a parent outside $\mathcal{G}_C$. Any node in $\mathcal{G}$ can have directed, outgoing edges to children in a (possibly non-forest) MEC the undirected connected component $\tilde{\mathcal{G}}$ may be a subgraph of.

By applying Lemma 5 to $\mathcal{G}_C$, we can orient all but at most one undirected edge in $\tilde{\mathcal{G}}_C$. We split the resulting analysis into the two cases of Lemma 5 leaving either 0 or 1 undirected edge. In the first case, we can orient all edges in $\tilde{\mathcal{G}}_C$ with Lemma 5. In this case, we know that the root of $\mathcal{G}_C$ is the node $v_i$ (see *Remark* of Lemma 5). By the first Meek rule, we can recursively orient all additional edges in $\tilde{\mathcal{G}}$ outside of $\tilde{\mathcal{G}}_C$ away from $v_i$, except for the subtrees of $\tilde{\mathcal{G}}$ connected to $v_i$ itself (Figure 9). This leaves at most a single connected undirected subtree containing $v_i$ and strictly less than $d$ vertices.

In the second case, we orient all but one edge $(v_i, v_{i+1})$ in $\tilde{\mathcal{G}}_C$ by applying Lemma 5. In this case, we know that the root of $\mathcal{G}_C$ is either the node $v_i$ or $v_{i+1}$ (see *Remark* of Lemma 5). Similar to the first case, we can recursively use the first Meek rule to orient all additional edges in $\tilde{\mathcal{G}}$ pointing away from $v_i$ and $v_{i+1}$, except for the subtrees of $\tilde{\mathcal{G}}$ connected to $v_i$ and $v_{i+1}$ itself. Since $v_i$ and $v_{i+1}$ are connected by an undirected edge, we are left with a single connected subtree containing the undirected edge $(v_i, v_{i+1})$ that is strictly smaller than before.

In both cases, we orient at least one undirected edge of $\tilde{\mathcal{G}}$, because the longest undirected chain in $\tilde{\mathcal{G}}$ with $|\mathcal{V}| > 2$ has at least length 2. We always obtain at most a single undirected connected tree component with strictly less than $d$ vertices, allowing us to apply the inductive hypothesis and complete the proof.

$\square$

ıSCM

**Theorem 4** (Nonidentifiability of linear Gaussian iSCMs with forest DAGs). *Let $\widetilde{\mathbf{x}}$ be modeled by a linear iSCM (1) with forest DAG $\mathcal{G}$ and additive Gaussian noise of equal variances $\mathrm{Var}[\varepsilon_i]$. Then, for every DAG $\mathcal{G}'$ in the MEC of $\mathcal{G}$, there exists a linear iSCM with DAG $\mathcal{G}'$ that has the same observational distribution as $\widetilde{\mathbf{x}}$, the same noise variances, and the same weights on the corresponding edges in the MEC.*

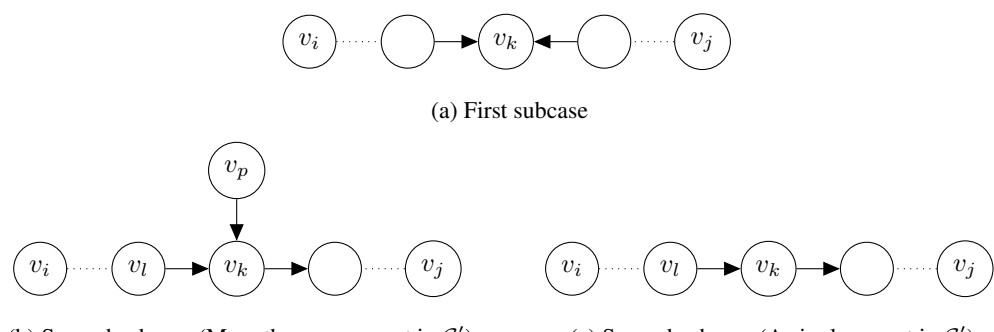

(a) First subcase

(b) Second subcase (More than one parent in $\mathcal{G}'$)    (c) Second subcase (A single parent in $\mathcal{G}'$)

Figure 10: **Proof subcases of Theorem 4.** (a) Path with a collider. In other words, a path blocked by an empty set. In the case of forests, this configuration implies that $v_i$ and $v_j$ are $d$-separated. (b) Unblocked path connecting $v_i$ and $v_j$ with one of the path nodes having a parent both in the path and outside the path. The weight $w_{p,k}$ influences the weight $\widetilde{w}_{l,k}$ in the implied model of the iSCM. If this structure is present in a forest, it has to be present in other graphs in the same MEC. (c) Unblocked path connecting $v_i$ and $v_j$ with the only parent of $v_k$ being part of the considered path. The weight $\widetilde{w}_{l,k}$ depends only on $w_{l,k}$, irrespective of the edge direction.

*Proof.* Because we consider linear iSCMs with Gaussian noise, the implied model is a linear SCM with additive Gaussian noise (see Appendix A.2). Hence, the observational distribution is a multivariate Gaussian with mean zero. In iSCMs, the marginal variance of an observed variable is always 1. Hence, we prove the statement if we show that for all $\widetilde{x}_i, \widetilde{x}_j$ in the iSCM with graph $\mathcal{G}$, and the corresponding $\widetilde{x}_i', \widetilde{x}_j'$ in the iSCM with graph $\mathcal{G}' = (\mathcal{V}, \mathcal{E}')$, $\mathrm{Cov}[\widetilde{x}_i, \widetilde{x}_j] = \mathrm{Cov}[\widetilde{x}_i', \widetilde{x}_j']$.

Let $\widetilde{x}_i'$ and $\widetilde{x}_j'$ be the random variables associated with the nodes $v_i$ and $v_j$ from $\mathcal{G}'$, respectively. We consider two cases. First, if there is no path between $v_i$ and $v_j$ in the skeleton of $\mathcal{G}'$ then there is no path between $v_i$ and $v_j$ in the skeleton of $\mathcal{G}$ and hence $\mathrm{Cov}[\widetilde{x}_i, \widetilde{x}_j] = \mathrm{Cov}[\widetilde{x}_i', \widetilde{x}_j'] = 0$. In the second case, there is a path between $v_i$ and $v_j$ in the skeleton of $\mathcal{G}'$, so there also exists a path in the skeleton of $\mathcal{G}$, as both graphs have the same skeleton. Due to the acyclicity of the skeleton in forests, this path is the only one connecting $v_i$ and $v_j$ in both $\mathcal{G}$ and $\mathcal{G}'$.

We further break this second case into two subcases. In the first subcase, this path contains a collider in $\mathcal{G}$ as shown in Figure 10a. Because the skeleton cannot have undirected cycles under the forest assumption, this collider forms a $v$-structure. $\mathcal{G}' \in \tilde{\mathcal{G}}$ implies that the same $v$-structure must be present in $\mathcal{G}$. Hence, $v_i$ and $v_j$ are $d$-separated in both $\mathcal{G}$ and $\mathcal{G}'$. By the global Markov condition, this implies that $\widetilde{x}_i'$ and $\widetilde{x}_j'$ are independent, and that $\widetilde{x}_i$ and $\widetilde{x}_j$ are independent. This implies that both $\mathrm{Cov}[\widetilde{x}_i', \widetilde{x}_j'] = \mathrm{Cov}[\widetilde{x}_i, \widetilde{x}_i] = 0$.

In the second subcase, there exists an unblocked path between $v_i$ and $v_j$ in both $\mathcal{G}$ and $\mathcal{G}'$. Here, we denote the weight matrix associated with both iSCMs by $W := [w_{i,j}]$, with $W$ being symmetric, so that $w_{i,j} = w_{j,i}$ is the linear weight of the edge $(i, j)$ regardless of its orientation in the graph.

We now derive the analogous weights $\widetilde{W}, \widetilde{W}'$ in the implied SCMs for $\mathcal{G}, \mathcal{G}'$ respectively. Ultimately, we will demonstrate that the implied SCMs have the same weights. Specifically, we will show that $\widetilde{w}_{k,l} = \widetilde{w}_{k,l}'$. Given this, Lemma 1 implies that both iSCMs have the same covariance matrix over the observed variables.

Without loss of generality, since the node labelling is arbitrary, let $v_k$ have at least as many incoming edges as $v_l$ in $\mathcal{G}'$. We divide the analysis into two cases: $v_k$ having only 1 parent in $\mathcal{G}'$, and $v_k$ having more than 1 parent. The node $v_k$ must have at least one parent, since at least one of $v_k, v_l$ have an incoming edge in $\mathcal{G}'$, and we chose $v_k$ to have at least as many incoming edges as $v_l$.

**More than one parent in $\mathcal{G}'$**  We know that any collider in $\mathcal{G}'$ will appear as part of a $v$-structure in $\tilde{\mathcal{G}}$ due to the forest assumption, and therefore will also be a collider in $\mathcal{G}$. Therefore, if $v_k$ has more than one parent in $\mathcal{G}'$ (see Figure 10b), all pairs of edges incoming to $v_k$ will form $v$-structures, so $v_k$ must have exactly the same set of parents in $\mathcal{G}$.

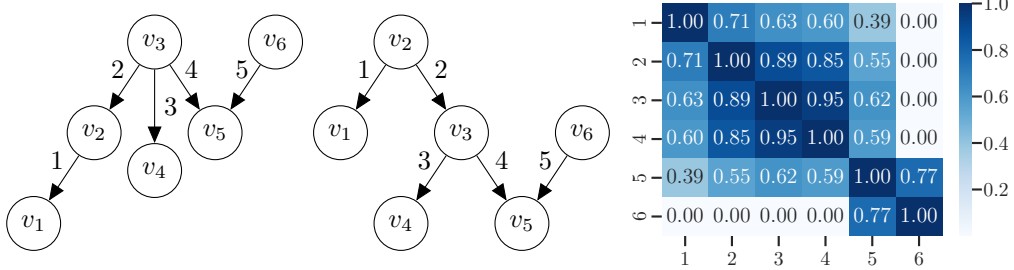

Figure 11: **Illustrating Theorem 4 for trees in the same MEC.** Covariance matrix of observed iSCM variables for two example forests belonging to the same MEC with the same weights assigned to the edges of the skeleton.

Moreover, any two parents of $v_k$ are d-separated in $\mathcal{G}$ and $\mathcal{G}'$ by the forest assumption, since the blocked path going through $v_k$ is the only path connecting them. By the global Markov condition, the parents are pairwise independent. Hence, we can use Equation (11) to compute $\widetilde{w}_{k,l}, \widetilde{w}'_{k,l}$. Since the parent sets are the same between the two graphs, and $W$ is shared between the two iSCMs, the weight associated with the edge $(l, k)$ in both graphs in the implied models is given by

$$\widetilde{w}_{l,k} = \widetilde{w}'_{l,k} = \frac{w_{l,k}}{\sqrt{\sum_{u \in \mathrm{pa}(k)} w_{u,k}^2 + \sigma^2}} \, . \tag{24}$$

**A single parent in $\mathcal{G}'$** Let $(l, k)$ be the only incoming edge to $v_k$ in $\mathcal{G}'$, as depicted in Figure 10c. Then, the edge connecting $v_l$ and $v_k$ in $\mathcal{G}$ is either the only incoming edge to $v_k$ or the only incoming edge to $v_l$. To see this, suppose that it was not the only incoming edge to $v_k$ or $v_l$ in $\mathcal{G}$. This would make $v_k$ or $v_l$ a collider that would be common to both graphs, implying that $v_k$ or $v_l$ would have at least two parents in $\mathcal{G}'$. We operate under the assumption that $v_k$ has at least as many parents as $v_l$, so it would imply that $v_k$ has more than one parent, contradicting the assumption we made for case we consider in this paragraph. Irrespective of the direction, the weight associated with the edge $(l, k)$ in the skeleton of both graphs in the implied model is, similar to Equation (21), given by

$$\widetilde{w}_{l,k} = \widetilde{w}'_{l,k} = \frac{w_{l,k}}{\sqrt{w_{l,k}^2 + \sigma^2}} \, . \tag{25}$$

Equations (24) and (25) show that, for the SCM form of each iSCM, the edges connecting the same nodes irrespective of their direction in $\mathcal{G}'$ and $\mathcal{G}$ have the same weights. By Lemma 1, the covariance between any $\widetilde{x}_i$ and $\widetilde{x}_j$ can be expressed as a product of the weights in the implied SCM corresponding to the edges on the path between $v_i, v_j$. Hence, $\mathrm{Cov}[\widetilde{x}_i, \widetilde{x}_j] = \mathrm{Cov}[\widetilde{x}'_i, \widetilde{x}'_j]$. □

Figure 11 shows an example for Theorem 4 for two trees from the same MEC.

**Remark** In Figure 12, we empirically demonstrate that Theorem 4 no longer holds if we drop the forest assumption. For data generated from an iSCM and two graphs from the same $\tilde{\mathcal{G}}$ with the same weights assigned to the skeleton edges, we observe that the estimated covariances differ. The two systems entail different observational distributions.

## D  BACKGROUND ON RELATED WORK

### D.1  HEURISTICS FOR MITIGATING VARIANCE ACCUMULATION

Here, we review existing heuristics for avoiding the exploding variance in structure learning benchmarking with linear SCMs as defined in Equation (1). We also describe how these heuristics limit the causal dependencies that can be modeled in terms of the correlations among the SCM variables or their cause-explained variance, both of which do not occur in linear iSCMs. Finally, in Figure 13 we show that **the heuristics fail to induce data that is both not** $\mathrm{Var}$-**sortable and not** $R^2$-**sortable.**

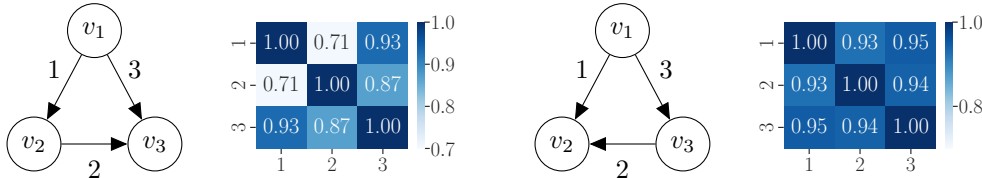

Figure 12: **Non-forest counterexample for Theorem 4.** Covariance matrix of observed iSCM variables for two non-forests belonging to the same MEC with the same weights assigned to the edges of the skeleton.

**Scaling weights by the inverse weight norm**   Mooij et al. (2020, Section 5.2) sample the edge weights in linear SCMs as $w_{i,j} \sim \text{Unif}_{\pm}\,[0.5, 1.5]$. To achieve a comparable variance of each variable $x_j$ in the SCM, they propose re-scaling the sampled weights prior to the data-generating process as

$$w_{i,j} \leftarrow \frac{w_{i,j}}{\sqrt{1 + \sum_{i \in \text{pa}(j)} w_{i,j}^2}}\,.$$

If all parents of $x_j$ are *i.i.d.* Gaussian with variance 1, this adjustment ensures that the variance of $x_j$ is similar for all $x_j$. However, this approximation does not take into account the covariances of the parents. Moreover, since $\text{Var}[\varepsilon_j]$ is unchanged, the scaling limits the strength of the causal effect that parents can have on $x_j$. For example, when $x_1 = \varepsilon_1$ and $x_2 = wx_1 + \varepsilon_2$ with $\text{Var}[\varepsilon_j] = 1$ as for Mooij et al. (2020), the adjusted weight is $w' = w/\sqrt{1 + w^2} < 1$. Thus, for any $w \neq 0$, we have

$$|\text{Corr}[x_1, x_2]| = \frac{|\text{Cov}[\varepsilon_1, w'\varepsilon_1 + \varepsilon_2]|}{\sqrt{\text{Var}[\varepsilon_1]\,\text{Var}[w'\varepsilon_1 + \varepsilon_2]}} = \frac{|w'|}{\sqrt{w'^2 + 1}} < \frac{1}{\sqrt{2}} \approx 0.707\,.$$

This is the maximum correlation between neighbouring variables that any SCM can model under the proposed re-scaling when $\text{Var}[\varepsilon_j] = 1$, since additional parents decrease the parent-child correlations. By contrast, iSCMs can model any level of correlation by sampling arbitrary values of $w_{i,j}$, while guaranteeing unit-variance observations $x_j$. Intuitively, iSCMs achieve this by standardizing $x_j$ after the exogenous noise $\varepsilon_j$ is added to the endogenous contributions of the parents $\mathbf{x}_{\text{pa}(j)}$, while weight scaling is done before $\varepsilon_j$ is added to $x_j$.

**Scaling weights by the incoming variance**   Squires et al. (2022, Section 5.1) sample the weights of linear SCMs as $w_{i,j} \sim \text{Unif}_{\pm}\,[0.25, 1.0]$. Given the initial edge weights, they propose adjusting the weights during the generative process by first estimating the total variance $\hat{\sigma}_j^2$ that the parents of $x_j$ contribute to $x_j$ from samples drawn under an initial level of additive noise with $\text{Var}[\varepsilon_j] = 1$ and then re-scaling the weights as

$$w_{i,j} \leftarrow \frac{w_{i,j}}{\sqrt{2\hat{\sigma}_j^2}}\,.$$

When using additive noise with $\text{Var}[\varepsilon_j] = 0.5$ to generate the actual samples, this scaling results in $\text{Var}[x_j] = 1$ with a constant fraction of cause-explained variance $\text{CEV}_{\text{f}}[x_i] = 0.5$. In benchmarks, however, we may be interested in evaluating SCMs with arbitrary levels of cause-explained variance. iSCMs allow this by construction. Contrary to Squires et al. (2022), iSCMs scale the variables $x_j$ rather than the weights $w_{i,j}$ while leaving the exogenous noise $\varepsilon_j$ unchanged, which enables modeling arbitrarily small or large levels of unexplained variation.

## D.2   SORTABILITY METRICS

In this section, we describe the definition of a sortability metric as introduced by Reisach et al. (2024), which we use in Section 5. For a function $\tau$, $\tau$-sortability assigns a scalar in $[0, 1]$ to the variables $\mathbf{x}$ and graph $\mathcal{G}$ (with weight matrix $W_{\mathcal{G}}$) as

$$\frac{\sum_{i=1}^{d} \sum_{p_{s \to t} \in W_{\mathcal{G}}^i} \text{incr}(\tau(\mathbf{x}, s), \tau(\mathbf{x}, t))}{\sum_{i=1}^{d} \sum_{p_{s \to t} \in W_{\mathcal{G}}^i} 1} \qquad \text{where incr}(a, b) = \begin{cases} 1 & \text{if } a < b \\ \frac{1}{2} & \text{if } a = b \\ 0 & \text{if } a > b \end{cases}$$

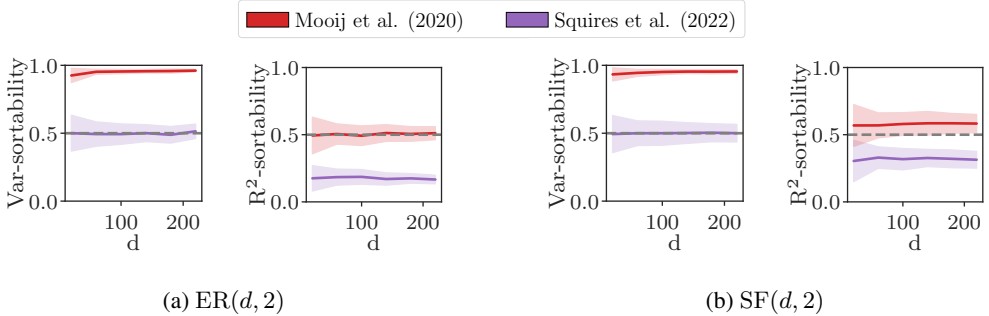

Figure 13: **Sortabilities for data generated according to heuristics that aim to remove artifacts**. Weight ranges were assumed as in the original papers: $w_{i,j} \sim \pm \text{Unif}_{[0.5,1.5]}$ for Mooij et al. (2020), $w_{i,j} \sim \pm \text{Unif}_{[0.25,1.0]}$ for Squires et al. (2022). For every model, we evaluate 100 systems each using $n = 1000$ samples. Lines and shaded regions denote mean and standard deviation respectively.

and $W_{\mathcal{G}}^i$ is the $i$-th power of the adjacency matrix $W_{\mathcal{G}}$ and $p_{s \to t} \in W_{\mathcal{G}}^i$ if and only if at least one directed path from $v_s$ to $v_t$ of length $i$ exists in $\mathcal{G}$. If $\tau(\mathbf{x}, t) = \text{Var}[x_t]$, we obtain Var-sortability from Reisach et al. (2021). If

$$\tau(\mathbf{x}, t) = R^2[x_t] = 1 - \frac{\text{Var}[x_t - \mathbb{E}[x_t | \mathbf{x}_{\{1,...,d\} \setminus \{t\}}]]}{\text{Var}[x_t]},$$

we obtain $R^2$-sortability. Estimating $R^2[x_t]$ requires performing regression of $x_t$ onto $\mathbf{x}_{\{1,...,d\} \setminus \{t\}}$.

## D.3 STRUCTURE LEARNING ALGORITHMS

To complement the interpretation of the results in Section 5, we provide some background on the structure learning methods we evaluate.

**NOTEARS (Zheng et al., 2018)** NOTEARS uses continuous optimization to minimize the regularized mean-squared error (MSE) between the the variables modeled by a linear SCM and the observations, while enforcing a differentiable acyclicity constraint. The objective function of NOTEARS is given by $F(\mathbf{W}) = ||\mathbf{X} - \mathbf{X}\mathbf{W}||_F^2 / 2n + \lambda ||\mathbf{W}||_1$, where $|| \cdot ||_F$ and $|| \cdot ||_1$ are a Frobenius and $\ell_1$ norm respectively. When the objective is minimized, weights below a fixed threshold are set to zero.

**AVICI (Lorch et al., 2022)** AVICI is an amortized variational inference method that approximates the posterior distribution over causal structures given a dataset through a pretrained inference model. The variational approximation of AVICI uses a fully-factored product of Bernoulli distributions for every possible graph edge. The inference model is a neural network that predict the variational parameters of the Bernoulli distributions by minimizing the expected forward KL divergence between the true posterior and the approximation. To train the inference model, AVICI can be optimized on any training distribution of (synthetic) dataset-graph pairs. Lorch et al. (2022) publish the pretrained parameters of inference models trained on standardized SCMs with linear and nonlinear mechanisms, which we evaluate in this work.

**SortnRegress methods (Reisach et al., 2021; 2024)** The SORTNREGRESS methods order the vertices by a chosen statistic and sparsely regress every node on all of its predecessors in the obtained order. They use Lasso regression with the Bayesian Information Criterion to learn the regression function for a given variable. Var-SORTNREGRESS uses estimated marginal variances as the sorting criterion. $R^2$-SORTNREGRESS uses $R^2$ coefficient of determination estimated after performing a regression of every variable onto all remaining variables. RAND-SORTNREGRESS orders the vertices randomly.

# E  EXPERIMENTAL SETUP

## E.1  DATA

**Causal mechanisms**    We consider systems with additive noise, where

$$f_i(\mathbf{x}, \varepsilon_i) = h_i(\mathbf{x}) + \varepsilon_i,$$

for a chosen function $h_i$. The LINEAR systems used in this experiments have causal mechanisms as defined in Equation (1). To model nonlinear systems, we use smooth nonlinear functional mechanisms as used by Lorch et al. (2022). Specifically, the function $h_i$ that models the relationship between $x_i$ and its parents is sampled from a Gaussian Process

$$h_i \sim \mathcal{GP}(0, k_i),$$

where $k$ is a squared exponential kernel $k_i(\mathbf{x}, \mathbf{x}') = c_i^2 \exp\left(-||\mathbf{x} - \mathbf{x}'||_2^2 / 2l_i^2\right)$ with output and length scales $c_i$ and $l_i$ respectively. We can approximately express the function sample $h_i$ analytically using random Fourier features (Rahimi and Recht, 2007) by sampling

$$h_i(\mathbf{x}) = c_i \sqrt{\tfrac{2}{M}} \sum_{j=1}^{M} \alpha^{(i)} \cos\left(\tfrac{\omega^{(i)} \cdot \mathbf{x}}{l_i} + \delta^{(i)}\right)$$

where $\alpha^{(i)} \sim \mathcal{N}(0,1)$, $\omega^{(i)} \sim \mathcal{N}(0,\mathbf{I})$, and $\delta^{(i)} \sim \text{Unif}\,[0, 2\pi]$. In this work, we use $M = 100$.

**Generating a random model**    Following prior work (Section 2), we sample random systems in any simulation performed in this work by first drawing a graph $\mathcal{G}$ from the specified random graph distribution. Given the graph $\mathcal{G}$, we sample function parameters of the structural mechanisms over $\mathcal{G}$. For linear systems, we sample $w_{i,j} \sim \text{Unif}_{\pm}[a, b]$, where $a, b$ are fixed, *i.i.d.* for every graph edge. Similarly, for nonlinear systems, for every graph vertex, we draw the length scales $l_i \sim \text{Unif}[a_1, b_1]$ and output scales $c_i \sim \text{Unif}[a_2, b_2]$ with predefined $a_1, b_1, a_2, b_2$.

**Sampling data from a model**    Given a graph $\mathcal{G}$, noise distribution $\mathcal{P}_{\boldsymbol{\varepsilon}}$, and a set of functions $\{f_1, ... f_d\}$, we sample $n$ datapoints from an SCM by traversing $\mathcal{G}$ in a topological ordering. For every vertex $v_i$, we draw a noise sample $\varepsilon_i \sim \mathcal{P}_{\varepsilon_i}^n$. The sample for $x_i$ is then deterministically computed by $f_i$ from the exogenous $\varepsilon_i$ and the parents of $x_i$. To sample from a Standardized SCM, we draw a dataset from an SCM and standardize it. To sample from an iSCM, we use Algorithm 1.

## E.2  EXPERIMENT CONFIGURATIONS

**Sortability**    For Figures 4a, 16a, and 17a we generate Erdős-Rényi graphs $\text{ER}(d, k)$ (Erdős and Rényi, 1959), with $d$ denoting the graph size and $k$ the expected node degree. For Figures 4b, 16b, and 17b we generate undirected scale-free graphs $\text{SF}(d, k)$ (Barabási and Albert, 1999), where $d$ is the graph size and $k$ the number of outgoing edges generated for each vertex. Then, we orient the edges in the graph according to a random topological ordering. We do not sample directed scale-free graphs initially to avoid high sortability by in-degree, which may confound the results.

For all four figures, we generate LINEAR systems with weights sampled from three possible distributions $w_{i,j} \sim \text{Unif}_{\pm}[0.3, 1.8]$, $w_{i,j} \sim \text{Unif}_{\pm}[0.5, 2.0]$ or $w_{i,j} \sim \text{Unif}_{\pm}[1.3, 3.0]$ and noise sampled from $\varepsilon_i \sim \mathcal{N}(0, 1)$. For every model configuration, we sample 100 systems and $n = 1000$ data points each. To create Figures 4 and 16 we sampled graphs of sizes $\{20, 60, 100, 140, 180, 220\}$. To obtain Figure 17, we sampled graphs with $k \in \{4, 8, 12, 16, 20\}$.

**Structure Learning (Section 5.2)**    For Figures 5 and 14, we sample LINEAR systems with weights $w_{i,j} \sim \text{Unif}_{\pm}[0.5, 2.0]$. Following Lorch et al. (2022), NONLINEAR mechanisms have length scales $l_i \sim \text{Unif}[7.0, 10.0]$ and output scales $c_i \sim \text{Unif}[10.0, 20.0]$. Both mechanisms are defined in Appendix E.1. For Figures 15a and 15b, we generate LINEAR systems with weights $w_{i,j} \sim \text{Unif}_{\pm}[0.3, 0.8]$ and $w_{i,j} \sim \text{Unif}_{\pm}[1.3, 3.0]$. For all four figures, we sample random $\text{ER}(20, 2)$ and $\text{ER}(100, 2)$ graphs with noise $\varepsilon_i \sim \mathcal{N}(0, 1)$. For every model configuration, we sample 20 systems and $n = 1000$ data points each.

**Noise Transfer** For Figure 6 (top), we sample SCMs, standardized SCMs, and iSCMs with exactly the same underlying graph and weights sampled from $w_{i,j} \sim \text{Unif}_{\pm}[0.5, 2.0]$. The noise variables are drawn from $\varepsilon_i \sim \mathcal{N}(0, 1)$. Then, for every triple of SCM, standardized SCM, and iSCM that shares a graph and weights, we create two more SCMs with the same marginal variances as the SCM, but with the noise variances of the implied models of the standardized SCM and iSCM, respectively. Appendix E.5 provides a motivation and detailed explanation of this procedure. Figure 6 (top) shows the performance of NOTEARS on the original SCMs and the two SCMs with transferred noise.

For Figure 6 (bottom), we sample multiple instances of standardized SCMs, and iSCMs with weights drawn from $w_{i,j} \sim \text{Unif}_{\pm}[0.5, 2.0]$ and noise from $\varepsilon_i \sim \mathcal{N}(0, 1)$. For every model instance, we approximate the density of the inverse of their implied noise variances using kernel density estimation. The figure shows the mean and standard deviation of the p.d.f. values over 100 systems. For both figures, we use $\text{ER}(100, 2)$ graphs.

### E.3 METHODS

**NOTEARS (Zheng et al., 2018)** To run NOTEARS, we use the original implementation provided by the authors of Zheng et al. (2018) (Apache-2.0 license). Before benchmarking NOTEARS, we run a hyperparameter search to calibrate the weight penalty ($\lambda$) and threshold on held-out instances of each data generation method. The hyperparameters can be found in Appendix E.4.

**AVICI (Lorch et al., 2022)** To evaluate AVICI, we use the code and model checkpoints provided by the authors of the method (MIT license). Specifically, we use the model trained on linear data to benchmark the method on LINEAR systems and the model trained on nonlinear data to benchmark on NONLINEAR systems. We score an edge as predicted if the probability prediction by AVICI is greater than $0.5$. Since the parameters are pretrained, the method has otherwise no tuneable hyperparameters.

**Sortabilities and SORTNREGRESS methods (Reisach et al., 2021; 2024)** To compute the sortability metrics and run the SORTNREGRESS baselines, we use the `CausalDisco` library (BSD-3-Clause license) created by the authors of the method. The algorithms require no tuneable hyperparameters.

**GOLEM (Ng et al., 2020)** For GOLEM-EV, we tune $\lambda_1$ (sparsity penalty coefficient), $\lambda_2$ (acyclicity penalty coefficient) and the threshold for zeroing weights. For GOLEM-NV, we tune the same hyperparameters as for GOLEM-EV. We do not initialize the model with the solution returned by GOLEM-EV, as done in the original paper, since we want to evaluate a method that does not assume equal noise variances at any point. Not initializing with the GOLEM-EV weights is consistent with the benchmarking approach of Reisach et al. (2021). We use the implementation of the original work (Ng et al., 2020).

**PC Algorithm** For linear data, we use a Gaussian conditional independence test. For nonlinear data, we use the Hilbert-Schmidt Independence Criterion (HSIC) gamma test. We treat the test significance level as a hyperparameter that we tune. We use the implementation by the Causal Discovery Toolbox (Kalainathan et al., 2020).

**GES** GES uses the linear Gaussian BIC score function and does not require hyperparameter tuning. We use the implementation by the Causal Discovery Toolbox (Kalainathan et al., 2020).

**CAM** CAM estimates a causal ordering using maximum likelihood and then performs sparse nonlinear regression using splines on the possible parents in this ordering. We use the implementation from the `dodiscover` library (MIT license) and include the preliminary neighbor search option to make the algorithm scale to large graphs. We tune the cutoff value $\alpha$ for variable selection with hypothesis testing over regression coefficients, and the number and order of splines to use for the feature function.

**LINGAM** LINGAM uses independent component analysis, an algorithm for source separation, to find a causal ordering, which is identifiable in linear systems if the additive noise in an SCM is non-Gaussian. We use the implementation from the `cdt` (Causal Discovery Toolbox) library (MIT license).

### E.4 Hyperparameter Selection

For all algorithms that require hyperparameter tuning, we perform the search on separate, held-out systems that follow the same configurations as the ones we present in our final experimental results. We run the algorithms 20 times per configuration and choose the median F1 score as the criterion for selecting the best hyperparameters.

To run NOTEARS, we need to specify the regularisation strength $\lambda$ and a weight threshold $\eta$ for thresholding the final weights for graph structure prediction. To select these hyperparameters, we run a parameter search with $\lambda \in \{0.0, 0.05, 0.1, 0.15, 0.2, 0.25, 0.3\}$ and three possible values of the weight threshold $\{0.1, 0.2, 0.3\}$. Table 1 presents all final hyperparameter configurations for NOTEARS. For some hyperparameter configurations, 1 in 20 runs experienced numerical issues caused by the acyclicity constraint. However, this never occurs for the selected, optimal hyperparameters, neither when performing the hyperparameter search nor when running the reported experiments.

To run the PC algorithm, one needs to choose a test significance level $\alpha$. During the hyperparamter search we consider $\alpha \in \{0.01, 0.001, 0.0001\}$. Table 4 presents all final hyperparameter configurations for the PC algorithm.

To run GOLEM-EV and GOLEM-NV we need to tune sparsity penalty coefficient $\lambda_1$, acyclicity penalty coefficient $\lambda_2$ and the weight threshold $\eta$. We consider $\lambda_1 \in \{0.02, 0.002, 0.0002\}$, $\lambda_2 \in \{2.0, 5.0, 8.0\}$ and $\eta \in \{0.1, 0.2, 0.3\}$. Tables 2 and 3 present the best configurations.

To run CAM we need to tune the cutoff value $\alpha \in \{0.05, 0.10, 0.15\}$ for variable selection with hypothesis testing over regression coefficients and the number and order of splines to use for the feature function for which we consider sets $\{5, 10\}$ and $\{2, 3\}$ respectively. Table 5 presents the best configurations.

### E.5 Transferring Noise Variances While Keeping Var-Sortability Unchanged

Reisach et al. (2021) show that post-hoc standardization of SCM data strongly impairs the performance of NOTEARS. When comparing the performance of NOTEARS between data sampled from iSCMs and standardized SCMs, there are at least two factors that can affect the performance of NOTEARS, low Var-sortability and the violation of the equal noise variance assumption. Our experiments in Figure 6 of Section 5 aim at isolating the effect of the latter. Specifically, we investigate whether NOTEARS performs better on Var-sortable datasets that have the noise scale patterns implied when assuming SCMs generated the data—when in fact the data was sampled from iSCMs or standardized SCMs. To achieve this, we ensure that the Var-sortability metrics of the data sampled from the models is the same, here close to 1.

Given two linear SCMs $S^a$ and $S^b$ with the same underlying graph $\mathcal{G}$, our goal is to construct a system $S^t$ with the same marginal variances as $S^a$ (condition 1) and the same noise variances as $S^b$ (condition 2). For this task to be well-defined, we assume that the noise variances of the root variables in $S^a$ and $S^b$ are the same. The first step in constructing $S^t$ is to copy the noise variances from $S^b$, so that for every $i \in \{1, ..., d\}$.

$$\sigma_i^{2^t} := \sigma_i^{2^b}.$$

This satisfies condition 2. Given this, we define $x_i^t$ as

$$x_i^t := \sqrt{\frac{\mathrm{Var}[x_i^a] - \sigma_i^{2^b}}{\mathrm{Var}[\mathbf{w}_i^{aT}\mathbf{x}_{\mathrm{pa}(i)}^t]}} \mathbf{w}_i^{aT}\mathbf{x}_{\mathrm{pa}(i)}^t + \varepsilon_i^t,$$

where $\varepsilon_i^t$ has variance $\sigma_i^{2^t}$. By construction, the condition of $S^t$ sharing the noise variances with $S^b$ and the marginal variances with $S^a$ is fulfilled for the root variables. For all the remaining variables,

Table 1: **NOTEARS hyperparameters for all experiments.** Final settings for the regularization strength $\lambda$ and the weight threshold $\eta$ after hyperparameter tuning on the respective models and data-generating processes together with the F1 (median) validation scores achieved by NOTEARS.

(a) ER$(20, 2)$ DAGs, LINEAR mechanisms

| Weight Distribution | Model | $\lambda$ | $\eta$ | F1 (median) |
|---|---|---|---|---|
| Unif$_\pm [0.3, 0.8]$ | SCM | 0.05 | 0.20 | 0.97 |
| Unif$_\pm [0.3, 0.8]$ | Standardized SCM | 0.15 | 0.10 | 0.59 |
| Unif$_\pm [0.3, 0.8]$ | iSCM | 0.15 | 0.10 | 0.57 |
| Unif$_\pm [0.5, 2.0]$ | SCM | 0.00 | 0.30 | 0.98 |
| Unif$_\pm [0.5, 2.0]$ | Standardized SCM | 0.15 | 0.20 | 0.30 |
| Unif$_\pm [0.5, 2.0]$ | iSCM | 0.15 | 0.10 | 0.50 |
| Unif$_\pm [1.3, 3.0]$ | SCM | 0.05 | 0.30 | 0.98 |
| Unif$_\pm [1.3, 3.0]$ | Standardized SCM | 0.25 | 0.10 | 0.24 |
| Unif$_\pm [1.3, 3.0]$ | iSCM | 0.20 | 0.10 | 0.40 |

(b) ER$(100, 2)$ DAGs, LINEAR mechanisms

| Weight Distribution | Model | $\lambda$ | $\eta$ | F1 (median) |
|---|---|---|---|---|
| Unif$_\pm [0.3, 0.8]$ | SCM | 0.10 | 0.10 | 0.99 |
| Unif$_\pm [0.3, 0.8]$ | Standardized SCM | 0.10 | 0.10 | 0.83 |
| Unif$_\pm [0.3, 0.8]$ | iSCM | 0.10 | 0.10 | 0.84 |
| Unif$_\pm [0.5, 2.0]$ | SCM | 0.05 | 0.30 | 0.94 |
| Unif$_\pm [0.5, 2.0]$ | Standardized SCM | 0.15 | 0.10 | 0.47 |
| Unif$_\pm [0.5, 2.0]$ | iSCM | 0.15 | 0.10 | 0.76 |
| Unif$_\pm [1.3, 3.0]$ | SCM | 0.10 | 0.30 | 0.82 |
| Unif$_\pm [1.3, 3.0]$ | Standardized SCM | 0.20 | 0.10 | 0.30 |
| Unif$_\pm [1.3, 3.0]$ | iSCM | 0.15 | 0.10 | 0.70 |

(c) ER$(20, 2)$ DAGs, NONLINEAR mechanisms

| Model | $\lambda$ | $\eta$ | F1 (median) |
|---|---|---|---|
| SCM | 0.15 | 0.30 | 0.58 |
| Standardized SCM | 0.15 | 0.10 | 0.33 |
| iSCM | 0.15 | 0.20 | 0.42 |

(d) ER$(100, 2)$ DAGs, NONLINEAR mechanisms

| Model | $\lambda$ | $\eta$ | F1 (median) |
|---|---|---|---|
| SCM | 0.30 | 0.30 | 0.50 |
| Standardized SCM | 0.15 | 0.10 | 0.43 |
| iSCM | 0.15 | 0.10 | 0.61 |

(e) Noise transfer experiment: ER$(100, 2)$ DAGs, LINEAR mechanisms $w_{ij} \sim$ Unif$_\pm [0.5, 2.0]$

| Model | $\lambda$ | $\eta$ | F1 (median) |
|---|---|---|---|
| Original | 0.05 | 0.30 | 0.96 |
| Noise from standardized SCM | 0.10 | 0.30 | 0.72 |
| Noise from iSCM | 0.05 | 0.30 | 0.82 |

Table 2: **GOLEM-EV hyperparameters for all experiments.** Final settings for the sparsity penalty coefficient $\lambda_1$, acyclicity penalty coefficient $\lambda_2$ and the weight threshold $\eta$ after hyperparameter tuning on the respective models and data-generating processes together with the F1 (median) validation scores achieved by GOLEM-EV.

(a) ER$(20, 2)$ DAGs, LINEAR mechanisms

| Weight Distribution | Model | $\lambda_1$ | $\lambda_2$ | $\eta$ | F1 (median) |
|---|---|---|---|---|---|
| Unif$_{\pm}$ [0.5, 2.0] | SCM | 0.002 | 5.00 | 0.30 | 1.00 |
| Unif$_{\pm}$ [0.5, 2.0] | Standardized SCM | 0.020 | 8.00 | 0.10 | 0.15 |
| Unif$_{\pm}$ [0.5, 2.0] | iSCM | 0.020 | 2.00 | 0.10 | 0.36 |
| Unif$_{\pm}$ [1.3, 3.0] | SCM | 0.002 | 5.00 | 0.30 | 1.00 |
| Unif$_{\pm}$ [1.3, 3.0] | Standardized SCM | 0.020 | 8.00 | 0.10 | 0.12 |
| Unif$_{\pm}$ [1.3, 3.0] | iSCM | 0.020 | 5.00 | 0.10 | 0.34 |
| Unif$_{\pm}$ [0.3, 0.8] | SCM | 0.020 | 2.00 | 0.10 | 1.00 |
| Unif$_{\pm}$ [0.3, 0.8] | Standardized SCM | 0.020 | 5.00 | 0.10 | 0.33 |
| Unif$_{\pm}$ [0.3, 0.8] | iSCM | 0.020 | 5.00 | 0.10 | 0.36 |

(b) ER$(100, 2)$ DAGs, LINEAR mechanisms

| Weight Distribution | Model | $\lambda_1$ | $\lambda_2$ | $\eta$ | F1 (median) |
|---|---|---|---|---|---|
| Unif$_{\pm}$ [0.5, 2.0] | SCM | 0.020 | 2.00 | 0.20 | 1.00 |
| Unif$_{\pm}$ [0.5, 2.0] | Standardized SCM | 0.020 | 8.00 | 0.10 | 0.13 |
| Unif$_{\pm}$ [0.5, 2.0] | iSCM | 0.020 | 5.00 | 0.10 | 0.24 |
| Unif$_{\pm}$ [1.3, 3.0] | SCM | 0.020 | 8.00 | 0.30 | 0.90 |
| Unif$_{\pm}$ [1.3, 3.0] | Standardized SCM | 0.020 | 5.00 | 0.10 | 0.08 |
| Unif$_{\pm}$ [1.3, 3.0] | iSCM | 0.020 | 5.00 | 0.10 | 0.19 |
| Unif$_{\pm}$ [0.3, 0.8] | SCM | 0.020 | 2.00 | 0.20 | 1.00 |
| Unif$_{\pm}$ [0.3, 0.8] | Standardized SCM | 0.020 | 5.00 | 0.10 | 0.30 |
| Unif$_{\pm}$ [0.3, 0.8] | iSCM | 0.020 | 2.00 | 0.10 | 0.40 |

(c) ER$(20, 2)$ DAGs, NONLINEAR mechanisms

| Model | $\lambda_1$ | $\lambda_2$ | $\eta$ | F1 (median) |
|---|---|---|---|---|
| SCM | 0.020 | 8.00 | 0.30 | 0.39 |
| Standardized SCM | 0.002 | 8.00 | 0.10 | 0.20 |
| iSCM | 0.020 | 2.00 | 0.10 | 0.25 |

(d) ER$(100, 2)$ DAGs, NONLINEAR mechanisms

| Model | $\lambda_1$ | $\lambda_2$ | $\eta$ | F1 (median) |
|---|---|---|---|---|
| SCM | 0.020 | 8.00 | 0.10 | 0.27 |
| Standardized SCM | 0.020 | 8.00 | 0.10 | 0.14 |
| iSCM | 0.020 | 5.00 | 0.10 | 0.14 |

Table 3: **GOLEM-NV hyperparameters for all experiments.** Final settings for the sparsity penalty coefficient $\lambda_1$, acyclicity penalty coefficient $\lambda_2$ and the weight threshold $\eta$ after hyperparameter tuning on the respective models and data-generating processes together with the F1 (median) validation scores achieved by GOLEM-NV.

(a) ER$(20, 2)$ DAGs, LINEAR mechanisms

| Weight Distribution | Model | $\lambda_1$ | $\lambda_2$ | $\eta$ | F1 (median) |
|---|---|---|---|---|---|
| Unif$_\pm [0.5, 2.0]$ | SCM | 0.0002 | 2.00 | 0.20 | 0.16 |
| Unif$_\pm [0.5, 2.0]$ | Standardized SCM | 0.0200 | 2.00 | 0.10 | 0.38 |
| Unif$_\pm [0.5, 2.0]$ | iSCM | 0.0200 | 2.00 | 0.20 | 0.45 |
| Unif$_\pm [1.3, 3.0]$ | SCM | 0.0002 | 2.00 | 0.10 | 0.20 |
| Unif$_\pm [1.3, 3.0]$ | Standardized SCM | 0.0002 | 5.00 | 0.10 | 0.37 |
| Unif$_\pm [1.3, 3.0]$ | iSCM | 0.0200 | 2.00 | 0.20 | 0.37 |
| Unif$_\pm [0.3, 0.8]$ | SCM | 0.0020 | 5.00 | 0.20 | 0.13 |
| Unif$_\pm [0.3, 0.8]$ | Standardized SCM | 0.0200 | 2.00 | 0.20 | 0.55 |
| Unif$_\pm [0.3, 0.8]$ | iSCM | 0.0200 | 2.00 | 0.10 | 0.58 |

(b) ER$(100, 2)$ DAGs, LINEAR mechanisms

| Weight Distribution | Model | $\lambda_1$ | $\lambda_2$ | $\eta$ | F1 (median) |
|---|---|---|---|---|---|
| Unif$_\pm [0.5, 2.0]$ | SCM | 0.002 | 5.00 | 0.20 | 0.10 |
| Unif$_\pm [0.5, 2.0]$ | Standardized SCM | 0.020 | 2.00 | 0.10 | 0.32 |
| Unif$_\pm [0.5, 2.0]$ | iSCM | 0.020 | 2.00 | 0.10 | 0.51 |
| Unif$_\pm [1.3, 3.0]$ | SCM | 0.002 | 2.00 | 0.10 | 0.21 |
| Unif$_\pm [1.3, 3.0]$ | Standardized SCM | 0.002 | 5.00 | 0.10 | 0.18 |
| Unif$_\pm [1.3, 3.0]$ | iSCM | 0.020 | 2.00 | 0.10 | 0.46 |
| Unif$_\pm [0.3, 0.8]$ | SCM | 0.020 | 2.00 | 0.10 | 0.18 |
| Unif$_\pm [0.3, 0.8]$ | Standardized SCM | 0.020 | 2.00 | 0.20 | 0.65 |
| Unif$_\pm [0.3, 0.8]$ | iSCM | 0.020 | 2.00 | 0.10 | 0.67 |

(c) ER$(20, 2)$ DAGs, NONLINEAR mechanisms

| Model | $\lambda_1$ | $\lambda_2$ | $\eta$ | F1 (median) |
|---|---|---|---|---|
| SCM | 0.002 | 8.00 | 0.20 | 0.07 |
| Standardized SCM | 0.020 | 2.00 | 0.20 | 0.30 |
| iSCM | 0.020 | 2.00 | 0.10 | 0.41 |

(d) ER$(100, 2)$ DAGs, NONLINEAR mechanisms

| Model | $\lambda_1$ | $\lambda_2$ | $\eta$ | F1 (median) |
|---|---|---|---|---|
| SCM | 0.002 | 5.00 | 0.20 | 0.07 |
| Standardized SCM | 0.020 | 2.00 | 0.10 | 0.24 |
| iSCM | 0.020 | 2.00 | 0.10 | 0.36 |

Table 4: **PC hyperparameters for all experiments.** Final settings for the significance level $\alpha$ after hyperparameter tuning on the respective models and data-generating processes together with the F1 (median) validation scores achieved by PC.

(a) $\mathrm{ER}(20, 2)$ DAGs, LINEAR mechanisms

| Weight Distribution | Model | $\alpha$ | F1 (median) |
|---|---|---|---|
| $\mathrm{Unif}_{\pm}[0.3, 0.8]$ | SCM | 0.01 | 0.71 |
| $\mathrm{Unif}_{\pm}[0.3, 0.8]$ | Standardized SCM | 0.01 | 0.70 |
| $\mathrm{Unif}_{\pm}[0.3, 0.8]$ | iSCM | 0.01 | 0.72 |
| $\mathrm{Unif}_{\pm}[0.5, 2.0]$ | SCM | 0.01 | 0.47 |
| $\mathrm{Unif}_{\pm}[0.5, 2.0]$ | Standardized SCM | 0.01 | 0.46 |
| $\mathrm{Unif}_{\pm}[0.5, 2.0]$ | iSCM | 0.01 | 0.58 |
| $\mathrm{Unif}_{\pm}[1.3, 3.0]$ | SCM | 0.01 | 0.35 |
| $\mathrm{Unif}_{\pm}[1.3, 3.0]$ | Standardized SCM | 0.01 | 0.38 |
| $\mathrm{Unif}_{\pm}[1.3, 3.0]$ | iSCM | 0.01 | 0.48 |

(b) $\mathrm{ER}(100, 2)$ DAGs, LINEAR mechanisms

| Weight Distribution | Model | $\alpha$ | F1 (median) |
|---|---|---|---|
| $\mathrm{Unif}_{\pm}[0.3, 0.8]$ | SCM | 0.01 | 0.82 |
| $\mathrm{Unif}_{\pm}[0.3, 0.8]$ | Standardized SCM | 0.01 | 0.85 |
| $\mathrm{Unif}_{\pm}[0.3, 0.8]$ | iSCM | 0.01 | 0.86 |
| $\mathrm{Unif}_{\pm}[0.5, 2.0]$ | SCM | 0.01 | 0.62 |
| $\mathrm{Unif}_{\pm}[0.5, 2.0]$ | Standardized SCM | 0.01 | 0.57 |
| $\mathrm{Unif}_{\pm}[0.5, 2.0]$ | iSCM | 0.01 | 0.79 |
| $\mathrm{Unif}_{\pm}[1.3, 3.0]$ | SCM | 0.01 | 0.42 |
| $\mathrm{Unif}_{\pm}[1.3, 3.0]$ | Standardized SCM | 0.01 | 0.43 |
| $\mathrm{Unif}_{\pm}[1.3, 3.0]$ | iSCM | 0.01 | 0.71 |

(c) $\mathrm{ER}(20, 2)$ DAGs, NONLINEAR mechanisms

| Model | $\alpha$ | F1 (median) |
|---|---|---|
| SCM | 0.01 | 0.53 |
| Standardized SCM | 0.01 | 0.54 |
| iSCM | 0.01 | 0.65 |

(d) $\mathrm{ER}(100, 2)$ DAGs, NONLINEAR mechanisms

| Model | $\alpha$ | F1 (median) |
|---|---|---|
| SCM | 0.01 | 0.53 |
| Standardized SCM | 0.01 | 0.63 |
| iSCM | 0.01 | 0.68 |

Table 5: **CAM hyperparameters for all experiments.** Final settings for the cutoff value $\alpha$ for variable selection with hypothesis testing over regression coefficients, the number and order of splines to use for the feature function, together with the F1 (median) validation scores achieved by CAM.

(a) ER$(20, 2)$ DAGs, LINEAR mechanisms

| Weight Distribution | Model | $\alpha$ | Number of Splines | Spline Order | F1 (median) |
|---|---|---|---|---|---|
| Unif$_\pm [0.3, 0.8]$ | SCM | 0.05 | 5 | 3 | 0.49 |
| Unif$_\pm [0.3, 0.8]$ | Stand. SCM | 0.05 | 5 | 2 | 0.46 |
| Unif$_\pm [0.3, 0.8]$ | iSCM | 0.05 | 5 | 3 | 0.57 |
| Unif$_\pm [0.5, 2.0]$ | SCM | 0.10 | 10 | 3 | 0.31 |
| Unif$_\pm [0.5, 2.0]$ | Stand. SCM | 0.10 | 10 | 2 | 0.23 |
| Unif$_\pm [0.5, 2.0]$ | iSCM | 0.10 | 5 | 2 | 0.53 |
| Unif$_\pm [1.3, 3.0]$ | SCM | 0.05 | 10 | 2 | 0.24 |
| Unif$_\pm [1.3, 3.0]$ | Stand. SCM | 0.05 | 10 | 3 | 0.27 |
| Unif$_\pm [1.3, 3.0]$ | iSCM | 0.05 | 5 | 2 | 0.42 |

(b) ER$(100, 2)$ DAGs, LINEAR mechanisms

| Weight Distribution | Model | $\alpha$ | Number of Splines | Spline Order | F1 (median) |
|---|---|---|---|---|---|
| Unif$_\pm [0.3, 0.8]$ | SCM | 0.05 | 10 | 3 | 0.54 |
| Unif$_\pm [0.3, 0.8]$ | Stand. SCM | 0.05 | 10 | 2 | 0.57 |
| Unif$_\pm [0.3, 0.8]$ | iSCM | 0.05 | 5 | 3 | 0.61 |
| Unif$_\pm [0.5, 2.0]$ | SCM | 0.05 | 10 | 2 | 0.39 |
| Unif$_\pm [0.5, 2.0]$ | Stand. SCM | 0.05 | 5 | 2 | 0.39 |
| Unif$_\pm [0.5, 2.0]$ | iSCM | 0.05 | 5 | 3 | 0.62 |
| Unif$_\pm [1.3, 3.0]$ | SCM | 0.05 | 5 | 3 | 0.27 |
| Unif$_\pm [1.3, 3.0]$ | Stand. SCM | 0.05 | 10 | 3 | 0.26 |
| Unif$_\pm [1.3, 3.0]$ | iSCM | 0.05 | 5 | 3 | 0.61 |

(c) ER$(20, 2)$ DAGs, NONLINEAR mechanisms

| Model | $\alpha$ | Number of Splines | Spline Order | F1 (median) |
|---|---|---|---|---|
| SCM | 0.05 | 10 | 2 | 0.50 |
| Standardized SCM | 0.05 | 5 | 2 | 0.52 |
| iSCM | 0.05 | 5 | 2 | 0.57 |

(d) ER$(100, 2)$ DAGs, NONLINEAR mechanisms

| Model | $\alpha$ | Number of Splines | Spline Order | F1 (median) |
|---|---|---|---|---|
| SCM | 0.05 | 10 | 3 | 0.50 |
| Standardized SCM | 0.05 | 10 | 2 | 0.51 |
| iSCM | 0.05 | 10 | 3 | 0.57 |

it holds that

$$
\begin{aligned}
\mathrm{Var}[x_i^t] &= \mathrm{Var}\left[ \sqrt{\frac{\mathrm{Var}[x_i^a] - \sigma_i^{2^b}}{\mathrm{Var}[\mathbf{w}_i^{aT}\mathbf{x}_{\mathrm{pa}(i)}^t]}} \, \mathbf{w}_i^{aT}\mathbf{x}_{\mathrm{pa}(i)}^t + \varepsilon_i^t \right] \\
&= \frac{\mathrm{Var}[x_i^a] - \sigma_i^{2^b}}{\mathrm{Var}[\mathbf{w}_i^{aT}\mathbf{x}_{\mathrm{pa}(i)}^t]} \, \mathrm{Var}[\mathbf{w}_i^{aT}\mathbf{x}_{\mathrm{pa}(i)}^t] + \sigma_i^{2^b} \\
&= \mathrm{Var}[x_i^a] \, ,
\end{aligned}
$$

which satisfies condition 1. Since the systems $S^t$ and $S^a$ have the same marginal variances, they have the same $\mathrm{Var}$-sortability. In the noise transfer experiment of Figure 6, we transfer the noise variances from the implied models of iSCMs and standardized SCMs. To obtain the noise variances in the implied models, we divide the original noise variances (equal to 1) by the estimated marginal variances of the corresponding variable before standardization, which we estimate from $n = 1000$ datapoints. For iSCM, this corresponds to an empirical statistics of Equation (7).

### E.6 COMPUTE RESOURCES

Our experiments were run on an internal cluster. All experiments in this work were computed using CPUs with 3GB of memory per CPU, with an exception of the AVICI runs on graphs with 100 vertices, which used 12GB per CPU. The data generation takes less than a few minutes on a single CPU, with the exception of the sortability results (Section 5.1). For the sortability results, it takes around 30 minutes to generate the datasets for a single graph specification across all weight supports and graph sizes. This is due to a bigger number of configurations and repetitions than in the other experiments. For a single graph specification and across all weight supports and graph sizes, it takes around 6 hours to compute the sortability statistics on a single CPU. All benchmarked methods take no longer than a few minutes per small graph ($d = 20$) and no longer than half an hour per big graph ($d = 100$). The SORTNREGRESS baselines run in less than 1min per graph.

## F ADDITIONAL EXPERIMENTAL RESULTS

### F.1 STRUCTURE LEARNING

Figure 14 summarizes the structural Hamming distance (SHD) between the predicted and true graphs for the same datasets and algorithms as in Figure 5.

In Figures 15a and 15b, we present the F1 scores and SHD attained by the structure learning algorithms on data of LINEAR iSCMs, SCMs, and standardized SCMs, across different weight distribution supports and graph sizes. We find that the difference in performance of NOTEARS on data sampled from iSCM and standardized SCMs is larger for larger weight magnitudes and for bigger graphs. For smaller weights, the difference in the mean F1 score of NOTEARS between the two standardization approaches is smaller, which is in line with our proposed explanation about the shifts of the implied noise variance distribution in Section 5.2.

In Figure 15a, we also find that when weight magnitudes are below 1, $R^2$-SORTNREGRESS performs similarly for both standardized SCMs and iSCMs. We also observe this for AVICI. Meanwhile, for larger weights with support extending above 1, these algorithms achieve significantly higher F1 scores on standardized SCMs. This suggests that our condition of $|w_{i,j}| > 1$ for all edges $(v_i, v_j)$ in the statement of Theorem 3, concerning the identifiability of linear standardized SCMs, may have a more fundamental practical significance, rather than being merely an artifact of the analysis.

In Figure 18, we report results for when the additive noise in the ground-truth SCMs is non-Gaussian. In this setting, the causal graphs of SCMs are identifiable from observational data (see Section 2). Here, we also benchmark LINGAM (Shimizu et al., 2006), which is designed for this setting. While LINGAM performs very well as expected, it performs significantly worse on standardized SCMs, possibly because independent component analysis suffers in practice under the very low noise scales implied by post-hoc standardization. This would be in line with our discussion in Section 5.2.

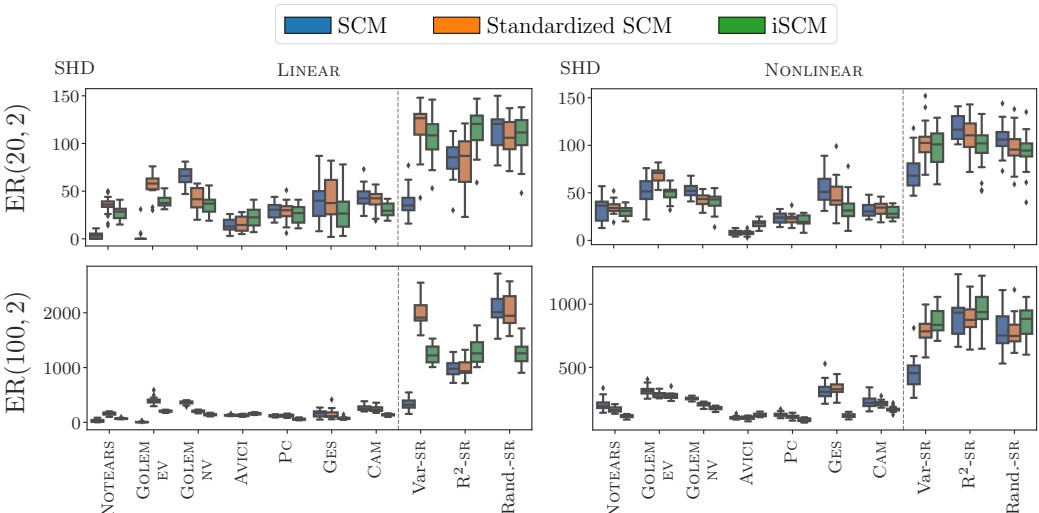

Figure 14: **SHD to the true causal graph for LINEAR and NONLINEAR mechanisms.** Box plots show median and interquartile range (IQR). Whiskers extend to the largest value inside $1.5\times$IQR from the boxes. Left (right) column shows results for linear (nonlinear) causal mechanisms with additive noise $\varepsilon_i \sim \mathcal{N}(0, 1)$. LINEAR mechanisms have weights $w_{i,j} \sim \mathrm{Unif}_{\pm}\,[0.5, 2.0]$.

## F.2 $R^2$-SORTABILITY

Figure 16 reports the $R^2$-sortability statistics across varying graph sizes and weight distributions but for the denser graphs $ER(d, 4)$ and $SF(d, 4)$. We again observe $R^2$-sortability very close to $0.5$ for datasets sampled from iSCM and high degrees of $R^2$-sortability for data drawn from standardized SCMs. Moreover, in Figure 17, we show the $R^2$-sortability for varying expected node degrees in the graph. Data sampled from iSCMs remains close to not $R^2$-sortable for denser graphs drawn from the graph families considered here. We omit standard SCMs from the plots as the datasets of SCMs and their standardized versions have the same $R^2$-sortability, since the $R^2$ coefficient is scale invariant.

## F.3 IMPLIED NOISE SCALES

Figure 19 shows the inverse implied noise scales of standardized SCMs and iSCMs for linear models with smaller weights magnitudes than in Figure 6 of the main text. In this setting with smaller weights, the distributions of the implied noise scales of standardized SCMs and iSCMs show significantly greater overlap than in Figure 6. Since the weights are smaller, the effect of the exploding marginal variances and thus collapsing implied noise scales is weaker in the SCMs.

In Figure 15 (left), we evaluate the algorithms considered in Section 5 on these systems with smaller weights. We see that, in this setting, NOTEARS performs very similarly on standardized SCMs and iSCMs, with NOTEARS slightly outperforming on iSCMs for bigger graphs, since we do not remove the growing variance problem completely even for weights of small magnitude. This is inline with our reasoning in Section 5.2.

## F.4 COVARIANCE MATRICES FOR FIGURE 1

Figure 20 visualizes the full mean absolute covariance (correlation) matrices of the systems presented in Figure 1. The matrix shows that the pattern of increasing mean absolute covariance in standardized SCMs is not only a feature of neighboring nodes, but it also occurs for vertex pairs further apart, though less strongly. This is not the case for iSCMs, where any two pairs of equally spaced vertices have equal covariances in expectation over the weight sampling distribution.

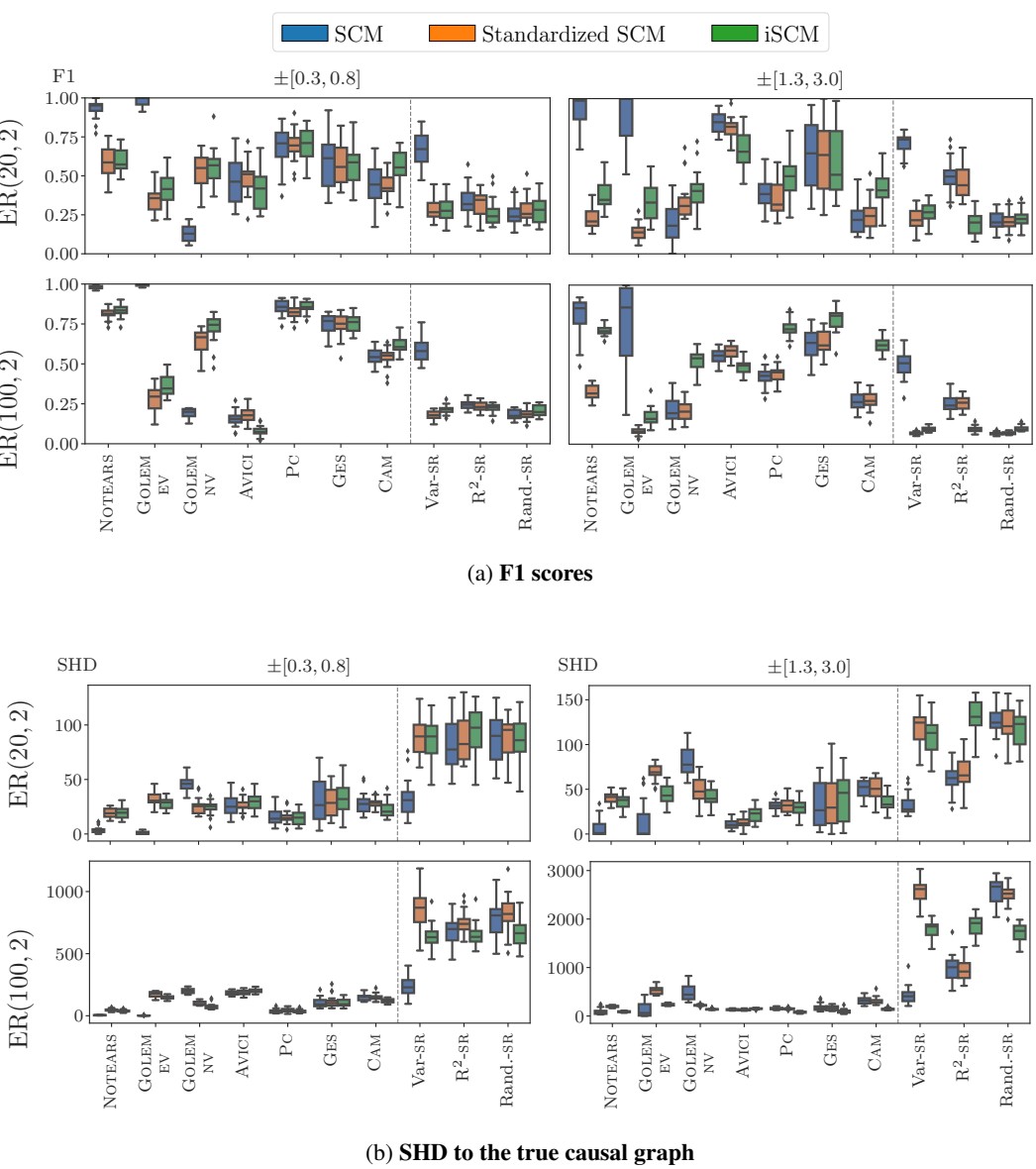

Figure 15: **Structure learning results for different LINEAR weight ranges.** Results for LINEAR causal mechanisms with additive noise $\varepsilon_i \sim \mathcal{N}(0, 1)$ and weights sampled uniformly from support indicated above each column. Box plots show median and interquartile range (IQR). Whiskers extend to the largest value inside $1.5 \times$IQR from the boxes. For every model, we sample 20 systems and $n = 1000$ data points each.

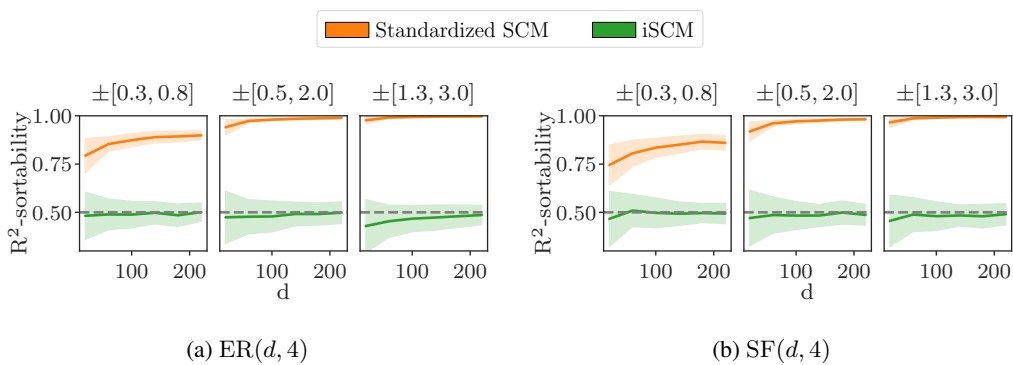

Figure 16: $R^2$-**sortability for different graph sizes.** Linear standardized SCMs and iSCMs with $\varepsilon_i \sim \mathcal{N}(0, 1)$ and weights drawn from uniform distributions with supports given above each plot. For every model, we sample 100 systems and $n = 1000$ data points each. Lines and shaded regions denote mean and standard deviation of $R^2$-sortability across runs. Datasets that satisfy $R^2$-sortability $= 0.5$ (dashed) are not $R^2$-sortable.

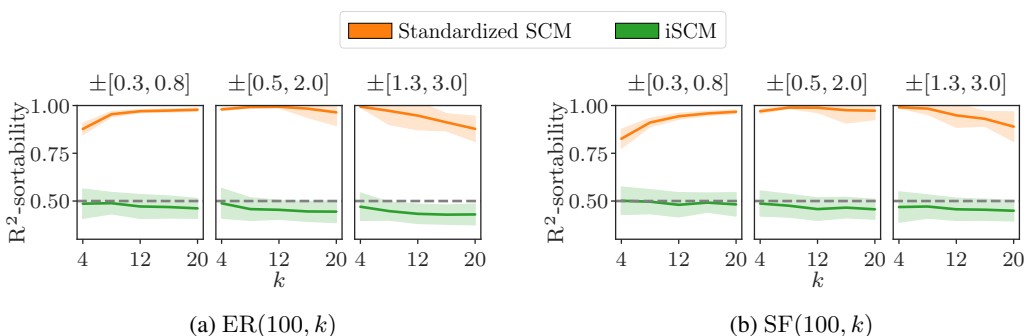

Figure 17: $R^2$-**sortability for different (expected) node degrees.** Linear standardized SCMs and iSCMs with $\varepsilon_i \sim \mathcal{N}(0, 1)$ and weights drawn from uniform distributions with supports given above each plot. For every model, we evaluate 100 systems and $n = 1000$ samples each. Lines and shaded regions denote mean and standard deviation. Datasets that satisfy $R^2$-sortability $= 0.5$ (dashed) are not $R^2$-sortable.

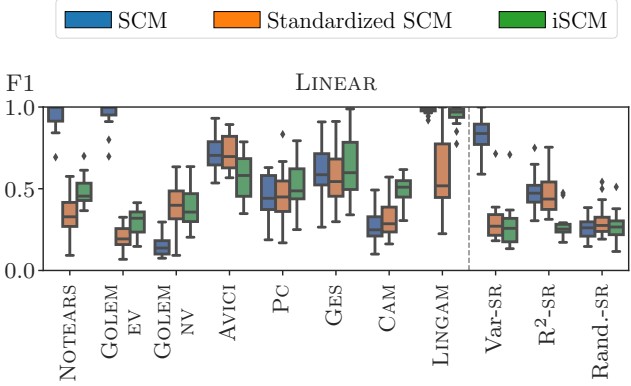

Figure 18: **Structure learning results for *non-Gaussian* noise distributions.** Causal mechanisms have additive noise $\varepsilon_i \sim \text{Unif}\,[-\sqrt{3}, \sqrt{3}]$, which induces $\text{Var}[\varepsilon_i] = 1$, and LINEAR mechanisms with weights $w_{i,j} \sim \text{Unif}_\pm [0.5, 2.0]$. Graphs are sampled from $\text{ER}(20, 2)$. To obtain the results, we use the same hyperparameters as the ones we used to obtain the top-left panel of Figure 5. Box plots show median and interquartile range (IQR). Whiskers extend to the largest value inside $1.5 \times$IQR from the boxes.

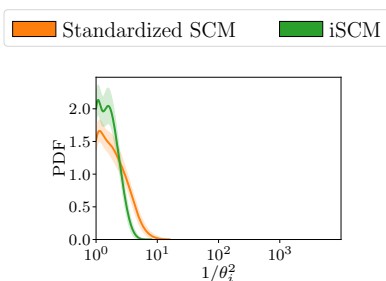

Figure 19: **Distribution over inverse implied noise scales in the implied SCMs** for $\mathrm{ER}(100, 2)$ graphs with smaller weights $w_{i,j} \sim \pm \mathrm{Unif}_{[0.3,0.8]}$, estimated with kernel density estimation. Lines and shading denote mean and standard deviation respectively.

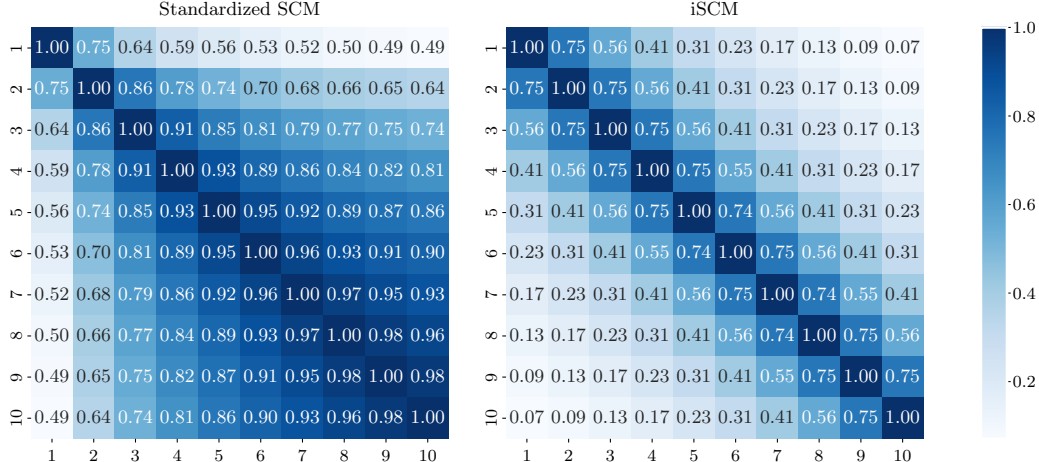

Figure 20: **Mean absolute covariance matrices for models in Figure 1.** Linear standardized SCMs (left) and iSCMs (right) with 10-variable chain DAGs from $x_1$ to $x_{10}$ and weights $w_{i,j} \sim \mathrm{Unif}_{\pm}[0.5, 2.0]$ and additive noise from $\mathcal{N}(0, 1)$. Mean covariances are estimated from $n = 100{,}000$ datapoints and averaged over $100{,}000$ models. Since both models have unit marginal variances, covariance equals correlation.

