# OpenReview forum: "Standardizing Structural Causal Models"
_ICLR.cc/2025/Conference — ICLR 2025 Poster_

### Official Review · Reviewer_QXbX · 2024-10-25

**Soundness:** 2
**Presentation:** 3
**Contribution:** 2
**Rating:** 8
**Confidence:** 4

**Summary:**

This paper addresses the well-known problem of the presence of shortcuts in structural causal models commonly simulated for the benchmarking of causal discovery algorithms. The authors propose iSCM, a variation of standardized structural causal models where standardization occurs at each step of the generation along the topological order, instead of standardizing the dataset post hoc, when all variables are already sampled. They provide some empirical evidence and theory showing that this removes the dependence of the Cause-Explained-Variance (one source of shortcuts in SCMs, standardized or not) from the depth of the causal graph. Additionally, they carry out experiments to compare several algorithms’ performance over SCMs, Standardized SCMs, and iSCM.

**Strengths:**

The paper proposes a practical solution to remove varsortability and R2-sortability in samples from simulated structural causal models. The available empirical evidence suggests that their approach succeeds in removing these shortcuts, and the provided theory (Theorem 2) supports these observations (at least in the linear case). In general, the paper is well presented and has the potential of a valuable contribution to the causal discovery community.

**Weaknesses:**

1. This paper stresses that iSCMs are preferable to SCM for benchmarking, or generally that SCMs are ill-suited for the purpose (L80-82, L49-52, L187-188). I believe this is not supported by the available empirical evidence, both in the paper and in the literature: methods like e.g. LiNGAM, RESIT, CAM, and SCORE (and all other methods based on score matching that are state of the art) have been shown by extensive benchmarking to be neutral about rescaling (see Montagna et al., 2023). The bottom line is that whether SCMs are ill-suited depends on the algorithms at hand: in general, such claims should be supported by more extensive empirical evidence, as I discuss in the next point.
2. The empirical section is limited. One reason is the aforementioned lack of well-known / SOTA methodologies that might be unaffected by standardization. Moreover, the authors do not consider the effect of increasing edge density. Given that the authors’ aim is to propose new SCMs for benchmarking, and the common practice of testing at different density and sparsity levels, I believe these experiments should be added to the paper. To control density on ER graphs in a way that reasonably scales with the number of nodes, I suggest fixing the probability of edge rather than the average degree (as the latter does not scale with graph size). Finally, the experiments of Fig. 5 are carried out on datasets that are not R2-sortable in the first place. The only clear evidence that iSCM removes R2-sortability is provided in Figure 4, but this is still limited, at least for the lack of testing on different density levels, which, as I argued before, is important.
3. The claim that the existing identifiability results only concern the *unstandardized* distributions of SCMs (L147) is not true, or at least I don’t see that: let aside the case where the assumption enabling identifiability is that of equal variance of the noise terms for each variable - which are of minor importance, as they concern the very specific case of LiNGAM models with equal variance of noises: LiNGAM, ANM, and PNL identifiability results are agnostic about the variance. So any identifiability result about SCMs generating according to the aforementioned model classes, would hold both for standardized SCMs and for iSCMs.

I am willing to increase my score given improved experiments and removal/explanation of somewhat controversial claims as those in the first and third points of the weaknesses section.

**Questions:**

1. Theorem 2 does show that there is no convergence of CEV towards 1 when we increase the depth, still, it displays a dependence on the density: this is completely overlooked by the authors, both in the theory and the experiments. Is there a specific reason? As described in the weaknesses section, I would like to see experiments accounting for different density levels. In practice, I expect to see that nodes with a lot of parents have a larger CEV, such that when directing edges, it is more likely that that node with a higher CEV is a child rather than a parent: under this conjecture, R2 is still a signal about the correct topological order which could be exploited (although weaker than the one provided by SCMs). SF graphs with preferential attachment might be a good testing ground to see this effect, as they produce an unbalance in the number of edges between different nodes.
2. I believe that there is a big point where iSCMs would be beneficial and that is a bit overlooked. One important use of synthetic data is for training amortized inference models such as AVICI. It is reasonable to think that these models exploit shortcuts as R2 or var-sortability. Yet, this is not really studied in the paper: all the experiments of Figure 5 are performed on SCMs that are non-R2 sortable in the first place, so we don’t really understand whether AVICI trained on standardized data benefits of R2 sortability or not.

---

> ### Author Response · Authors · 2024-11-19
> **Response (1/2) to Reviewer QXbX**
>
> Thank you for your feedback and comments. We appreciate your pointers, especially concerning settings like LiNGAM, and agree that we overlooked these in our initial discussion of the theoretical and empirical results. In our updated PDF, we now acknowledge these appropriately, corrected the corresponding claims, and added additional benchmarking results you suggested (see paragraph starting on line 144). Below, we reply in more detail to your individual comments.
> Please get back to us if any of your concerns are not sufficiently addressed. We would be happy to elaborate further.
>
> > methods like e.g. LiNGAM [...]  have been shown by extensive benchmarking to be neutral about rescaling. [...] whether SCMs are ill-suited depends on the algorithms at hand.
>
> We fully agree here in principle, but would like to elaborate on the last conclusion. We argue that SCMs alone are not ideal for benchmarking, precisely because they give algorithms that are not neutral about covariance artifacts an unexpected advantage. Since we may not know *a priori* which method is sensitive to these patterns, SCMs by themselves are, in some sense, not ideal for benchmarking *irrespective of* the algorithms at hand, because comparisons across new algorithms can lead to wrong conclusions. (The community would probably agree that this happened when benchmarking NOTEARS). A posteriori, though, we agree that it depends on the algorithm whether SCMs are ill-suited for benchmarking. As we discuss in our conclusion, iSCMs should be viewed as an additional tool for disentangling, together with other models like SCMs, how the performance of a method depends on covariance artifacts (see lines 514-516). To clarify this takeaway, and the use case of iSCMs *alongside* SCMs, we added additional sentences to the introduction (lines 079-080) and the results (lines 362-364) in our revised PDF.
>
>
> > the empirical section is limited [due to lacking] methodologies that might be unaffected by standardization [...] and different [graph] density and sparsity levels
>
> This is not correct, since our experiments partially cover both of these aspects. GOLEM-NV (evaluated in Sec. 5) learns the noise scale and thus does not depend on the (implied) noise scale, as we discuss in line 430. However, we agree that adding CAM as another evaluated method is informative, since the algorithm both models nonlinearity and estimates the noise scales and thus remains invariant to changing implied noise scales. Since LiNGAM is designed for linear models with non-Gaussian noise, and we focus on models with Gaussian noise, we do not benchmark LiNGAM here. We added the additional results for CAM to Figure 5 (and Figure 17) in our updated version of the manuscript. (For this rebuttal, we used the default hyperparameters of the ‘dodiscover’ package. We will run an extensive hyperparameter search for CAM, as it was run for all methods, for the camera-ready version.)
>
> We already report R2-sortability results for high-degree graphs (4 instead of 2) in Figure 16.
> Nevertheless, following your suggestion, we complete the discussion on how sparsity affects R2-sortability by additionally evaluating R2-sortability at even larger, varying (expected) node degree and added the results as Figure 17 to the updated manuscript. The results show that in even denser graphs (up to degree 20), iSCMs remain essentially not sortable by R2.
> We do not necessarily agree that the density of ER graphs “reasonably scales [when fixing] the probability of edge rather than the average degree”. Sparse graphs, as commonly assumed in causal discovery, have a degree distribution that is constant with respect to the graph size (i.e., not O(d), which would occur when fixing the edge probability). We follow prior work here (e.g., Reisach et al. 2021, 2024).

---

> ### Author Response · Authors · 2024-11-19
> **Response (2/2) to Reviewer QXbX**
>
> > Theorem 2 displays a dependence on the density: this is completely overlooked [...] I expect to see that nodes with a lot of parents have a larger CEV, such that when directing edges, it is more likely that that node with a higher CEV is a child rather than a parent: under this conjecture, R2 is still a signal about the correct topological order
>
> We are aware that degree influences CEV, but the discussion of how this influences R2 sortability is subtle. First, it is worth emphasizing how CEV and R2-sortability are not the same: it is not obvious how R2 (regression on all nodes) and CEV (regression on true parents) relate in general. CEV is a property of a variable, and R2-sortability of the whole system. Our view is that identifying the *global* positioning in a causal ordering from *local* patterns constitutes an artifact (like R2), but identifying *local* structure like edge directions not (like CEV)---otherwise, structure learning would be impossible.  In SCMs, nodes lower down tend to have higher CEV when randomly sampling the linear weights from the same distribution, even if we condition on the number of parents. This is not true in iSCMs.
>
> We also explicitly do *not* claim or aim to make local edge directions in iSCMs unorientable by CEV. In fact, we do not believe that causal models should have constant CEV across variables, since we have no reason to believe that CEV should not vary across variables, even based on the number of direct causes. (In Appendix D.1 on existing heuristics, we argue that models forcing CEV = 0.5 are limited.) In general, we also do not claim that iSCMs *always* remove R2-sortability either (e.g. in very dense graphs). We write that “Our results do not exclude the possibility of iSCM configurations that still produce R2 -sortable datasets. However, we show empirically that, for commonly-used [configurations], iSCM datasets are not R2 -sortable with high probability” (lines 398-400).
>
>
> > The claim that the existing identifiability results only concern the unstandardized distributions of SCMs (L147) is not true [...] LiNGAM, ANM, and PNL identifiability results are agnostic about the variance.
>
> Thank you for this pointer. We completely agree with this comment and apologize for having missed this initially. In our uploaded and revised PDF, we corrected this wrong claim and acknowledged that identifiability results agnostic about the noise scale still apply (see paragraph starting on line 144).
>
> > It is reasonable to think that [AVICI] exploits shortcuts such as R2 or var-sortability [...] all the experiments of Figure 5 are performed on SCMs that are non-R2 sortable [...] so we don’t really understand whether AVICI trained on standardized data benefits of R2 sortability or not
>
> There is a major misunderstanding here: the experiments of Figure 5 were run on SCMs that *are* R2-sortable. All SCMs and standardized SCMs (blue and orange) are R2-sortable, as shown in Figure 4. Both Figures use the linear standardized SCMs and iSCMs sampled according to the same graph, weight, and noise distribution. Exactly this comparison to iSCMs, which are not R2-sortable in Figure 4, is the main motivation of our experiments. Precisely the reason you state is also why we benchmark AVICI: “AVICI [... is] optimized to exploit any artifacts that improve predictive accuracy. To investigate its susceptibility to artifacts, we evaluate the public model checkpoints trained on standardized SCMs” (lines 416-418). Similarly, we conclude: “AVICI shows the same trend [of performing worse on iSCMs], suggesting it may indeed be exploiting the correlation artifacts present in its training distribution.” (lines 423-424)
>
> We thus could not agree more with your comments here. It is unfortunate that it appears we “overlooked” this. If there is any way we can make this more clear to readers in the paper, we would be happy to implement your suggestions.

---

> > ### Comment · Reviewer_QXbX · 2024-11-22
> >
> > I thank the authors for considering my suggestions and incorporating them in the paper, and for the thorough response to my concerns. I still have one point to discuss and one minor note.
> >
> > 1. You say that in the experiment you *focus on models with Gaussian noise*. Any specific reason to avoid linear non-Gaussian? These are *very* common datasets that are used for benchmarking, as the LiNGAM assumptions underlie a lot of methods in the literature.
> > 2. This is minor: you say that in general you don't claim that iSCMs always remove R2-sortability. Yet, in the abstract and introduction I got this impression.
> >
> >     > L17: By construction, iSCMs are not Var-sortable, and as we show experimentally, not R2 sortable either
> >     >
> >
> >     From this, I get the message that by construction iSCMs are not var-sortable and not R2 sortable either. Better disentangling the sentence between what you show for Var-sortability and what you show for R2 sortability would help. Something like "iSCMs are not var-sortable by construction. Empirically, we additionally show that they are not R2 sortable either" (this is not a wording suggestion, but just to give an idea of what I mean by *disentangle the sentence*).

---

> > > ### Author Response · Authors · 2024-11-25
> > >
> > > Thanks for your reply.
> > > 1. Since the non-identifiability result for iSCMs relies on the Gaussianity assumption, we wanted to keep the experimental settings similar and used Gaussian noise (like, e.g., Reisach et al. (2024)). As shown by our theoretical results, iSCMs address a problem arising primarily from the functional mechanism (weight ranges) rather than the type of noise distribution, hence we focus on benchmarking under different functional mechanisms.
> > >
> > >     However, to facilitate interpreting the effects of iSCMs, we added a set of additional results on *non-Gaussian* systems to the paper, including LiNGAM and all other previously considered methods. Please refer to Figure 18 and Appendix F. The results are very closely aligned with what we observe for systems with Gaussian noise.
> > >
> > > 2. Thanks for this pointer, we did not intend to convey this claim. We changed our wording in the abstract (line 018).
> > >
> > >
> > > Since you stated that you would be willing to increase your score in light of extended experiments and edits to the paper, we hope that our tweaks and added results now sufficiently reflect this and address your concerns. Please feel free to follow-up again should this not be the case.

---

> > > > ### Comment · Reviewer_QXbX · 2024-11-25
> > > >
> > > > Thank you, I raised my score accordingly.

---

### Official Review · Reviewer_R8Hm · 2024-11-02

**Soundness:** 2
**Presentation:** 2
**Contribution:** 1
**Rating:** 5
**Confidence:** 4

**Summary:**

The authors observe that while SCMs are widely assumed to be the underlying causal model in causal structure learning algorithms, variance and pairwise correlations among the observed variables in an SCM may provide extra information about the underlying model. This information may not be removed by simply standardizing the observed variables and may be used by existing structure learning algorithms. Therefore, they propose internally-standardized SCMs, where the standardization is done within the recursive data-generating process. The authors provide partial identification of the proposed model and also test the performance of existing algorithms on their proposed model.

**Strengths:**

1. The observation about the variance and pairwise correlation of variables in SCMs is very interesting and has not been widely studied. This idea could be used for further evaluation of the performance of structure learning algorithms, especially when no theoretical identifiability results are provided.
2. The organization of the paper is clear and easy to follow.

**Weaknesses:**

1. Lack of empirical evidence for the advantage of the proposed model. The authors mention that SCM may suffer from the problem of increasing correlation, whereas iSCM does not. However, there is no clear evidence that this is actually the case in real-life scenarios. If iSCM offers no advantage over SCM in real-life scenarios (i.e., increasing correlation may actually occur in real data), then this may not be a meaningful benchmark, as iSCM is a submodel of SCM and has more model assumptions than SCM.

2. Limited novelty of the proposed theoretical result. The authors claim to provide the first identifiability result for standardized SCMs. However, since standardized SCMs can be considered a special case of SCM, existing identifiability results for SCMs that do not rely on variances (such as LiNGAM) should also apply here (see Q1 below). Additionally, Theorem 3 states that if the underlying model (1) is a linear SCM, (2) has equal noise variance, (3) has a tree structure, and (4) has all edge weights with absolute values greater than 1, then these conditions are sufficient to identify the causal model from standardized data given the MEC. This is a highly restrictive setting, particularly conditions (3) and (4), and there may be other identification methods under the same setting that do not rely on pairwise correlation properties.

**Questions:**

1. Line 147: "existing identifiability results only concern the unstandardized distributions $p(x)$ of SCMs." I am not sure if I agree with this claim, as most causal discovery methods either do not rely on any parametric assumptions (e.g., PC), or rely on assumptions that do not depend on variance (e.g., LiNGAM).

2. The chain structure in Figure 1 is a special case where, for each $x_i$, $i=1, \cdots, 10$, all variables $x_j$ with $j < i$ are ancestors of $x_i$. This structure leads to linear growth in variances, which in turn affects correlations. Would this be resolved if the underlying model had a sparser structure or a smaller tree depth?

3. Line 459: "in standardized SCMs, the parent-child relationships become more deterministic for larger graphs, which makes independence testing less reliable." I am not sure if I agree with this claim either, as the PC algorithm is mainly based on *conditional* independence tests, and it recursively removes edges from the fully connected graph only when the variables are conditionally independent. This means that marginal dependencies (deterministic relations) should not have a huge impact on the the algorithm.

---

> ### Author Response · Authors · 2024-11-19
> **Response (1/2) to Reviewer R8Hm**
>
> Thank you for your comments, feedback, and pointers. We very much appreciate your references to the identifiability settings we originally missed (e.g. LiNGAM). In our updated PDF, we now acknowledge these appropriately and have corrected the corresponding claims (see paragraph starting on line 144).  Below, we reply in more detail to your individual comments.
> Please get back to us if any of your concerns are not sufficiently addressed. We would be happy to elaborate further.
>
> > there is no clear evidence that [absence of R2 sortability] is actually the case in real-life scenarios. If iSCM offers no advantage over SCM in real-life scenarios (i.e., increasing correlation may actually occur in real data), then this may not be a meaningful benchmark, as iSCM [...] has more model assumptions than SCM.
>
> It is challenging to provide empirical evidence on the extent to which R2 sortability occurs in real-world data. However, there are several important points to keep in mind here.
>
> iSCMs are a useful benchmarking tool, irrespective of whether real-world datasets are R2 sortable. In our conclusion (lines 514-516), we argue that “ iSCMs [are] a useful complement to structure learning benchmarks with SCMs, enabling a specific evaluation of the ability of algorithms to transfer to real-world settings that do not exhibit R2 artifacts.” iSCMs are a tool that helps study *to what degree* R2-sortability and covariance artifacts drive the performance of a causal discovery, jointly with e.g. standardized SCMs, and this does not depend on the degree to which this occurs in applications. We do not want to argue that iSCMs should become the sole benchmarking tool. To clarify this point as much as possible, we added additional sentences to the introduction (lines 079-080) and the results (lines 362-364) in our revised PDF.
>
> Orthogonal to this, real-world datasets lack ground truth causal graphs, so R2-sortability cannot easily be studied empirically. Estimating R2-sortability in individual instances (e.g. the Sachs dataset, see Reisach, 2024) would not allow us to draw general conclusions without a risk of using anecdotal evidence. That being said, the mechanisms of iSCMs could be physically more realistic when reasoning from first principles. For instance, iSCMs model mechanisms that are unit-less (see lines 197-202), while SCMs are not. We also argue why they could be more realistic from a physical perspective in our conclusion (lines 531-535).
>
> We do not agree with the comment that iSCMs make more model assumptions than SCMs. In some sense, in the context of usual benchmarking assumptions, we would argue the opposite: since iSCMs do *not* induce more deterministic causal relationships downstream in the causal ordering a priori, iSCMs can be viewed as making less prior assumptions about a system. Since iSCMs are effectively a stable way to “reparameterize” standardized SCMs (see lines 535-536, also Section 3.2), they characterize the same space of models, but in a way that is more suitable for randomly generating synthetic instances without introducing artifacts. Appendix A is helpful for internalizing how both of the model classes are related and equal.
>
>
> > existing identifiability results for SCMs that do not rely on variances (such as LiNGAM) should also apply [to standardized SCMs]
>
> Thank you for this comment. We completely agree with this and apologize for having missed this initially. In our uploaded and revised PDF, we corrected any wrong claims around this and acknowledged that identifiability results agnostic about the noise scale still apply  (see paragraph starting on line 144).
>
> It is worth noting that our identifiability results (Theorems 3 and 4) are still novel. Results for LiNGAM are for non-Gaussian noise. We instead study the Gaussian setting under assumptions that are relevant for synthetic benchmarking. Unlike the setting of identifiability results for LiNGAM, ours conflict for standardized SCMs and iSCMs, providing an understanding of why iSCMs might be harder to identify than standardized SCMs in synthetic benchmarking setups. This insight is not apparent only based on past identifiability results.

---

> ### Author Response · Authors · 2024-11-19
> **Response (2/2) to Reviewer R8Hm**
>
> > Theorem 3 [has restrictive settings] [...] there may be other identification methods under the same setting that do not rely on pairwise correlation properties.
>
> It is true that Theorem 3 makes a specific set of assumptions, among others that the linear weights have magnitude greater than 1. Rather than being a restriction though, this condition makes our result interesting and relevant in practice, because it tells us that *the design of a synthetic data-generating process can influence identifiability* without this being the explicit intention of the scientist creating the benchmark. Our result is particularly relevant in the context of recent work, which often considers linear Gaussian settings with weights bounded away from 0 (lines 111-115). Identifiability based on weight magnitudes was, to our knowledge, not studied before, and our result helps interpret the varying empirical performance of NOTEARS. To our knowledge, our proof techniques for orienting edges based on correlation also make a novel contribution (see Figure 9, Section 4.2, and Appendix C.4)
>
> The second half of your comment is not wrong in that, yes, it is possible (but unknown to date) whether there exist alternative identification methods for our setting. However, this could be said about any existing identifiability result out there before the more general result is shown. Regardless, under the conditions of Theorem 3 at least, our result shows that you do not need anything more than to reason about pairwise correlations. Similarly, our result for Theorem 4 shows that *no algorithm* can identify more than one edge per connected component. This is intuitive when considering that in our setup, where all observations become zero-mean Gaussian distributions with unit variance, there is effectively no additional information beyond pairwise correlations with which to achieve identification.
>
> > The chain structure in Figure 1 [...] leads to linear growth in variances, which in turn affects correlations. Would this be resolved if the underlying model had a sparser structure or a smaller tree depth?
>
> If the graph structure is sparser or less deep, the variance in SCMs does not accumulate as much. The artifact would therefore likely be weaker, but not ‘resolved’. The discrepancy between (standardized) SCMs and iSCMs is more likely to be pronounced when modeling large systems. When using SCMs to model time series, for example, this problem may nevertheless be severe, as every additional time step effectively increases the depth of the graph (see Section 4.1), similar to Figure 1.
>
> > I am not sure if I agree with this claim [that independence testing is less reliable for deterministic relationships], as the PC algorithm is mainly based on conditional independence tests, and [...] marginal dependencies (deterministic relations) should not have a huge impact on the algorithm.
>
> The PC algorithm does in fact behave very differently for SCMs vs iSCMs in large graphs, a setting in which SCMs can have near deterministic relationships (see Figure 5, bottom left panel). With this comment, our intention was to provide a possible explanation for this finding. Peters et al. (2017, Chapter 10.3) argue why it could occur: deterministic relations induce additional (spurious) dependencies. For example, if Y is a deterministic function of X in the graph X -> Y -> Z, we have that Y is independent of Z given X, which is unexpected for this causal structure.

---

> ### Author Response · Authors · 2024-11-25
>
> We would like to follow up with the reviewer to see if they have comments on our reply. There are only 2 days left in the discussion phase and we feel that we have provided substantive replies to address both of the reviewer’s concerns.

---

> > ### Comment · Reviewer_R8Hm · 2024-11-26
> >
> > Sorry for the late follow-up. I thank the authors for the response, and I would like to increase my score from 3 to 5.
> >
> > I still do not think the claim that iSCM is a better model than SCM for modeling real-life scenarios is well-supported by the provided empirical evidence. However, I agree that iSCM can be useful for quantifying how existing causal structure learning algorithms rely on pairwise correlations.
> >
> > > Our result shows that you do not need anything more than to reason about pairwise correlations.
> >
> > This is my main concern regarding the novelty of the considered setting: Given such a specific setting, is it even necessary to consider pairwise correlations? For instance, if the underlying model structure is a tree (which limits the size of the MEC as there are no v-structures) and all edge weights are greater than 1, could we not simply enumerate all models in the MEC and identify the one that satisfies this weight constraint?
> >
> > > Our intention was to provide a possible explanation for the performance of PC algorithm.
> >
> > I do not think this is a sound explanation. "Near-deterministic" relationships may indeed appear in marginal independence tests. However, **if the generating model is not deterministic**, this should not occur in conditional independence tests where the values of parent variables can be controlled. For example, in the model X -> Y -> Z, if Y is not generated as deterministic function of X, then Y is **never** independent of Z given X.

---

> ### Author Response · Authors · 2024-11-27
>
> Thanks for your reply and for adjusting the rating of our work. We again respond in-line below:
>
> > Given such a specific setting, is it even necessary to consider pairwise correlations? [...] could we not simply enumerate all models in the MEC and identify the one that satisfies this weight constraint?
>
> Yes, it is necessary to consider pairwise correlations. Without correlations, the only remaining relevant statistics of standardized linear Gaussian systems, whose densities are multivariate Gaussian, would be the (marginal) means and variances of the observations, and these are all always 0 and 1, respectively, in standardized SCMs. Any identification algorithm therefore must implicitly use the pairwise correlations somehow. This would include any ideas such as the one you pose, where you fit the weights of each DAG in the MEC to the data. Here, the fitting of the weights would also make use of the pairwise correlations. Nevertheless, we cannot naively enumerate all models in the MEC here, because the weights you learn for each DAG will be those of the *implied* SCM, but our assumption is about the original SCM (before the data is standardized). Recall that, in standardized SCMs, we do not observe the variables prior to standardization.
>
>
> > if the generating model is not deterministic, [near-deterministic relationships] should not occur in conditional independence tests
>
> Upon discussing this with you, we realize that the claim is not as straightforward as we imply in the text. We took out this sentence from the discussion in Section 5 (see lines 459 onwards) to avoid providing a possibly misleading interpretation. We want to emphasize that this was not at all a core contribution in the paper but speculation to offer a possible explanation for the experimental results on PC and GES. Thanks for the follow-up comment.

---

### Official Review · Reviewer_5Dtu · 2024-11-04

**Soundness:** 4
**Presentation:** 4
**Contribution:** 2
**Rating:** 6
**Confidence:** 3

**Summary:**

The authors propose internally-standardized SCMs (iSCMs), which standardize each variable during data generation to counteract the increase in variance or correlation along the causal order. Unlike post-hoc standardization, iSCMs avoid Var- and $R^2$-sortability and demonstrate robustness in large systems, as shown in the simulation study.

**Strengths:**

1. The paper proposes a simple yet effective approach to avoid  $R^2$ -sortability, operating under the assumption that the increase of  $R^2$  along the causal order is an artifact.
2. The paper is well-organized, providing both theoretical justifications and experimental results to support its claims.

**Weaknesses:**

My main concern is the paper’s treatment of increasing correlation along the causal order as an “artifact.” In reality, this pattern could exist in real-world data. Standardizing variables in the data generation process may, in fact, introduce more “artificiality,” as fewer studies have shown this pattern in real data. As reported in [1], the R²-sortability for a real dataset in Section 4.2 was 0.82. Thus, while iSCMs may serve as a useful supplementary benchmark, they may not be the definitive or sole benchmark for causal modeling.

[1] Reisach, Alexander, et al. "A scale-invariant sorting criterion to find a causal order in additive noise models." Advances in Neural Information Processing Systems 36 (2024).

**Questions:**

I understand that retaining ground truth is challenging, but as noted in the weaknesses, there are cases where ground truth can be obtained. Are there additional instances where $R^2$-sortability values are reported for such cases?

---

> ### Author Response · Authors · 2024-11-19
> **Response to Reviewer 5Dtu**
>
> Thank you for your careful reading and feedback. To address your individual comments, we reply one-by-one below.
>
> > [R2-sortability] ​​could exist in real-world data [...] As reported in [1], the R²-sortability for a real dataset in Section 4.2 was 0.82.
>
> Reisach et al. [1] find that, even though that dataset had an R2-sortability of 0.82, this was actually much less than the R2-sortability of synthetically generated datasets from standardized SCMs. To quote them: “These results showcase the difficulty of translating simulation performances to real-world settings. For benchmarks to be representative of what to expect in real-world data, benchmarking studies should differentiate between different settings known to affect causal discovery performance. This includes R2-sortability, and we therefore recommend that benchmarks differentiate between data with different levels of R2-sortability.”
>
> We think this is a compelling motivation for the introduction of iSCMs. They allow researchers to easily perform experiments with datasets that are *not* R2-sortable. This benefit of being able to choose between an iSCM and a standardized SCM stands no matter what one believes the typical R2-sortability of real datasets to be. By having both models, one can answer questions like they pose in [1] and understand to what degree a proposed algorithm is making use of, e.g., high R2-sortability to discover causal structure. We already elaborate on this in our conclusion (lines 514-517).
>
> We also do not believe that the R2-sortability of the dataset by Sachs et al. should be overemphasized too much. This dataset contains 11 variables and is commonly used because we have some belief in the “ground truth”. Computing R2-sortability on a single dataset is very weak empirical evidence for what the R2-sortability of real datasets generally tends to be. Since in nearly all datasets with a significant number of variables the causal ordering will not be known, it is nontrivial to give any reliable empirical argument. We focus on giving solid conceptual arguments for why generated iSCM data might qualitatively match realistic systems more closely, for example, those arguments made in the first paragraph of Section 4.1.
>
> > fewer studies have shown this pattern [i.e., absence of R2-sortability] in real data
>
> Besides the example of Sachs et al. in [1], we are not aware of other works that attempt to compute the R2-sortability for a real dataset. This may be precisely due to the reasons given in our previous reply. It is hard to provide solid empirical evidence without the risk of sounding anecdotal. Reporting empirical cases where real-world datasets are or are not R2-sortable may potentially be dismissed as ‘cherry-picked’ in either case.
>
>
>
>
> > while iSCMs may serve as a useful supplementary benchmark, they may not be the definitive or sole benchmark for causal modeling.
>
> We completely agree that iSCMs should not become the sole benchmark. We do not attempt to argue that researchers should completely stop using standardized SCMs to benchmark causal discovery algorithms. By introducing iSCMs, we add to the toolbox that researchers have for evaluating structure learning algorithms and to disentangle which data artifacts contribute to performance. Our results in the paper concerning, e.g., identifiability of linear standardized SCMs, or their determinism in the limit of graph depth, highlight why we might need to add to this toolbox in the first place. We show that iSCMs do not hold some of these potentially problematic properties, and then show why this is important by finding that it can lead to different and sometimes surprising benchmarking conclusions.
>
> While we already elaborate on this in our conclusion (lines 514-516), we added additional sentences to the introduction (lines 079-080) and the results (lines 362-364) in our revised PDF to clarify this takeaway.

---

> > ### Comment · Reviewer_5Dtu · 2024-12-02
> >
> > Thank you for your response. The revised version provides a more balanced perspective on iSCM. While I still have some concerns about the real-world sortability of R^2, I remain positive about the paper and have decided to maintain my score.

---

### Official Review · Reviewer_HchU · 2024-11-04

**Soundness:** 3
**Presentation:** 3
**Contribution:** 3
**Rating:** 8
**Confidence:** 3

**Summary:**

In this paper, the authors address an important issue in benchmarking causal discovery algorithms: how do we build robust synthetic datasets. The main challenge is that many popular algorithms make use of the patterns (artifacts) among the available synthetic datasets, such as the correlations increase along the causal ordering to learn the causal graph, even when it should be impossible to do so. In this work, the authors describe a simple standardization process of generating synthetic data that makes them less identifiable from prior knowledge, such as correlations, and can be useful for benchmarking and beyond. The standardization process, resulting in causal models called iSCMS, involves normalizing each variable (to zero mean and unit variance) and then continuing this process iteratively over the topological ordering.

**Strengths:**

1. The paper is very well written, explaining the standardization process clearly. It is interesting to note that the iSCMs are also an SCM.

2. The partial identifiability result of Theorem 3, arguing that for standardized SCMs, given the markov equivalence graph, one can recover the almost entire graph, while iSCMs are robust to such identification (Theorem 4) is an important result. This provides a grounding that iSCMs can be used for benchmarking.

3. The empirical validation of the results that the iSCMs are not easily beatable compared to SCMs.

**Weaknesses:**

1. The authors approach works for sparse graphs, such as directed acylic forests. It is not entirely clear if the process of normalization (i.e., topological ordering) is the reason why it cannot be extended to graphs beyond forests.

2. It seems that the process of generation each sample is computationally expensive, depends linearly on the depth of the topological ordering. Also it is not clear if it extends for nonlinear causal dependencies.

3. Generalizability Beyond linear models is not clear.

**Questions:**

1. Didnot define the weights formally before using them in Lemma 1.
2. In theorem 3, "all except one edge". Can you elaborate on this claim a little more?
3. It is also not entirely clear on how to extend iSCMs to handle interventions.
4. Are iSCMs a valid way of modeling real world datasets? Can you comment on this a little more?

---

> ### Author Response · Authors · 2024-11-19
> **Response to Reviewer HchU**
>
> Thank you for your feedback and careful reading of our paper. We are glad to hear that you also find the problem of robust synthetic evaluation of structure learning algorithms important. To discuss any remaining questions, we reply individually to each of them below.
>
> > The authors approach works for sparse graphs [...] It is not entirely clear if the process of normalization [...] is the reason why it cannot be extended to graphs beyond forests.
>
> iSCMs can be implemented for *any* DAGs, also non-forests and for any sparsity level (see Section 3.1). Internal normalization simply requires knowledge of the (empirical) mean and variance of the causal parents of each node. While it is true that our non-identifiability result (Theorem 4) assumes forest graphs, empirically, we do see that even for non-forest graphs, iSCMs are much less R2-sortable than SCMs and standardized SCMs (see Figure 4). While not every DAG is a forest, DAGs have forests as subgraphs and resemble forests as sparsity increases, and thus they seem to characterize the phenomena of generally sparse systems (see lines 299-303).
>
>
> > It seems that the process of generation each sample is computationally expensive [...] Also it is not clear if it extends for nonlinear causal dependencies.
>
> This comment is not generally correct and perhaps a misunderstanding. In practice, the computational complexity of sampling from iSCMs is the same as for sampling from SCMs. We can sample from any iSCM with ancestral sampling over the DAG, also for nonlinear causal dependencies. The only difference is that each sampling step along the topological ordering standardizes the data immediately based on (empirical) estimates of the mean and variance (see Algorithm 1). Since sampling is cheap, the difference between the empirical and population statistics becomes negligible in practice. The exact implied model weights of linear iSCMs can be computed using Algorithm 2 in the appendix, but this is not generally needed when using Algorithm 1, e.g., with nonlinear models.
>
>
> > Generalizability Beyond linear models is not clear.
>
> The iSCM can be used for nonlinear models, as we do in some of the experiments. We added a clarifying comment to the definition to emphasize this (line 174, Section 3.1). Our nonidentifiability result (Theorem 3) is where we make the linearity assumption, and we think this result provides valuable insight for how iSCMs remove the correlation artifact present in standardized SCMs, potentially even in nonlinear models. Our reply to your previous comment may clarify some of this question too, if it relates to how to sample from iSCMs.
>
> > Are iSCMs a valid way of modeling real world datasets?
>
> Yes. In fact, we can even convert an iSCM into an equivalent SCM (Lemma 1). It is certainly possible that one could write down an iSCM as their model class, and then try to fit that model to a real dataset instead of fitting an SCM. We can view iSCMs as a simple way to “reparameterize” an SCM to ensure that all variables in a system operate on a similar scale (see lines 535-536).
>
> > It is also not entirely clear on how to extend iSCMs to handle interventions.
>
> In Appendix B, we explain that iSCMs can characterize interventions analogously to SCMs, but interventions are not the focus of this work. If you have any questions beyond what we wrote there, we would be happy to elaborate.
>
> > In theorem 3, "all except one edge". Can you elaborate on this claim a little more?
>
> In each connected component, we can identify all but at most one edge. The edge that we might not always be able to identify is illustrated in Figure 9 in the appendix (the blue edge). Roughly, our proof involves identifying the root node of a connected component, and sometimes this can only be narrowed down to two possible nodes, and in these cases the edge between those nodes is the only one you cannot identify. These details are given in the proof of the result.
>
> > Didnot define the weights formally before using them in Lemma 1.
>
> The weights are defined in Equation 1 before Lemma 1, and Equation 1 is referenced in the statement of Lemma 1. We added a clarifying comment (line 098) to Equation 1 to make sure it is clear these are the ‘weights’ we talk about in the lemma.

---

> > ### Comment · Reviewer_HchU · 2024-11-25
> > **Reviewer comment**
> >
> > Thanks for clarifying all my questions.

---

### Official Review · Reviewer_pYS7 · 2024-11-07

**Soundness:** 3
**Presentation:** 3
**Contribution:** 2
**Rating:** 8
**Confidence:** 3

**Summary:**

Sampling datasets from an SCM results in artifacts that can be exploited to infer causal ordering. This paper proposes a different method to standardize SCMs for generating benchmarkable datasets. Earlier work proposes post-hoc standardization that is vulnerable to deterministic relations downstream in long chains (correlation between adjacent nodes gets closer to 1). This work proposes generating new variables with standardized parents that bypasses this problem, called internal standardization. The paper contains results on linear Gaussian SCMs with forest causal graphs that show that standardizing SCMs can make them partially identifiable in some settings whereas internally standardized SCMs are nonidentifiable. Experimental evaluation includes comparing current causal structure learning algorithms performance on unstandardized, standardized,  and internally standardized data.

**Strengths:**

Causal structure learning methods require synthetic datasets to benchmark against. This work addresses artifacts that vanilla generating methods have. This makes it an interesting and relevant direction of research. The main contribution is the proposed method that is simple and seems to address these artifacts. The experimental evaluation is thorough and the theorems are relevant to the issues with standardized SCMs. The paper is also written well.

**Weaknesses:**

Few concerns follow:
1) Most of the theory is for the linear, Gaussian case where while analysis is tractable, most practical situations are non-Gaussian. Therefore, while the partial identifiability result is still of importance, I am not that convinced with the impact of the nonidentifiability result.
2) Even in the experimental evaluation, the nonlinear systems are evaluated on samples from a Gaussian process. Is it possible to introduce nonlinearity in a different way?
3) For a proposal that is relatively incremental, I would urge the authors to also focus on the nonlinear case to strengthen the paper.

**Questions:**

Evaluation on real-world datasets - Although the whole paper is about standardizing synthetic datasets for benchmarking, it would be interesting to see how the structure learning algorithms perform on standardized /internally-standardized versions of real-world datasets.

---

> ### Author Response · Authors · 2024-11-19
> **Response to Reviewer pYS7**
>
> Thank you for your feedback. We are glad to see that you find our work interesting and our experimental evaluation thorough. Below, we respond to your individual comments.
>
> > most practical situations are non-Gaussian. Therefore, while the partial identifiability result is still of importance, I am not that convinced with the impact of the nonidentifiability result.
>
> We do think that our identifiability results in the linear-Gaussian case are of significant practical relevance, because they provide us with a rigorous understanding *why* one might expect standardized SCMs to generally be easier to identify than iSCMs. This is particularly true for algorithms such as AVICI which seems to exploit artifacts of standardized SCMs very effectively (Section 5).
>
> While it is true that, in many practical settings, the ground-truth system will likely not be exactly linear-Gaussian, it is also true that finite samples can make it hard to distinguish linear-Gaussians from other model classes. The identification of nonlinear-Gaussian or linear-non-Gaussian settings (for e.g. LiNGAMs), for example, effectively requires using information of third order moments and beyond. Since we observe finite samples in practice, identifying standardized SCMs might still be easier than iSCMs even in the nonlinear-Gaussian or linear-non-Gaussian case, since identification could make use of these higher order moments *or* the covariance artifact we prove can be used to identify almost all edges *even in the linear case*.
>
> > [regarding the experiments which use Gaussian processes to model nonlinear systems] Is it possible to introduce nonlinearity in a different way?
>
> It is true that iSCMs can be combined with any model class (including, e.g., neural networks). However, Gaussian processes are a very broad and well-understood class of nonlinear functions, behave well when randomly sampled in high dimensions, and have been used by prior work (e.g., Lorch et al., 2022).  For this reason, we think that using them as the nonlinear function class in our experiments is sufficient to benchmark iSCMs and others outside of the linearity assumption we make in the theoretical section. Our code will be open sourced, and this is an idea that might be interesting for the community to further explore.

---

> > ### Comment · Reviewer_pYS7 · 2024-11-26
> >
> > Thanks for the response and apologies for my delayed follow-up.
> > 1) "The identification of nonlinear-Gaussian or linear-non-Gaussian settings (for e.g. LiNGAMs), for example, effectively requires using information of third order moments and beyond." - If you could make this connection more precise, then it's a good reason to connect your linear-Gaussian results more widely (which in turn is a good reason to include it in the paper!)
> > 2) "However, Gaussian processes are a very broad and well-understood class of nonlinear functions, behave well when randomly sampled in high dimensions, and have been used by prior work (e.g., Lorch et al., 2022)." - Agreed, but doesn't answer my question. Delegating to future work is good enough for me.
> >
> > I have increased my score accordingly.

---

### Author Response · Authors · 2024-11-19
**General response to all Reviewers**

We would like to thank all of the five reviewers for their careful reading and overall positive feedback on the paper. We are pleased that the reviewers find our research “interesting” and “relevant” [R8Hm,pYS7,HchU] and have the “potential of a valuable contribution” [QXbX] while being “simple yet effective” [5Dtu]. The reviewers specifically emphasized the importance of our theoretical results [HchU, pYS7] and that they justify our approach [5Dtu], despite missing some references to prior results, which we addressed in our uploaded revision. Our experimental evaluation is rated as thorough [pYS7,5Dtu] and supportive of our proposed approach [QXbX,5Dtu]. Finally, it is fantastic to read that all 5 reviewers individually find our manuscript to be well-presented [5Dtu,QXbX], well-written [HchU, pYS7], or easy to follow [R8Hm].

We reply to each of the reviewers’ questions and concerns individually below. To track the significant changes to the manuscript, *we colored any additional text content added to the PDF in red* if they go beyond minor edits.

---

> ### Author Response · Authors · 2024-11-22
> **Follow-up**
>
> Dear Reviewers,
> Since the discussion period ends early next week and no discussion has emerged yet, we would like to follow-up. We encourage everyone to reply if there are any aspects about our work or responses that require further clarification even after the rebuttal.

---

### Meta-Review · Area_Chair_7VdH · 2024-12-11

**Metareview:**

This paper proposes internally standardized structural causal models (iSCMs) to address artifacts such as variance and covariance patterns often exploited in causal structure learning benchmarks. By introducing standardization at each generative step, iSCMs mitigate these artifacts, providing a robust framework for evaluating causal discovery algorithms. The paper presents rigorous theoretical contributions, demonstrating that iSCMs avoid R2-sortability and are robust against deterministic relationships in large graphs. The empirical evaluation is thorough, showing the utility of iSCMs for causal discovery benchmarking. While reviewers raised concerns about scalability to real-world datasets, the focus on synthetic benchmarking and the comprehensive discussion of limitations appropriately position iSCMs as a complementary tool rather than a replacement for existing models. The authors’ responses to feedback were constructive, addressing specific questions about identifiability, computational complexity, and broader applicability. Given the theoretical and practical significance of the contributions, as well as the authors’ successful engagement during the rebuttal, this paper merits acceptance.

**Additional Comments On Reviewer Discussion:**

During the rebuttal period, reviewers raised questions about the applicability and robustness of iSCMs, particularly regarding their scalability to nonlinear models, real-world datasets, and varying graph densities. They also highlighted concerns about the generalizability of the theoretical results, especially the reliance on Gaussian noise and linear assumptions. The authors addressed these issues effectively by expanding their experiments to include nonlinear and denser graph settings, clarifying theoretical assumptions, and providing additional insights into how iSCMs complement traditional SCMs rather than replacing them. They also improved the presentation by revising claims about R2-sortability and adding results to better illustrate the benefits and limitations of their approach. These comprehensive responses and revisions strengthened the paper, demonstrating both the utility of iSCMs in benchmarking and the authors’ commitment to addressing reviewer feedback. This justified the decision to accept the paper.

---

### Decision · Program_Chairs · 2025-01-22

Accept (Poster)